# MixEval-X: Any-to-Any Evaluations from Real-World Data Mixtures

**Jinjie Ni**[a†], **Yifan Song**[c], **Deepanway Ghosal**[f*], **Bo Li**[b], **David Junhao Zhang**[a],
**Xiang Yue**[d], **Fuzhao Xue**[a], **Zian Zheng**[e], **Kaichen Zhang**[b], **Mahir Shah**[a], **Kabir Jain**[a],
**Yang You**[a], **Michael Qizhe Shieh**[a]

[a]National University of Singapore, [b]Nanyang Technological University, [c]Peking University,
[d]Carnegie Mellon University, [e]University of Waterloo, [f]Independent Researcher

https://mixeval-x.github.io/

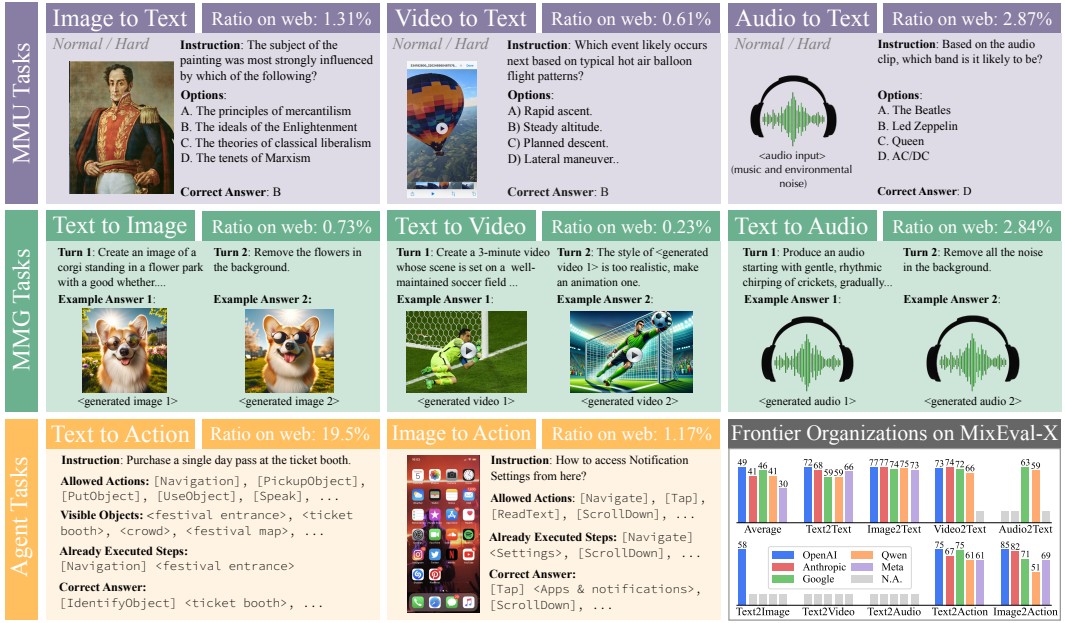

Figure 1: `MixEval-X` encompasses **eight input-output modality combinations** and can be further extended. Its data points reflect **real-world task distributions**. The last grid presents the scores of frontier organizations' flagship models on `MixEval-X`, normalized to a 0-100 scale, with MMG tasks using win rates instead of Elo. Section D presents example data samples and model responses.

## Abstract

Perceiving and generating diverse modalities are crucial for AI models to effectively learn from and engage with real-world signals, necessitating reliable evaluations for their development. We identify two major issues in current evaluations: (1) inconsistent standards, shaped by different communities with varying protocols and maturity levels; and (2) significant query, grading, and generalization biases. To address these, we introduce `MixEval-X`, the first any-to-any, real-world benchmark designed to optimize and standardize evaluations across diverse input and output modalities. We propose multi-modal benchmark mixture and adaptation-rectification pipelines to reconstruct real-world task distributions, ensuring evaluations generalize effectively to real-world use cases. Extensive meta-evaluations show our approach effectively aligns benchmark samples with real-world task distributions. Meanwhile, `MixEval-X`'s model rankings correlate strongly with that of crowd-sourced real-world evaluations (up to 0.98) while being much more efficient. We provide comprehensive leaderboards to rerank existing models and organizations and offer insights to enhance understanding of multi-modal evaluations and inform future research.

---

[†]Correspondence to: Jinjie Ni <jinjieni@nus.edu.sg>
[*]Now at Google DeepMind.

# 1 INTRODUCTION

Evaluations are crucial in the AI community for two main reasons: (1) they provide early signals to developers, assisting in the refinement of data and model, and (2) they guide users in selecting appropriate models for specific tasks. Thus, evaluations offer feedback signals to the entire community, driving model optimization.

Recently, models with diverse input (Achiam et al., 2023; Xue et al., 2024; Chu et al., 2024) and output (Betker et al., 2023; Yang et al., 2024; Majumder et al., 2024) modalities have been developed, with evaluations tailored to each. However, these communities often evolve in isolation, resulting in **significant disparities in evaluation standards and methods**. For example, while the large language model (LLM) community has hundreds of multi-task evaluations spanning various domains and methodologies, the audio language model community still relies heavily on task-specific benchmarks (Chu et al., 2024). This fragmentation results in inconsistent evaluation signals, misleading and bottlenecking the overall progress of various modalities.

Additionally, existing evaluations exhibit significant biases in **query**, **grading**, and **generalization**. Query bias occurs when evaluation tasks deviate from real-world task distributions, leading to discrepancy in evaluation results and real-world performance; grading bias arises from unfair scoring paradigms, and generalization bias stems from evaluation contamination. These biases skew evaluation signals, hindering model development. To address these issues, MixEval (Ni et al., 2024) proposes a low-bias paradigm for LLM evaluations (Text2Text). MixEval aligns benchmarks with real-world task distributions by matching web-mined queries to similar benchmark tasks. Its benefits include: (1) a comprehensive, less biased task distribution based on a large-scale web corpus, (2) fair grading due to the ground-truth-based paradigm, (3) dynamic benchmarking via a low-effort update pipeline, mitigating generalization bias, (4) accurate model ranking with a 0.96 correlation to Chatbot Arena, (5) fast, cost-effective, and reproducible execution, requiring only 6% of MMLU's time and cost, and (6) challenging problems with significant room for improvement.

To this end, to optimize and standardize evaluations across AI communities, we propose `MixEval-X`, the first any-to-any real-world benchmark optimizing benchmark mixtures for a wide range of input-output modalities. `MixEval-X` consists of eight subsets, each with distinct input-output modality combinations, categorized into three types: **multi-modal understanding (MMU)**, **multi-modal generation (MMG)**, and **agent** tasks (Figure 1). These modalities are not only the dominant ones in web queries but also central to various communities.

Specifically, we first use MixEval's web user query detection pipeline to gather a well-distributed set of real-world queries spanning diverse input-output modalities. For MMU tasks, we construct large-scale multi-modal benchmark pools from existing community benchmarks, prioritizing query-based ones like question-answering and examination tasks due to: (1) the query-based nature of real-world tasks, and (2) the convergence of AI tasks toward natural language queries for multi-task learning (Wei et al., 2021). We then match web queries to similar query-based tasks from the benchmark pool to reconstruct the benchmark distribution. This is followed by an automatic quality control step to eliminate the wrong or extreme samples. Additionally, we perform rejection sampling to select more challenging MMU tasks while preserving real-world distribution alignment. For MMG tasks, which are open-ended, we implement an adaptation-rectification pipeline where the adaptation step creates real-world tasks from web queries using frontier models, ensuring alignment with real-world task distributions, and the rectification step automatically fixes errors and distribution deviations. For agent tasks, lacking general-domain benchmarks, we use a similar adaptation-rectification pipeline to recreate task distributions and annotate reference answers. Optional human inspection was adopted to increase the annotation quality. The efficient evaluation creation pipelines for MMU, MMG, and agent tasks allow periodic data refreshes to mitigate contamination. Our meta-evaluations show that: (1) `MixEval-X` data closely aligns with real-world task distributions, and (2) `MixEval-X`'s evaluation results strongly correlate with real-world user-facing evaluations (up to 0.98), while being significantly more efficient.

**Why use MixEval-X?** (1) It extends all the benefits of MixEval to multi-modal evaluations, including comprehensive and less biased query distribution; fair grading (except open-ended tasks); dynamism; accurate model ranking; fast, cost-effective, reproducible execution; and challenging nature. (2) It establishes unified, high standards across modalities and communities. For single-modality models, it ensures its evaluation keeps up with the state-of-the-art standards; for multi-

modality models, it ensures consistent, high-standard evaluations across modalities, preventing any from becoming a bottleneck. (3) Beyond model evaluation, `MixEval-X` benchmarks different organizations (Figure 1) with balanced dimensions (modalities), unlocking a new level of evaluation.

**Research Contributions** (1) We propose the multi-modal benchmark mixture and adaptation-rectification pipeline to optimize AI evaluations, providing an efficient approach to create low-bias any-to-any benchmarks with real-world distributions. (2) We introduce `MixEval-X`, the first high-standard, unified real-world benchmark with diverse input-output modalities, reducing the bias and heterogeneity in AI evaluations. (3) We present comprehensive evaluation results, reranking models and organizations for a wide range of communities. (4) We conduct extensive meta-evaluations, offering valuable insights for guiding AI evaluations and future research.

## 2 MIXEVAL-X

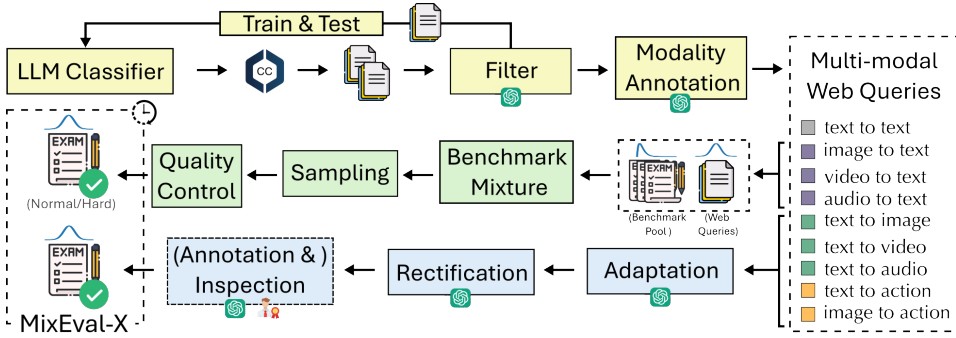

Figure 2: The overall pipeline for creating `MixEval-X`.

In this section, we introduce the methods used to construct the various subsets of `MixEval-X` and their respective grading mechanisms. As shown in Figure 2, MMU tasks are built with benchmark mixture, while MMG and agent tasks are created with an adaptation-rectification pipeline, both designed to align evaluation tasks with real-world distributions. The grading for MMU tasks is robust due to their ground-truth-based nature. All sub-benchmarks are dynamic, enabled by the efficient data creation pipelines. Moreover, we carefully refine annotation accuracy and task difficulty to ensure the quality and usage potential of `MixEval-X`.

### 2.1 WEB QUERY DETECTION

We create the same web query detection pipeline as MixEval to detect real-world user queries from Common Crawl (Computer, 2023). Both recall and precision are crucial to ensure the query distribution reflects real-world scenarios. Therefore, we developed two benchmarks to evaluate our query detector's performance. The first benchmark includes self-collected in-the-wild user queries as positive samples, with non-query datasets such as Wikipedia (Foundation, 2022) as negative samples. The second, higher-quality benchmark contains positive and negative samples hand-picked by our authors from in-the-wild query and non-query datasets. In preliminary experiments, direct prompting of open-source language models performed poorly on our benchmarks. Thus, we devised a loop to ensure high recall and precision cost-effectively. We started with a detection phase to gather training data. Testing various open-source LLMs, Vicuna 33B (Chiang et al., 2023) achieved a high recall (>99%) on our test sets with careful prompt engineering, ensuring that very few positive samples were missed initially. In this phase, we detected around 20k queries using Vicuna 33B over a subset of Common Crawl. We then used GPT-4 to more accurately label these data as positive or negative samples, and used the resulting data to train Vicuna 33B. The trained Vicuna 33B achieved high recall (>99%) and precision (>98%) on our benchmarks and detected 2M user queries from the Common Crawl subset. Finally, we prompted GPT-4 Turbo to further filter and classify them into various modalities, extracting well-distributed multi-modal queries for `MixEval-X` creation.

## 2.2 MMU Task Creation

**Benchmark Mixture** We perform benchmark mixture to mitigate query bias in MMU tasks. As illustrated in MixEval (Ni et al., 2024) and further in Figure 9, current benchmark query distributions deviate from real-world use, limiting the generalizability of evaluation outcomes. Using the MixEval web query detection pipeline, which trained precise query classifiers to extract well-distributed queries from the web, we crawl multi-modal user queries from web and map this web query distribution onto the constructed benchmark pool containing numerous ground-truth-based benchmarks. We sample problem-answer pairs from this benchmark pool by selecting the most similar one given a web query, constituting a new benchmark with natural ground-truths. The matching process is based on the similarities between the sentence embeddings (Reimers & Gurevych, 2019) of benchmark text queries and web queries. As such benchmarks exist only for MMU tasks, we apply benchmark mixture exclusively to these modalities. Due to varying community maturity, benchmark pools for certain modalities, such as Audio2Text, are less extensive compared to more established ones like Text2Text and Image2Text. However, a key advantage of `MixEval-X` is its capacity for self-refinement, enabling the benchmark pool to grow as the community develops. The benchmark pool composition is detailed in Section G.

**Challenge Set Sampling and Dynamism** The rapid advancement of frontier and open-source communities introduces two key challenges in model evaluation: saturation, where further score improvements are limited, and contamination, where models overfit to the test data. To enhance model differentiation, we applied rejection sampling (Ni et al., 2024) to select more challenging MMU tasks while preserving real-world distribution alignment. The effectiveness of this strategy is demonstrated later by the low scores on the hard split in Section 3.1, and the close distance between hard split queries to web queries in Figure 9. Since `MixEval-X`'s benchmark mixture pipeline is fully automated and updatable within minutes, refreshing data points is efficient and helps mitigate testset overfitting. Additionally, the benchmark pool also integrates newly released benchmarks to mitigate contamination. However, this dynamism alleviates but does not fully resolve contamination. Further discussions on benchmark mixture contamination can be found in Ni (2024).

**Quality Control** An inspection step is used to enhance the benchmark quality. We focus on the entries where frontier models erred most, excluding those that most models gave the same answer different from the ground-truth. Cases with extreme inputs were removed to streamline evaluation.

## 2.3 MMG and Agent Task Creation

**Adaptation-Rectification Pipeline** For MMG and agent tasks lacking natural general-domain benchmarks, alternative methods are needed to recreate real-world task distributions. MMG tasks are simpler to construct, being open-ended with no reference answers. Since web user queries are not clean and challenging enough for MMG models, we developed an adaptation-rectification pipeline that transforms them into well-formatted, challenging tasks. In adaptation, a language model modifies the query to match the required complexity and format while maintaining user intent. In rectification, the model inspects and corrects the task's logic, complexity, correctness, and alignment with the original task. The resulting MMG tasks have two turns–a generation turn that instructs the model to generate content and an edition turn that instructs the model to edit the generated content in the last turn.

Constructing agent tasks is more demanding, requiring careful annotation of reference answers. The task design follows the MMG approach, using the adaptation-rectification pipeline. To annotate answers for Text2Action and Image2Action tasks, frontier LLMs and VLMs provide initial annotations, refined through automated rectifications. Both MMG and agent tasks have an optional human review step. These sub-benchmarks are also dynamic due to the efficient data update pipeline. Detailed pipeline prompts are shown in Section E.

## 2.4 Grading

Grading bias undermines evaluation accuracy, even for ground-truth tasks with narrow answer spaces. As noted in Ni (2024), rule-based parsers are unstable when grading across multiple models and ground-truth-based benchmarks, while language model parsers provide a more reliable alternative. We use model-based parsers to grade tasks with narrower answer spaces, such as MMU and

Table 1: The statistics of `MixEval-X` subsets. All MMU tasks have both free-form and multiple-choice tasks except Audio2Text and Audio2Text-Hard.

| | Task Type | # Tasks | # Turns | Avg. # Toks per Query | Avg. # Inputs | Avg. Input Length | Min Input Length | Max Input Length | English Ratio |
|---|---|---|---|---|---|---|---|---|---|
| Image2Text | MMU | 2,000 | 1 | 12.1 | 1.0 | - | - | - | 99.2% |
| Image2text-Hard | MMU | 1,000 | 1 | 14.7 | 1.0 | - | - | - | 99.4% |
| Video2Text | MMU | 2,000 | 1 | 10.2 | 1.0 | 56.5 (s) | 1.5 (s) | 238.4 (s) | 100.0% |
| Video2Text-Hard | MMU | 1,000 | 1 | 10.8 | 1.0 | 70.7 (s) | 1.4 (s) | 238.4 (s) | 100.0% |
| Audio2Text | MMU | 1,000 | 1 | 8.2 | 1.0 | 40.2 (s) | 5.3 (s) | 146.5 (s) | 100.0% |
| Audio2Text-Hard | MMU | 500 | 1 | 9.2 | 1.0 | 54.6 (s) | 5.6 (s) | 149.5 (s) | 100.0% |
| Text2Action | Agent | 100 | 1 | 14.3 | 1.0 | 139.7 (toks) | 35 (toks) | 214 (toks) | 99.0% |
| Image2Action | Agent | 100 | 1 | 14.2 | 1.0 | 61.7 (toks) | 34 (toks) | 100 (toks) | 100.0% |
| Text2Image | MMG | 200 | 2 | 31.5 | - | - | - | - | 100.0% |
| Text2Video | MMG | 200 | 2 | 48.0 | - | - | - | - | 100.0% |
| Text2Audio | MMG | 200 | 2 | 54.5 | - | - | - | - | 100.0% |

agent tasks. For MMU, small language models assess answers given the problem, model response, and reference answers. For agent tasks, which have comparatively broader answer spaces, we use frontier LLMs to grade on a scale of 0-10, given the reference answer. MMG tasks, which are more open-ended, are harder to evaluate, where traditional automated metrics such as FID, CLIP, and FVD fail to capture nuanced quality and user preference (Jiang et al., 2024). Thus, we employ crowd-sourced workers to rank model pairs, computing Elo ratings using the Bradley–Terry (BT) model (Bradley & Terry, 1952), which is statistically robust for open-ended tasks (Chiang et al., 2024; Jiang et al., 2024). Alternative grading methods are also possible for MMG tasks. In Section 4.2, we explore the correlation between model evaluations and human preferences, advocating for more research into model-based MMG grading. Note that for MMG tasks, `MixEval-X` offers a task set that is highly representative of real-world use cases while remaining flexible in the grading methods. Users may choose between automatic metrics, model-based evaluation, or human judgment depending on the specific application. The grading prompts are presented in Section F.

Table 1 presents the statistics for the `MixEval-X` benchmarks. We regulate task count and input lengths for efficiency, especially for MMU tasks, which often require longer inputs. Input means the input contents other than the textual query. Input lengths are measured in seconds for Video2Text and Audio2Text, and in text tokens for other modalities (NLTK tokenizer (Loper & Bird, 2002)). MMG tasks generally have longer queries by design. Figure 40-45 details the distribution of the benchmark pool across MMU subsets.

## 3 EVALUATION

We present the evaluation results, and the settings are detailed in Section C.

### 3.1 MMU TASKS

**Image2Text** In Image2Text tasks, models generate language responses based on user-provided images and text. We evaluate a broad range of Image2Text models due to its established community. Figure 3 presents the leaderboard. Proprietary models like Claude, GPT, Gemini, and Reka series outperform open-source models, with Qwen and Llama leading the latter. Our analysis shows that input resolution limits and model size are key factors in rankings, while model architecture, training methods, and input formatting also influence performance. Most models use an encoder-decoder architecture, as decoder-only models remain underexplored in vision-language tasks.

**Video2Text** In Video2Text tasks, models generate language responses based on user-provided videos and text. All models are evaluated with the same number of frames, except for specialized models with frame limitations. The results are shown in Figure 4, with proprietary models again dominating. In addition to factors like model size and architecture, the maximum supported input frames significantly impact Video2Text performance. Models with limited frame capacity perform worse, especially on long video datasets like ActivityNet and EgoSchema, highlighting the need for further research in long video understanding.

| | Image2Text | Image2Text-Hard | SEED (Mixed) | MMMU (Mixed) | DocVQA (Mixed) | TextVQA (Mixed) | VisWiz (Mixed) | InfographicVQA (Mixed) | SEED-Hard (Mixed) | MMMU-Hard (Mixed) |
|---|---|---|---|---|---|---|---|---|---|---|
| Claude 3.5 Sonnet | **76.9** | **46.2** | **76.0** | 75.1 | **94.6** | 90.3 | 62.5 | 78.8 | 31.0 | **48.9** |
| GPT-4o | 76.6 | 45.8 | 75.6 | 74.1 | 87.4 | 90.9 | **66.9** | 79.0 | 29.3 | 45.9 |
| GPT-4V | 75.0 | 44.6 | 75.6 | 68.0 | 92.1 | 89.3 | 53.7 | 79.2 | 31.9 | 40.6 |
| Qwen2-VL-72B | 74.8 | 43.4 | 71.5 | 67.5 | 90.6 | 90.3 | 66.3 | 80.4 | 25.4 | 27.8 |
| Gemini 1.5 Pro | 74.2 | 42.2 | 72.2 | **77.2** | 85.6 | 86.8 | 63.7 | 76.7 | 29.7 | 44.4 |
| Llama 3.2 90B | 73.0 | 40.6 | 73.3 | 62.9 | 92.7 | 90.9 | 61.6 | **89.8** | 28.9 | 30.1 |
| InternVL2-26B | 71.5 | 41.5 | 71.5 | 55.8 | 90.3 | **91.2** | 58.2 | 70.2 | 32.3 | 28.6 |
| Claude 3 Opus | 69.5 | 41.1 | 72.0 | 66.5 | 84.2 | 86.7 | 56.9 | 66.9 | **34.9** | 44.4 |
| Qwen-VL-MAX | 69.2 | 37.5 | 70.0 | 68.5 | 83.1 | 87.2 | 53.1 | 66.1 | 27.6 | 37.6 |
| LLaVA-1.6-34B | 68.1 | 37.5 | 70.4 | 60.4 | 71.0 | 81.8 | 48.6 | 58.8 | 31.9 | 36.8 |
| Claude 3 Sonnet | 67.8 | 38.3 | 71.1 | 50.8 | 86.7 | 80.3 | 58.2 | 78.6 | 32.3 | 30.8 |
| Reka Core | 67.4 | 37.3 | 67.5 | 71.1 | 76.5 | 79.9 | 56.9 | 59.6 | 25.0 | 39.1 |
| Reka Flash | 67.4 | 36.6 | 73.6 | 53.8 | 71.3 | 76.8 | 59.6 | 62.5 | 32.8 | 23.3 |
| InternVL-Chat-V1.2 | 67.2 | 36.0 | 70.7 | 54.8 | 51.8 | 76.3 | 60.0 | 59.2 | 25.4 | 33.8 |
| Qwen-VL-PLUS | 67.0 | 35.9 | 66.2 | 56.9 | 84.1 | 83.1 | 57.5 | 52.7 | 19.8 | 27.1 |
| Claude 3 Haiku | 66.1 | 37.5 | 67.8 | 58.4 | 88.3 | 83.0 | 59.8 | 59.4 | 32.8 | 45.9 |
| Gemini 1.0 Pro | 66.1 | 35.0 | 67.6 | 60.9 | 70.3 | 81.3 | 55.7 | 51.8 | 29.3 | 39.8 |
| InternLM-XComposer2-VL | 62.1 | 33.6 | 66.9 | 40.6 | 54.7 | 74.9 | 56.3 | 46.5 | 23.9 | 24.8 |
| Yi-VL-34B | 58.5 | 30.6 | 68.0 | 53.8 | 21.5 | 59.7 | 53.3 | 41.4 | 27.6 | 29.3 |
| OmniLMM-12B | 58.2 | 29.2 | 67.3 | 54.8 | 42.3 | 70.2 | 48.6 | 26.9 | 31.9 | 32.3 |
| DeepSeek-VL-7B-Chat | 56.7 | 26.5 | 61.3 | 41.1 | 39.4 | 69.9 | 50.8 | 32.0 | 21.1 | 14.3 |
| Yi-VL-6B | 55.4 | 30.1 | 65.6 | 45.7 | 23.6 | 62.3 | 52.2 | 28.0 | 27.6 | 19.5 |
| InfiMM-Zephyr-7B | 53.7 | 29.4 | 62.5 | 44.2 | 21.9 | 46.1 | 46.1 | 27.6 | 26.7 | 25.6 |
| MiniCPM-V | 51.5 | 25.9 | 59.1 | 32.0 | 53.2 | 76.6 | 40.8 | 32.2 | 23.7 | 18.0 |
| Marco-VL | 50.5 | 24.3 | 56.0 | 37.1 | 48.2 | 58.1 | 37.3 | 40.6 | 19.0 | 27.8 |
| LLaVA-1.5-13B | 50.2 | 26.0 | 56.9 | 32.5 | 22.4 | 53.7 | 42.9 | 24.3 | 19.0 | 24.8 |
| SVIT | 49.9 | 25.4 | 59.1 | 35.5 | 19.9 | 51.2 | 42.9 | 27.8 | 27.6 | 15.8 |
| mPLUG-OWL2 | 48.9 | 22.5 | 57.5 | 28.9 | 26.9 | 59.7 | 39.8 | 29.4 | 28.0 | 10.5 |
| SPHINX | 47.5 | 23.8 | 54.5 | 39.1 | 16.4 | 51.0 | 41.4 | 24.5 | 19.8 | 18.0 |
| InstructBLIP-T5-XXL | 46.2 | 21.5 | 58.0 | 31.0 | 11.2 | 41.7 | 44.3 | 24.5 | 19.4 | 28.6 |
| InstructBLIP-T5-XL | 45.5 | 22.9 | 53.1 | 32.0 | 14.5 | 44.5 | 44.5 | 12.9 | 21.1 | 18.8 |
| BLIP-2 FLAN-T5-XXL | 45.2 | 21.6 | 55.1 | 33.0 | 13.5 | 46.3 | 42.2 | 29.6 | 22.8 | 17.3 |
| BLIP-2 FLAN-T5-XL | 43.0 | 20.0 | 52.5 | 33.5 | 16.3 | 40.9 | 39.2 | 9.4 | 23.3 | 11.3 |
| Adept Fuyu-Heavy | 37.4 | 19.4 | 43.5 | 26.4 | 6.9 | 41.1 | 35.5 | 8.2 | 21.6 | 11.3 |
| LLaMA-Adapter2-7B | 36.6 | 20.4 | 42.5 | 32.5 | 15.6 | 23.7 | 44.5 | 25.1 | 18.1 | 14.3 |
| Otter | 34.1 | 18.5 | 42.5 | 31.5 | 5.3 | 17.9 | 21.2 | 21.4 | 23.3 | 9.8 |
| MiniGPT4-Vicuna-13B | 32.1 | 15.8 | 38.2 | 25.4 | 15.4 | 23.4 | 33.7 | 18.4 | 15.5 | 13.5 |

Figure 3: The evaluation results of prominent models on `MixEval-X` Image2Text, Image2Text-Hard, and their subsets. Proprietary models are highlighted in blue. See Section H for details.

| | Video2Text | Video2Text-Hard | ActivityNet-QA (Mixed) | HowToQA (Mixed) | TVQA (Mixed) | MSVD-QA (Mixed) | NextQA-freetext (Mixed) | TGIF-QA (Mixed) | ActivityNet-QA-Hard (Mixed) | TVQA-Hard (Mixed) |
|---|---|---|---|---|---|---|---|---|---|---|
| Claude 3.5 Sonnet | **74.2** | **45.5** | **73.3** | 76.6 | 64.8 | 79.4 | **76.4** | **78.9** | **60.4** | 39.4 |
| GPT-4o | 72.7 | 38.9 | 64.6 | **78.2** | 74.6 | 80.9 | 70.1 | 78.2 | 32.4 | 48.0 |
| Gemini 1.5 Pro | 71.8 | 38.1 | 65.2 | 64.8 | **82.9** | 74.4 | 75.7 | 43.2 | **68.5** |
| GPT-4V | 71.0 | 40.0 | 63.4 | **78.2** | 69.5 | 77.9 | 69.5 | 78.5 | 37.2 | 37.8 |
| Qwen2-VL-72B | 66.5 | 32.0 | 55.1 | 76.6 | 58.1 | 74.2 | 65.0 | 78.5 | 27.3 | 17.3 |
| Gemini 1.5 Flash | 66.3 | 33.9 | 59.0 | 67.4 | 70.3 | 73.8 | 61.4 | 72.3 | 26.7 | 51.2 |
| LLaVA-OneVision-72B-OV | 64.7 | 32.0 | 56.0 | 77.0 | 64.4 | 71.2 | 64.9 | 70.6 | 35.6 | 28.3 |
| Qwen2-VL-7B | 64.2 | 31.9 | 54.3 | 74.7 | 52.1 | 74.9 | 62.6 | 68.9 | 27.2 | 26.0 |
| LLaVA-Next-Video-34B | 63.1 | 28.4 | 56.1 | 68.6 | 62.7 | 74.0 | 62.8 | 68.0 | 26.7 | 38.6 |
| Claude 3 Haiku | 58.7 | 29.4 | 52.3 | 63.6 | 48.7 | 70.8 | 62.7 | 70.2 | 23.6 | 29.1 |
| LLaVA-Next-Video-7B | 58.7 | 27.2 | 53.2 | 62.1 | 44.5 | 72.5 | 61.0 | 74.4 | 25.9 | 33.1 |
| Reka-edge | 58.7 | 27.3 | 51.7 | 72.4 | 46.6 | 69.1 | 59.3 | 65.2 | 29.0 | 22.8 |
| LLaMA-VID | 55.6 | 24.8 | 52.9 | 60.9 | 36.0 | 72.8 | 61.3 | 67.1 | 19.1 | 17.3 |
| VideoLLaVA | 55.3 | 22.6 | 51.7 | 64.0 | 39.4 | 66.7 | 61.9 | 64.7 | 18.2 | 26.0 |
| Video-ChatGPT | 46.4 | 20.7 | 45.7 | 46.7 | 25.4 | 72.2 | 56.3 | 64.8 | 24.7 | 14.2 |
| mPLUG-video | 39.1 | 17.8 | 41.5 | 36.4 | 23.3 | 71.9 | 56.7 | 61.8 | 22.7 | 7.9 |

Figure 4: The evaluation results of prominent models on `MixEval-X` Video2Text, Video2Text-Hard, and their subsets. Proprietary models are highlighted in blue. See Section H for details.

**Audio2Text** In Audio2Text tasks, models generate language responses based on user-provided audio and text inputs. This field is less developed compared to vision tasks, leading to a smaller benchmark pool and fewer models. As shown in Figure 5, the Gemini series is the only proprietary model natively supporting Audio2Text, with Gemini 1.5 Pro ranking first at 62.7% accuracy on the general split but showing room for improvement on the Audio2Text-Hard split. Qwen2-Audio-7B leads on the Audio2Text-Hard split. Rankings are primarily influenced by language model quality and input formatting strategies.

## 3.2 MMG TASKS

The results for MMG tasks are shown in Figure 6. We employed hundreds of human evaluators from Amazon Mechanical Turk to assess model outputs using a pairwise-ranking approach, as automatic metrics fail to capture the nuances in output quality (Jiang et al., 2024). We report only the overall scores, while MMG tasks consist of two turns—a generation turn and an edition turn. Turn-level scores are presented in Figure 46. See Section H for model details.

| | Audio2Text | Audio2Text-Hard | Clotho-AQA (Mixed) | DAQA (Mixed) | Clotho-AQA-Hard (Mixed) | DAQA-Hard (Mixed) |
|---|---|---|---|---|---|---|
| Gemini 1.5 Pro | **62.7** | 24.0 | **67.4** | **53.4** | 26.8 | 21.7 |
| Gemini 1.5 Flash | 60.1 | 23.0 | 67.1 | 46.9 | 27.4 | 19.7 |
| Qwen2-Audio-7B-Instruct | 58.8 | 23.5 | 64.7 | 46.0 | 22.5 | 23.5 |
| Qwen2-Audio-7B | 56.6 | **24.6** | 63.1 | 44.0 | **29.9** | 20.0 |
| SALMONN-13B | 52.5 | 20.9 | 57.6 | 41.4 | 14.9 | **25.4** |
| Qwen-Audio | 52.4 | 16.0 | 61.5 | 33.8 | 19.0 | 12.8 |
| Qwen-Audio-Chat | 50.2 | 20.0 | 55.7 | 39.4 | 19.8 | 19.7 |
| SALMONN-7B | 38.9 | 17.1 | 46.6 | 22.2 | 20.6 | 11.6 |
| Pengi | 22.6 | 8.2 | 26.9 | 14.4 | 12.5 | 3.8 |

Figure 5: The results of prominent models on `MixEval-X` Audio2Text, Audio2Text-Hard, and their subsets. Proprietary models are highlighted in blue. See Section H for details.

**Text2Image** In Text2Image tasks, models generate images based on human prompts. Like Image2Text, Text2Image has a well-developed community, with Flux (BlackForestLabs, 2024) achieving the highest Elo score among the evaluated models, as shown in Figure 6(a). Our analysis reveals that image quality, particularly realism, significantly impacts human pairwise evaluations. Although DALL·E 3 HD (Betker et al., 2023) shows high quality, it ranks lower in realism. A case study is shown in Figure D. Instruction-following ability also strongly influences human evaluations.

**Text2Video** In Text2Video tasks, models generate videos based on textual prompts. Figure 6(b) presents the Elo rankings for various Text2Video models. Human evaluators tend to prefer videos with higher quality, realism, smoothness, and adherence to the prompt. Most models struggle with generating realistic human faces, and unnatural or distorted faces greatly reduce human preference. In contrast, video length has less influence; for example, Show-1 generates longer sequences, but face distortions significantly lower its preference among crowd-sourced evaluators.

**Text2Audio** In Text2Audio tasks, models generate audio based on textual prompts. Figure 6(c) shows the Elo rankings for Text2Audio models. AudioLDM2 ranks first in human pairwise evaluations, followed by Make-An-Audio-2 and Stable Audio. However, we find that our tasks are generally too challenging for most Text2Video and Text2Audio models, especially Text2Audio. Consequently, the Elo scores reflect only relative rankings, with even AudioLDM2 failing to produce high-quality audio or follow instructions effectively. This highlights a significant performance gap across different communities.

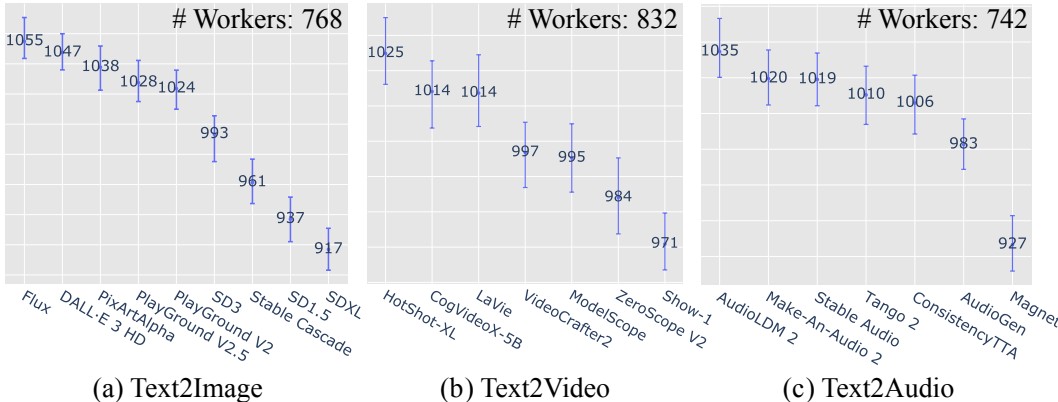

(a) Text2Image  (b) Text2Video  (c) Text2Audio

Figure 6: The overall Elo scores of MMG models on the `MixEval-X` MMG subsets, with error bars representing the 95% confidence intervals for the ratings. These scores are derived using the Bradley-Terry model, based on crowd-sourced user preferences. Additionally, the number of human evaluators per subset is provided for reference. The turn-level scores are shown in Figure 46.

### 3.3 AGENT TASKS

**Text2Action** Text2Action tasks involve models planning API-level actions based on textual inputs describing the environment and a user prompt. This setup simplifies real-world agent tasks to evaluate the action-planning capabilities of LLMs, offering more flexibility for optimizing task distribution. Figure 7 presents the results for the Text2Action subset. The model rankings differ from Text2Text

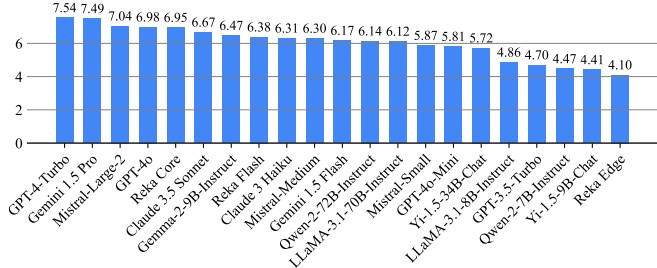

Figure 7: The evaluation results of prominent models on Text2Action. See Section H for details.

tasks (Ni et al., 2024), suggesting that strong text understanding does not guarantee proficiency in textual agent tasks.

**Image2Action** In Image2Action tasks, models with both image and text input capabilities plan API-level actions based on the observed environment (presented as an image) and the user prompt. Figure 8 presents the evaluation results for the Image2Action subset of `MixEval-X`. The rankings of vision-language models (VLMs) differ significantly from those in Image2Text tasks. Notably, some open-source models, not aligned with RLHF or similar techniques, often produce shorter or repeated action sequences, leading to lower scores.

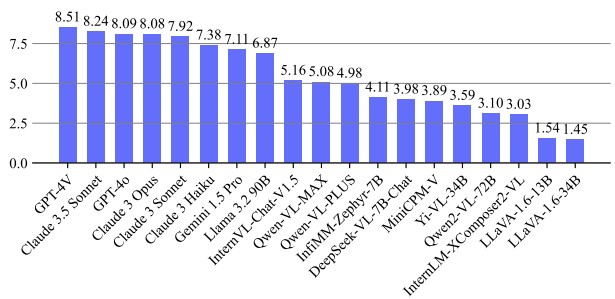

Figure 8: The evaluation results of prominent models on Image2Action. See Section H for details.

## 4 META EVALUATION

### 4.1 DISTRIBUTION ANALYSIS

**Setup** We aim to analyze the task distributions of `MixEval-X`. While many benchmarks and datasets are documented, we focus on their actual distributions in practice and their comparison to real-world tasks. In Figure 9, we randomly sample 1000 task queries from each dataset, reduce their sentence embeddings to 2D using t-SNE, and visualize the distributions. Dimensionality reduction is performed in the same space for benchmarks with the same modality, using identical color schemes to facilitate direct comparison, i.e., datasets of the same modality are comparable in terms of their distributions. To further examine topic distributions, we segmented the aggregated queries of each modality (e.g., Image2Text) into 16 spatial patches in the 2D space. From each patch, we uniformly sampled 100 queries and used GPT-4 to summarize the topics (Figure 35-39). MMG benchmarks are not analyzed as their prompts are caption-like, making them non-comparable.

**MixEval-X tasks closely align with real-world task distributions while being the most comprehensive and diverse.** In Figure 9, C-Dist measures the cluster distances between each benchmark and corresponding web queries. Benchmarks are ranked by proximity to web queries, with closer ones ranked higher. `MixEval-X` sub-benchmarks, including Image2Text, Video2Text, Audio2Text, Text2Action, Image2Action, and their hard versions, are the closest to web queries, indicating their distributions strongly resemble real-world tasks. Moreover, `MixEval-X` benchmarks visually cover more diverse topics than others. This also illustrates that both the multi-modal benchmark mixture and the adaptation-rectification pipeline effectively aligns benchmarks with real-world

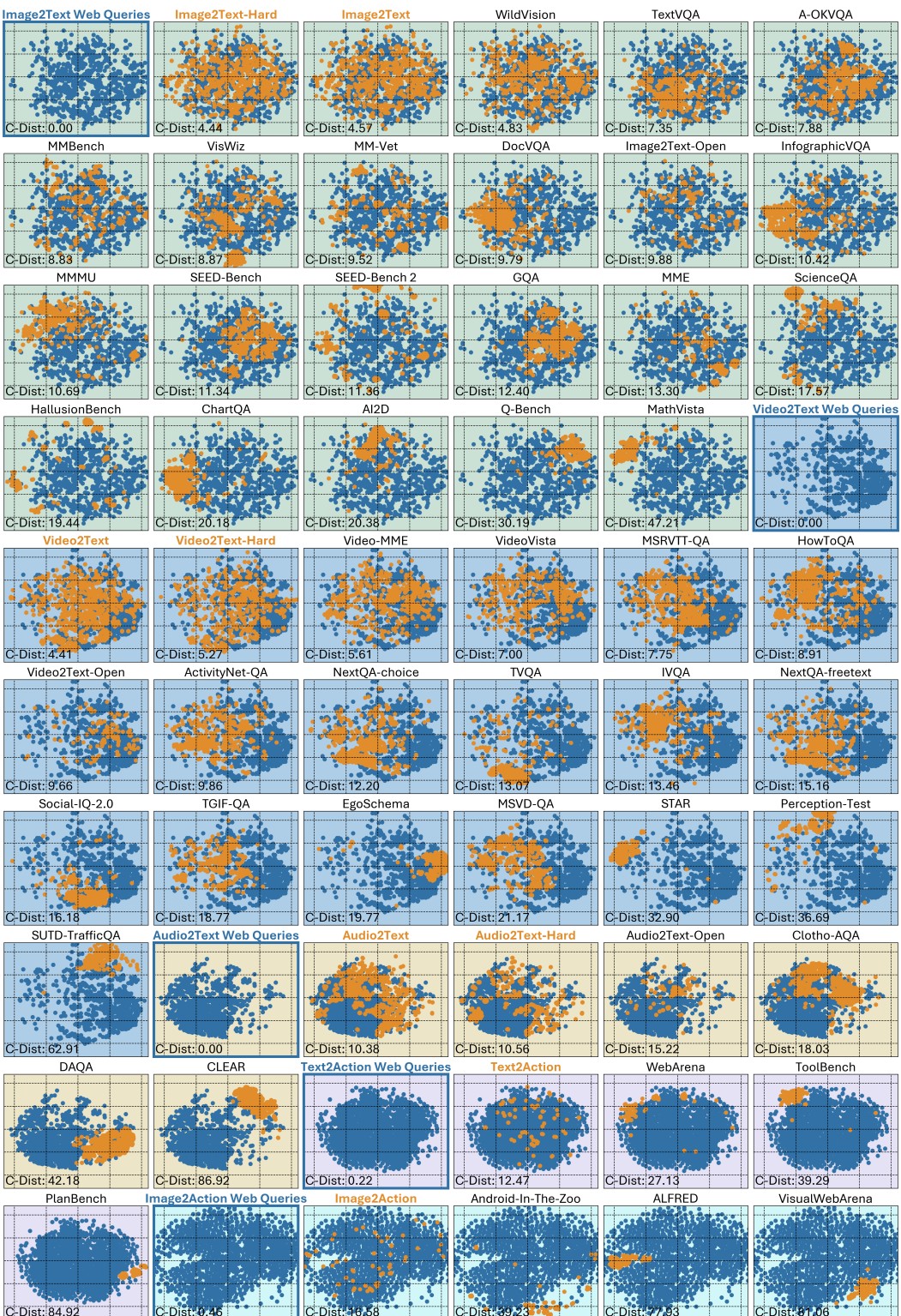

Figure 9: Task distribution of various modality benchmarks, with each modality uniquely color-coded. Benchmark data points (orange dots) are plotted against the detected web queries (blue dots) for the corresponding modality. The sentence embeddings of the queries were dimensionally reduced into a unified 2D space, enabling direct comparison of topic distributions across benchmarks. Benchmarks are sorted by their cluster distance (C-Dist) from the corresponding web queries.

distribution. Note that WildVision, the only real-world dataset available for these modalities, also aligns closely with web queries, Image2Text, and Image2Text-Hard, reinforcing this conclusion.

**The task distributions of most existing benchmarks deviate from real-world task distributions.** This deviation is expected, as most benchmarks are not designed with real-world tasks in mind. This creates a challenge since we expect evaluation results to generalize to real-world applications where models are deployed (Ni, 2024). Nonetheless, existing benchmarks, though deviated, can still be useful for evaluating particular aspects of a model. However, some benchmarks' actual distributions may not match their creators' claims, risking the use of inappropriate benchmarks for specific tasks. To better understand these benchmarks, users can refer to Figures 35-39, which use GPT-4 to summarize task topics from specific 2D grid locations in Figure 9, enabling more interpretable and accurate benchmark selection.

## 4.2 CORRELATION ANALYSIS

**MixEval-X demonstrates a strong correlation with real-world user-facing evaluations.** A key feature of `MixEval-X` is its alignment of benchmark task distributions with real-world tasks.

Beyond distribution analysis, we assess this alignment by evaluating the correlation between `MixEval-X` results and real-world evaluations. While many communities lack stable real-world evaluation leaderboards like Chatbot Arena (Chiang et al., 2024), the Image2Text community has two comparatively stable user-facing leaderboards: Vision Arena (Lu et al., 2024b) and Arena (Vision) (Chiang et al., 2024). Our Image2Text results show a strong Spearman's model ranking correlation with these, with **98.1%** correlation to Vision Arena and **96.3%** to Arena (Vision); Image2Text-Hard shows **94.5%** and **95%**, respectively. These high correlations, along with findings in Ni et al. (2024), highlight the effectiveness of our benchmark mixture approach. Although correlations for other modalities can't be verified at present due to the lack of stable real-world evaluations[1], we will assess them once suitable evaluations are available.

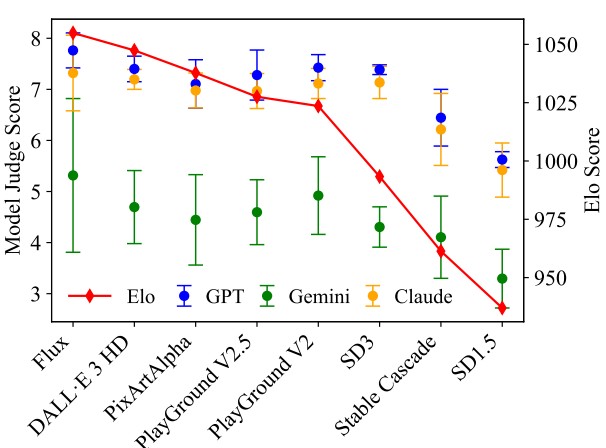

Figure 10: Model judge scores and crowd-sourcing Elo scores of the Text2Image subset. The upper and lower error bars represent the 1st and 2nd turn scores, respectively. Each data point is an average of five different runs.

**Multi-modal language models evaluate MMG tasks differently from crowd-sourced human preferences.** In this study, we employed 700-800 crowd-sourced workers for pairwise human preference evaluations of MMG tasks. Large-scale human evaluations provide meaningful assessments due to the Wisdom of the Crowd effect (Yi et al., 2012) and their relevance to real-world applications, but they are time-consuming and costly (Ni et al., 2024). Thus, we are exploring cheaper alternatives, like LLM-as-judge evaluations, which, though considered biased (Zheng et al., 2023; Ni et al., 2024), have shown promise in open-ended Text2Text tasks. We compared frontier model judges against crowd-sourced results (Figure 10). Nearly all model judges showed low correlations with human preferences (**78%** on average, see Table 2), suggesting that multi-modal models evaluate MMG tasks differently from humans, consistent with Zhang et al. (2024b). We focused on Image2Text results due to a lack of reliable judge models for video and audio tasks. Interestingly, correlations between model judges were relatively high (85%-95%), indicating a potential shared bias. These findings highlight the need for further research into cost-effective, low-bias grading methods for MMG tasks.

---

[1]GenAI-Arena, a real-world platform for MMG models, and the Video2Text leaderboard in Vision Arena remain unstable due to limited votes and models.

ACKNOWLEDGEMENT

We thank Graham Neubig, Yuntian Deng, Jason Phang, Piotr Nawrot, Luca Soldaini, Guanzhi Wang, Qinghong Lin, and Fanyi Pu for insightful discussions and pointers.

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

## A  FREQUENTLY ASKED QUESTIONS

### A.1  WHY ARE THERE ONLY EIGHT INPUT AND OUTPUT MODALITY COMBINATIONS COVERED?

The eight input and output modalities represent the most common cases identified in our web query analysis and are also widely regarded as central by the AI community. Expanding the scope to include more modalities could dilute the overall quality of the work. Hence, we have chosen to focus on the key modalities for now, leaving combinations like Image2Video for future exploration.

### A.2  WHY DO YOU MATCH TEXTUAL WEB QUERIES WITH TEXTUAL BENCHMARK QUERIES IN THE BENCHMARK MIXTURE OF MMU TASKS, INSTEAD OF MATCHING REAL-WORLD INPUT-QUERY PAIRS WITH BENCHMARK INPUT-QUERY PAIRS (WHERE INPUT DENOTES THE MULTI-MODAL INPUT, SUCH AS IMAGE OR VIDEO)?

A practical reason is the difficulty in obtaining large-scale, multi-modal tasks that reflect real-world distributions. In this paper, we detect such tasks from the web to capture real-world task distributions using MixEval's (Ni et al., 2024) detection pipeline, which is challenging but feasible. However, matching input-query pairs poses three significant challenges: (1) If the data source is the web, as in this work, multi-modal corpora with video or audio inputs are not readily available. (2) If the data source is real-world, we would need to create platforms, like ChatGPT or Chatbot Arena, that users actively engage with to collect input-query pairs with real-world distributions—this is time-intensive, costly, and difficult to control in terms of distribution. Moreover, without serving usable models supporting these modalities, such user inputs will not scale up. (3) Even if multi-modal input-query pairs were obtained, representing these tasks effectively remains a challenge due to the lack of robust representation models for such modalities.

Another reason is that, in most tasks, the textual query captures a substantial portion of the task information, making the benchmark mixture meaningful. Our goal is to optimize task distributions rather than achieve perfect representations.

### A.3  WHY NOT EVALUATE INTERLEAVED OPEN-ENDED MMU BENCHMARKS?

The primary reason is the lack of capable judges for video and audio tasks (and other modalities). Accurate evaluation requires a sufficiently robust judge model, such as GPT-4 for Text2Text tasks, which was employed in MT-Bench (Zheng et al., 2023). However, even with advanced models as judges, these models exhibit significant biases, including preference and length biases (Zheng et al., 2023; Ni et al., 2024).

### A.4 WHY NOT DEDUPLICATE THE MMU BENCHMARKS?

In this work, we match each web query with the most similar benchmark query and its corresponding ground truth. Given the finite size of the benchmark pool, it is likely that multiple web queries will be paired with the same benchmark entry. This duplication is expected, and deduplication would distort the real-world distribution.

## B RELATED WORK

**LLM Evaluations** The LLM community is the most advanced among different AI communities. In practical LLM evaluations, three primary biases compromise impartiality: (1) query bias—evaluation queries that lack comprehensiveness or proper distribution, (2) grading bias—significant bias or error in the grading process, and (3) generalization bias—model overfitting to the evaluation data. Current benchmarking approaches are either automatic or user-facing. Automatic benchmarks often use traditional, ground-truth-based frameworks like MMLU (Hendrycks et al., 2020), which fail to capture the complexity and nuance of real-world queries, though they offer a relatively unbiased grading process. Alternatively, open-ended benchmarks that employ LLMs as graders, such as MT-Bench (Zheng et al., 2023), face issues of grading bias and query incompleteness due to preference biases and the high cost of cutting-edge LLM judges. Furthermore, the static nature of automatic benchmarks introduces contamination over time, exacerbating generalization bias. These biases lead to significant deviations from gold-standard evaluations, hindering model development. In contrast, large-scale user-facing benchmarks like Chatbot Arena (Chiang et al., 2024) provide more reliable metrics for model development and address the three biases more effectively. (1) They capture a diverse array of real-world queries, ensuring better query comprehensiveness and distribution. (2) Their evaluation of varied model responses benefits from the "wisdom of the crowd" effect (Yi et al., 2012), where individual judgment noise is averaged across numerous samples, reducing grading bias. (3) Continuous influx of user queries minimizes benchmark contamination. Moreover, this approach steers model optimization towards practical applications, aligning models more closely with user needs. However, Chatbot Arena is costly, slow, and irreproducible. It is also not publicly accessible, limiting practitioners' ability to conduct quick and straightforward evaluations. Recently, MixEval (Ni et al., 2024) was introduced as an efficient, dynamic, and low-bias evaluation framework for LLMs, addressing the aforementioned biases.

**MMU Evaluations** Compared to the LLM (Text2Text) community, evaluations in the MMU domain remain underexplored. Among the modalities, the Image2Text community is relatively more mature, closely following the evaluation paradigms established in the LLM field. Existing evaluation approaches include ground-truth-based methods (Yue et al., 2024; Liu et al., 2023b), VLM-as-judge frameworks (Yu et al., 2023b), and user-facing evaluations (Jiang et al., 2024). However, user-facing evaluations in this domain tend to be unstable due to the limited availability of user votes. In contrast, the Video2Text and Audio2Text communities have only seen a limited number of ground-truth-based evaluations thus far (Yu et al., 2019; Mangalam et al., 2023; Xu et al., 2017; Lipping et al., 2022; Fayek & Johnson, 2020; Lin et al., 2021). `MixEval-X` performs benchmark mixture for a large MMU benchmark pool to achieve golden-standard MMU evaluations.

**MMG Evaluations** MMG tasks are inherently open-ended, making ground-truth-based evaluations ineffective. Traditional automatic metrics (Wang et al., 2004; Zhang et al., 2018; Heusel et al., 2017) fail to capture the subtleties of generation quality. Zhang et al. (2024b) has shown that current vision-language models are not reliable judges for MMG tasks. Currently, user-facing evaluations are regarded as the most reliable approach for MMG, either through crowdsourcing platforms or expert panels. GenAI-Arena is a crowdsourced, user-facing platform that evaluates MMG tasks via user votes, similar to Chatbot Arena. However, as with other modalities beyond Text2Text, the insufficient number of votes leads to instability in model rankings. `MixEval-X`'s MMG subsets offer several centralized sets of real-world generation and editing tasks to enable more unbiased and reproducible human evaluations, eliminating the need for arena-style platforms to collect decentralized user queries. Arena-like platforms require significantly more human resources and are inherently non-reproducible due to user randomness. In the future, `MixEval-X` MMG tasks may also utilize reliable model judges to enhance evaluation efficiency.

**(Multi-modal) Agent Evaluations** Current agent benchmarks (Liu et al., 2023a; Zhou et al., 2023; Koh et al., 2024) are typically tightly integrated with specific environments to obtain evaluation

signals (e.g., successful task completion), making them highly domain-specific. As a result, these benchmarks often fail to represent real-world task distributions, limiting their ability to generalize to certain real-world tasks. The `MixEval-X` agent subset introduces a general-purpose (multi-modal) agent benchmark that avoids relying on environment-specific signals to assess model actions. Instead, tasks are adapted from real-world user queries and framed as API-calling tasks. In this setup, the model is not required to complete the tasks directly but is evaluated based on its ability to invoke various provided API actions to achieve task goals. This approach is expected to generalize more effectively to real-world tasks due to its alignment with real-world task distributions.

## C   EVALUATION SETTINGS

In this section, we provide comprehensive evaluation results to offer more precise rankings for models and organizations in the field. We follow official settings for all open-source models to ensure fairness. For proprietary models, we use their official APIs. To avoid task-specific biases, we standardize the benchmark input formats, including prompts. Models supporting interleaving receive interleaved entries as input. Since current MMG models only accept caption-like prompts, we use a caption rewriter (GPT-4) to convert user instructions into caption-like inputs for MMG tasks.

## D   EXAMPLE DATA SAMPLES AND MODEL OUTPUTS

**Image2Text Free-form Tasks**

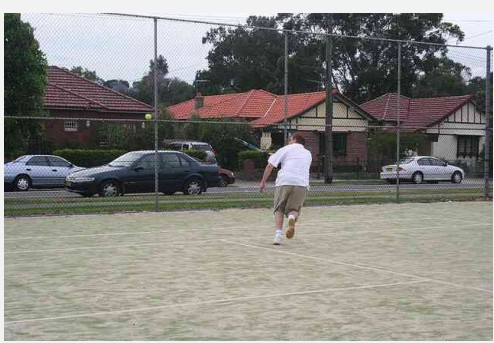

**Query:** What is the fence meant to block?
**Reference Answers:** ["balls", "tennis balls"]

**GPT-4o:** Tennis balls
**Claude 3.5 Sonnet:** Tennis balls
**Gemini-1.5-Pro:** Tennis balls

**Image2Text Multiple-choice Tasks**

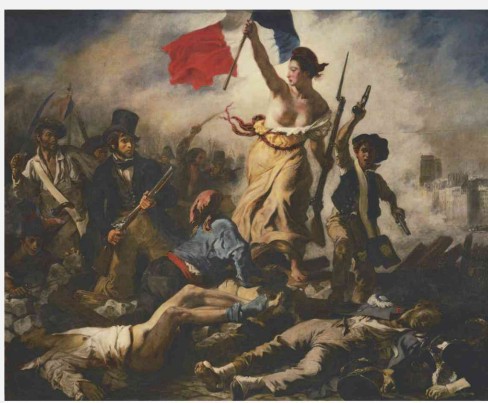

**Query:** Which work was most directly influenced by the painting?
**Options:**
A. Rude's La Marseillaise
B. Goya's Third of May
C. Picasso's Guernica
D. Johns's Three Flags
**Reference Answers:** A

**GPT-4o:** A. Rude's La Marseillaise
**Claude 3.5 Sonnet:** A. Rude's La Marseillaise
The painting shown is "Liberty Leading the People" by Eugŏ0e8ne Delacroix, which depicts a scene from the French Revolution of 1830. This work, with its allegorical figure of Liberty leading revolutionaries, most directly influenced Franŏ0e7ois Rude's sculpture "La Marseillaise" on the Arc de Triomphe in Paris. Both works share similar themes of French patriotism, revolutionary spirit, and the personification of liberty as a female figure.
**Gemini-1.5-Pro:** C.

### Image2Text Open-ended Tasks

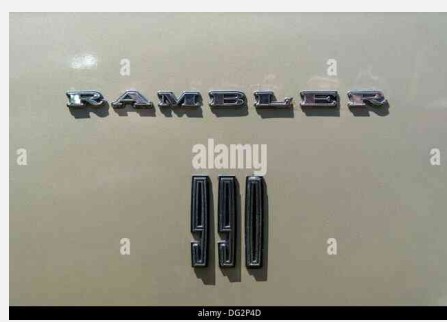
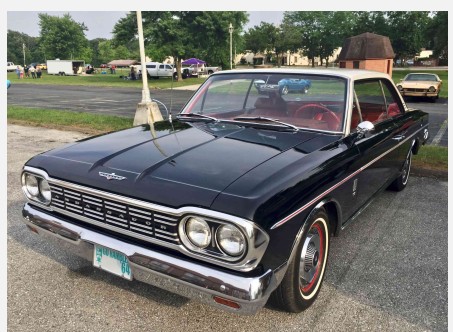

<image 1>                                <image 2>

**Turn 1:** <image 1>This black-and-white image shows the car logo from which manufacturer?
**Turn 2:** Is that the logo of the car shown in <image 2>?

Open-ended MMU datasets are not evaluated in this work. We only release the data instead. See Section A.3 for details.

### Video2Text Free-form Tasks

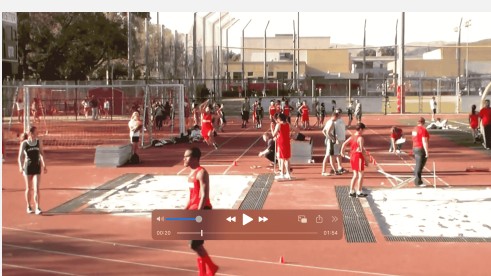

See the full video here.

**Query:** What happened to the referee after the first long jump?
**Reference Answers:** ["measuring distance"]

**GPT-4o:** Cheering
**GPT-4V:** The referee was accidentally hit by a long jumper during the event.
**Gemini-1.5-Flash:** He walked away.

## Video2Text Multiple-choice Tasks

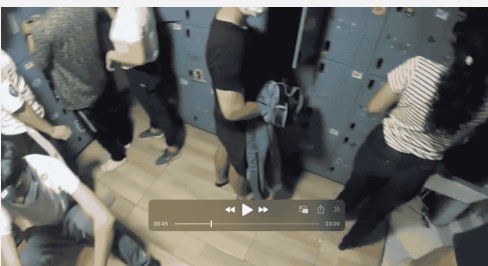

See the full video here.

**Query:** Identify the primary theme present throughout the video, and provide key actions or scenes that support your assessment of this theme.
**Options:**
A. The primary theme present throughout the video is anxiety. this is evident in the fact that c stares at her hand, touches her shoes, and looks around.
B. In the video, the primary theme consistently present throughout is nervousness. this is clearly evident due to the fact that character c repeatedly stares at her hand, fidgets with her shoes, and constantly looks around.
C. The primary theme present throughout the video is cleanliness. this is evident in the fact that c washes her hands multiple times, interacts with the man in a friendly way, and wipes her hands on a tissue paper.
D. The predominant primary theme present consistently throughout the entire video is friendliness. this aspect is clearly evident in the fact that character c interacts with the man in a genuinely friendly way.
E. The most noticeable primary theme present throughout the entire video is politeness. this aspect is clearly evident in the fact that character c courteously shakes the man's hand.
**Reference Answers:** C

**GPT-4o:** C
**GPT-4V:** E.
**Gemini-1.5-Flash:** C.

## Video2Text Open-ended Tasks

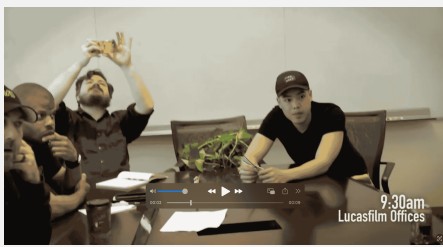

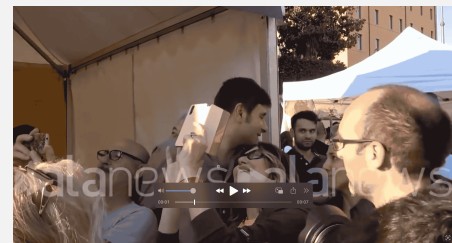

<video 1>
See the full video here.

<video 2>
See the full video here.

**Turn 1:** What is the scene of <video 1>? Describe the appearance of the boss.
**Turn 2:** Is the scene in <video 2>the same as that in <video 1>? Are they gathering by organization or just at random?

Open-ended MMU datasets are not evaluated in this work. We only release the data instead. See Section A.3 for details.

**Audio2Text Free-form Tasks**

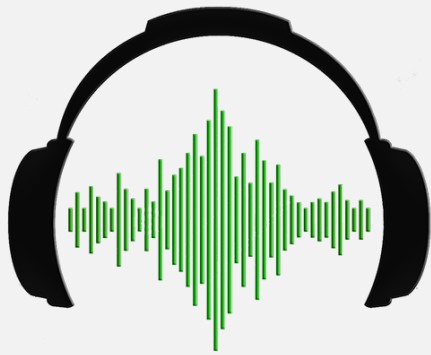

Listen to the full audio here.

**Query:** What did you hear before the phone ringing?
**Reference Answers:** ["human whistling"]

**Qwen2-Audio-7B:** dog barking
**SALMONN-13B:** A dog barking.
**Pengi:** motorcycle

**Audio2Text Open-ended Tasks**

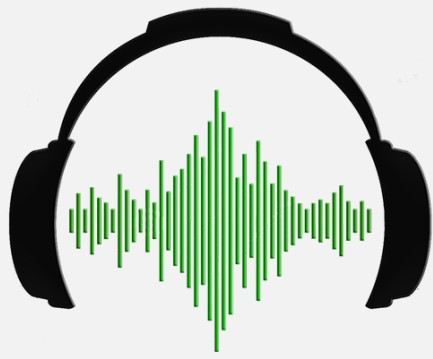 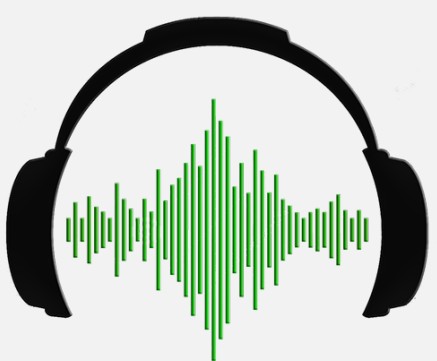

<audio 1>             <audio 2>
Listen to the full audio here.      Listen to the full audio here.

**Turn 1:** <audio 1>Is it possible to transcribe the words in <audio 2>into subtitles? Why?
**Turn 2:** What is the man in <audio 1>talking about? Try to recover the whole content he is talking (including those not covered by the audio).

Open-ended MMU datasets are not evaluated in this work. We only release the data instead. See Section A.3 for details.

**Text2Image Tasks**

**1st Turn Query (generation):** Design a vibrant cityscape banner featuring the iconic Sydney Opera House and Harbour Bridge under a dazzling sunset, with the City of Sydney̌2019s logo prominently displayed in the foreground, ensuring it catches the eye against the vivid backdrop.
**2nd Turn Query (edit):** Remove the City of Sydney̌2019s logo.

**Flux**

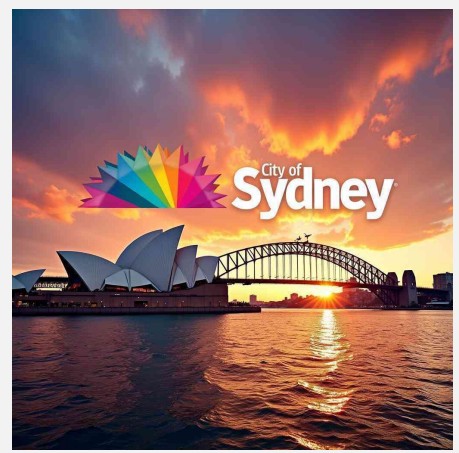 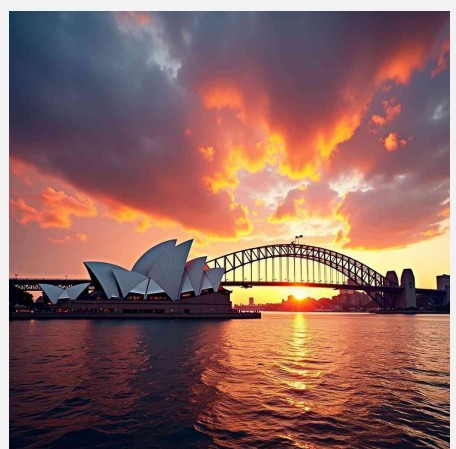

1st turn generation.            2nd turn generation.

**DALL·E 3 HD**

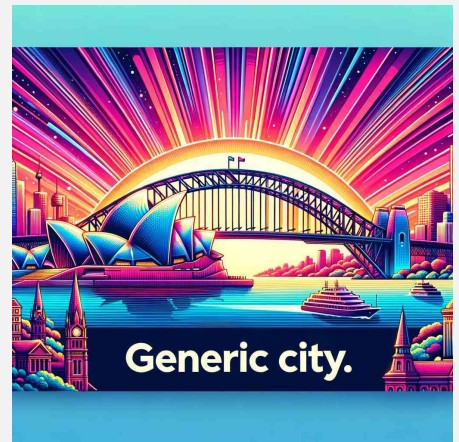 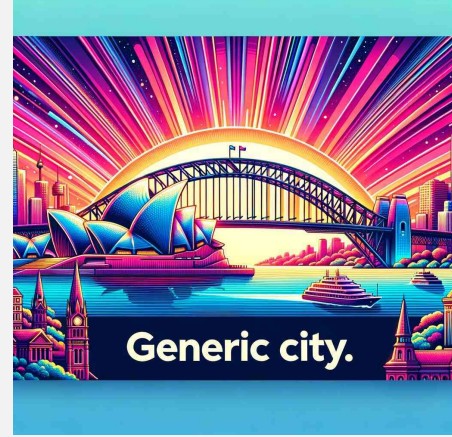

1st turn generation.            2nd turn generation.

**PlayGround V2.5**

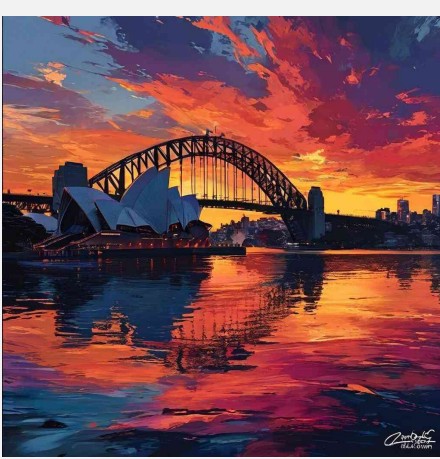 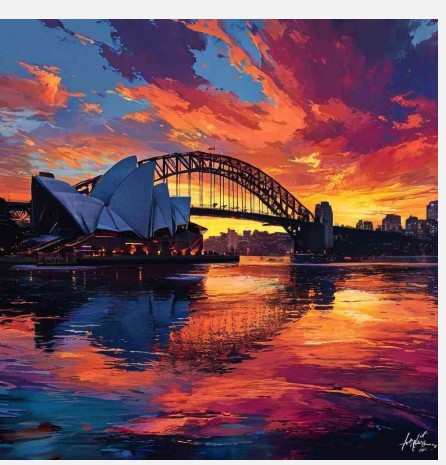

1st turn generation.            2nd turn generation.

**Text2Video Tasks**

**1st Turn Query (generation):** Create a dynamic video showcasing the energy and excitement of a live event. Focus on vibrant crowd reactions, close-ups of performers or speakers engaging passionately, and visually stunning moments that capture the essence of being there live. Ensure to include diverse camera angles to give a full experience of the event's atmosphere.
**2nd Turn Query (edit):** Highlight the vibrant crowd reactions.

**CogVideoX-5B**

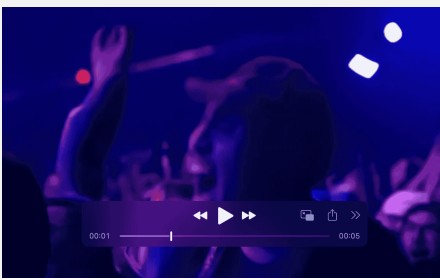

1st turn generation.
See the full video here.

2nd turn generation.
See the full video here.

**HotShot-XL**

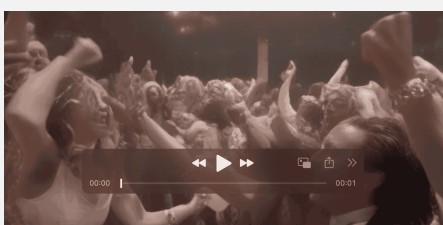
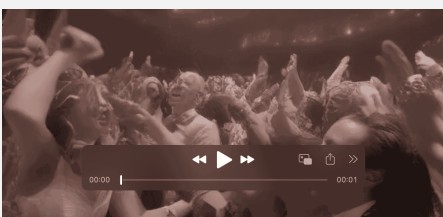

1st turn generation.
See the full video here.

2nd turn generation.
See the full video here.

**VideoCrafter2**

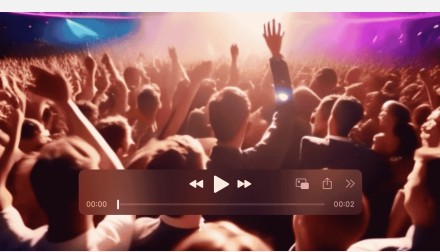
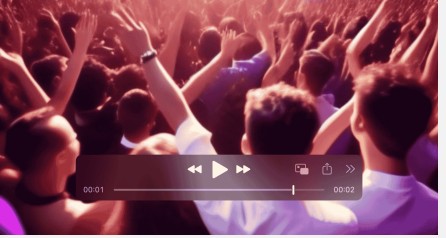

1st turn generation.
See the full video here.

2nd turn generation.
See the full video here.

**Text2Audio Tasks**

**1st Turn Query (generation):** Craft a brief audio snippet featuring a clear, firm voice stating the necessity of paying fees exclusively within Russia, highlighting the rule's strictness and the policy's geographical exclusivity, without mentioning specific numbers or amounts.
**2nd Turn Query (edit):** Remove the geographical exclusivity detail.

**AudioLDM 2**

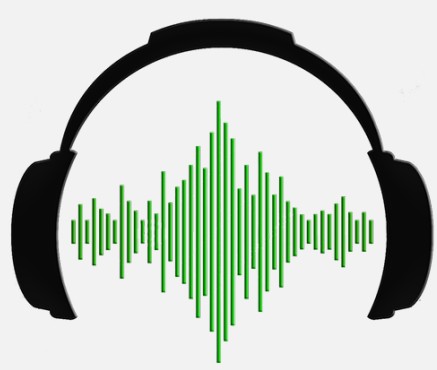

1st turn generation.
Listen to the full audio here.

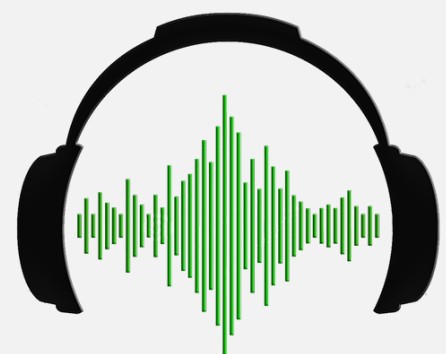

2nd turn generation.
Listen to the full audio here.

**Make-An-Audio 2**

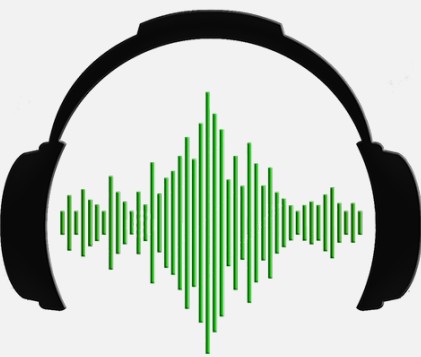

1st turn generation.
Listen to the full audio here.

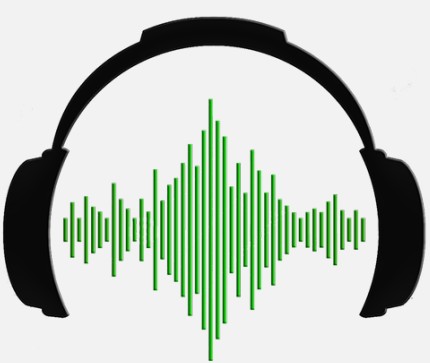

2nd turn generation.
Listen to the full audio here.

**Stable Audio**

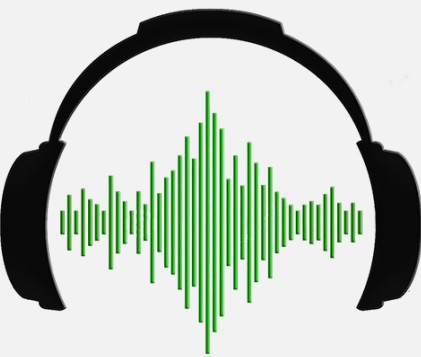

1st turn generation.
Listen to the full audio here.

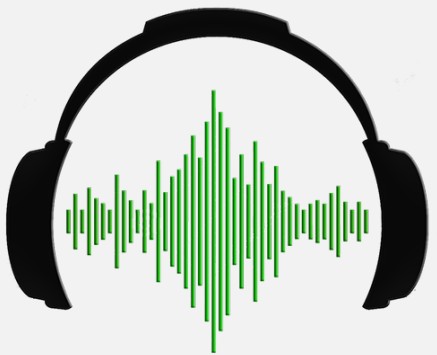

2nd turn generation.
Listen to the full audio here.

## Text2Action Tasks

**Task Description:** Find and send a copy of the Donor Privacy Policy to a requesting donor.
**Allowed Actions:** [Navigation], [PickupObject], [PutObject], [OpenObject], [CloseObject], [ReadText], [PrintObject], [ScanObject], [SendEmail], [TypeText], [AttachFile], [OpenEmailClient],

[SearchForObject], [IdentifyObject], [UseComputer]
**Visible Objects:** <donor privacy policy document>, <computer>, <printer>, <scanner>, <email client>, <file cabinet>, <desk>, <envelope>, <stamp>, <mail drop box>, <recipient address label>, <office supplies>, <trash can>, <outbox tray>
**Already Executed Steps:** [Navigation] <office>, [OpenObject] <file cabinet>, [SearchForObject] <donor privacy policy document>, [PickupObject] <donor privacy policy document>, [CloseObject] <file cabinet>
**Reference Answer:** [Navigation] <scanner>, [ScanObject] <donor privacy policy document>, [Navigation] <computer>, [OpenEmailClient] <email client>, [TypeText] <recipient address>, [AttachFile] <donor privacy policy document>, [SendEmail] <email client>

**GPT-4o:** Based on the given task and the steps that have already been executed, here is the plan for the remaining action-object sequences to complete the task: 1. [Navigation] <scanner>2. [PutObject] <donor privacy policy document><scanner>3. [ScanObject] <donor privacy policy document>4. [Navigation] <computer>5. [UseComputer] <computer>6. [OpenEmailClient] <email client>7. [TypeText] "Please find attached a copy of the Donor Privacy Policy you requested."<email client>8. [AttachFile] <donor privacy policy document scan><email client>9. [SendEmail] <email client>

**Claude 3.5 Sonnet:** Based on the task description and the already executed action-object sequences, here's the plan to complete the task of sending a copy of the Donor Privacy Policy to a requesting donor: [Navigation] <computer>[UseComputer] <computer>[OpenEmailClient] <email client>[ScanObject] <donor privacy policy document>[AttachFile] <donor privacy policy document>[TypeText] <email message>[SendEmail] <donor email>This sequence of actions will accomplish the task efficiently: 1. Navigate to the computer to use it for sending the email. 2. Use the computer to access necessary software. 3. Open the email client to prepare the email. 4. Scan the donor privacy policy document to create a digital copy. 5. Attach the scanned file to the email. 6. Type an appropriate message in the email body. 7. Send the email with the attached donor privacy policy to the requesting donor. This plan assumes that the donor's email address is known or can be easily retrieved. If that information is not readily available, additional steps might be needed to locate the donor's contact information before sending the email.

**Gemini-1.5-Pro:** Your Planning: 1. **[Navigation] <scanner>** 2. **[OpenObject] <scanner>** 3. **[PutObject] <donor privacy policy document><scanner>** 4. **[ScanObject] <donor privacy policy document>** 5. **[CloseObject] <scanner>** 6. **[Navigation] <computer>** 7. **[OpenEmailClient] <email client>** 8. **[TypeText] <email client>** (Compose email to donor) 9. **[AttachFile] <scanned document><email>** 10. **[SendEmail] <email>**

## Image2Action Tasks

**Task Description:** Find and purchase a wrench.
**Allowed Actions:** [RecognizeObject], [SelectObject], [DeselectObject], [ConfirmSelection], [Navigate], [ReadText], [ProcessList], [IdentifyObject], [Search], [PickUp], [Transaction]
**Visible Objects:**

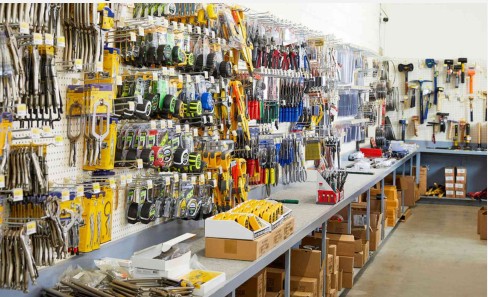

**Already Executed Steps:** [Search] <wrench>, [IdentifyObject] <wrench>
**Reference Answer:** [Navigate] <wrench>, [PickUp] <wrench>, [Navigate] <checkout counter>, [PickUp] <cash>, [Transaction] <cash>

**GPT-4o:** [PickUp] <wrench>, [Navigate] <checkout counter>, [Transaction] <wrench>

**Claude 3.5 Sonnet:** Based on the task description, allowed actions, visible objects, and already executed action-object sequences, here's the plan for the remaining action-object sequences to find

and purchase a wrench: [Navigate] <wrench section>[RecognizeObject] <wrench>[SelectObject] <wrench>[ReadText] <price tag>[ConfirmSelection] <wrench>[PickUp] <wrench>[Navigate] <checkout counter>[Transaction] <wrench>Explanation of the plan: 1. Navigate to the wrench section, which we can see in the image. 2. Recognize the specific wrench we want to purchase among the various tools. 3. Select the desired wrench. 4. Read the price tag to ensure it's within budget. 5. Confirm the selection of the wrench. 6. Pick up the wrench. 7. Navigate to the checkout counter (assumed to be present in the store). 8. Complete the transaction to purchase the wrench. This sequence of actions should successfully complete the task of finding and purchasing a wrench in the hardware store environment shown in the image.

**Gemini-1.5-Pro:** Your Planning: [Navigate] <wrench>, [PickUp] <wrench>, [Transaction] <wrench>

## E    ADAPTATION-RECTIFICATION PROMPTS

### Text2Image Adaptation-Rectification Prompt - Adaptation 1

**System:** In this task, I want you to act as an instruction rewritter, and imagine that you are rewritting instructions for image generation.

**Main:** The rewritten instruction will be used to instruct an image generation model that generates images. You will be provided with a noisy raw user instruction that decides the topic or content of the image generation task and you need to rewrite the raw instruction to make it clearer and more specific. Meanwhile, it should be practical for the image generation model to generate the corresponding image. The below examples are simplified, while your rewritten instructions could be either more detailed or more concise.

Example 1:
Raw User Instruction: Eligible artwork. The artwork MUST be of wildlife. The rewritten instruction: Create an image of a serene forest at sunrise, with deer by a stream, birds on branches, and rabbits in the underbrush, all bathed in the soft, golden light of the morning.

Example 2:
Raw User Instruction: Next up, how to draw a Lamborghini – the inner parts and details. The rewritten instruction: How about exhibiting a Lamborghini with open hood and doors, detailing the engine, dashboard, and leather seats in a brightly lit setting, emphasizing the car's luxury and mechanical complexity?

Given a noisy raw user instruction that implies the topic or content of the image generation task, brainstorm and rewrite the instruction to make it more clear and specific. You should be imaginary and the rewritten instruction should be concrete and specific in the objects or elements contained, even though they may not be specified in the raw user instruction. The rewritten instruction is not required to always use a fixed format starting from "Draw an image..." or "Create an image...". Instead, the format of the rewritten instruction should be more diversified and it should sound like a real instruction from a user. However, it's important to always include the request for image creation in the rewritten instruction. Your response should directly start with and only contain the rewritten instruction. Do not generate anything else.

Raw User Instruction:
The rewritten instruction:

### Text2Image Adaptation-Rectification Prompt - Adaptation 2

**System:** In this task, I want you to act as an image edit task designer, and imagine that you are instructing an image edit model to edit images.

**Main:** You will be provided with the image caption of an image to edit and you need to randomly pick some editing aspects to formulate an editing instruction. Besides that, you should provide the image

caption for the edited image. Below are some examples.

Example 1:
Image Caption: A serene forest at sunrise, with deer by a stream, birds on branches, and rabbits in the underbrush, all bathed in the soft, golden light of the morning.
The editing json dict: "editing instruction":"Change the time to night.", "caption of the edited image":"A serene forest at night, with deer by a stream, birds on branches, and rabbits in the underbrush, all bathed in the soft, golden light of the morning.".

Example 2:
Image Caption: A Lamborghini with open hood and doors, detailing the engine, dashboard, and leather seats in a brightly lit setting, emphasizing the car's luxury and mechanical complexity.
The editing json dict: "editing instruction":"Lamborghini → Ferrari.", "caption of the edited image":"A Ferrari with open hood and doors, detailing the engine, dashboard, and leather seats in a brightly lit setting, emphasizing the car's luxury and mechanical complexity.".

You should randomly pick a thing/feature in the original image caption to edit. Note that the editing instruction must be practical, i.e., the things/features to edit must be contained in the provided image caption. The caption of the edited image should be exactly aligned with that of the original image, i.e., the only differences between the two captions should be those things/features that need to be edited. Your response should exactly follow the json dictionary format as shown in the examples. Do not generate anything else.

Image Caption:
The editing json dict:

---

## Text2Image Adaptation-Rectification Prompt - Rectification

**System:** In this task, I want you to act as an inspector of the image generation task, and imagine that you are inspecting the quality of the designed image generation tasks.

**Main:** You will be provided with a task dictionary which is designed to instruct image generation models to generate and edit images, and you need to judge whether each component is valid. The task contains 2 turns, each with a user prompt and its corresponding image caption, and there is no answer or generated images in the task dictionary. The first-turn user prompt is mainly designed to instruct the image generation model to generate images, and the first-turn caption is the corresponding caption format of the first-turn user prompt, which contains the same features/elements as that of the first-turn user prompt. The second-turn user prompt is a short prompt designed to instruct the image generation model to edit images generated in the first turn, and the second-turn caption is the caption designed for the edited image, which is almost the same as the first-turn caption except for the items/features being edited. In the inspection json dictionary, answer 'Yes' if a component is correct; answer 'No' and specify the reason if it's not. Below examples only illustrate the format and some simple situations, while it does not cover all conditions.

Example 1:
Task json dictionary: "turn1-prompt": "Illustrate a close-up of a car's front, focusing on detailing the hood's contours and shapes, alongside the intricate designs of the headlights, under a clear, bright light to enhance the features.", "turn1-caption": "A close-up of a car's front, focusing on detailing the hood's contours, alongside the intricate designs of the headlights, under a clear, bright light to enhance the features.", "turn2-prompt": "Change the time to dusk.", "turn2-caption": "A close-up of a car's front, focusing on detailing the hood's contours and shapes, alongside the intricate designs of the headlights, under a clear, bright dusk light to enhance the features." Your inspection json dictionary: "turn1-prompt correct": "Yes", "turn1-caption correct": "No. The hood's shapes are missed from the caption.", "turn2-prompt correct": "No. The time is not contained in the turn-1 caption.", "turn2-caption correct": "Yes"

Example 2:
Task json dictionary: "turn1-prompt": "Generate an interactive digital photo album interface, with the 'Photos' tab highlighted. Upon clicking, display an organized index page showing thumbnails of various albums categorized by events and dates.", "turn1-caption": "An interactive digital photo album interface, with the 'Photos' tab highlighted, displaying an organized index page showing thumbnails

of various albums categorized by events and dates upon clicking.", "turn2-prompt": "Change the highlighted tab from 'Photos' to 'Videos'.", "turn2-caption": "An interactive digital photo album interface, with the 'Videos' tab highlighted, displaying fancy index pages which are categorized by events and dates." Your inspection json dictionary: "turn1-prompt correct": "Yes", "turn1-caption correct": "Yes", "turn2-prompt correct": "Yes", "turn2-caption correct": "No. The 'displaying fancy index pages which are categorized by events and dates.' in turn2-caption is not aligned with turn1-caption"

Given a task dictionary, check whether each component is correct in logic and format. Especially check the below four aspects: 1. Whether the turn1-prompt is a valid image generation request; 2. Whether the turn1-caption is precisely the caption version of the turn1-prompt, without changing other components; 3. Whether the turn2-prompt is a valid image editing request, with all its editing components exactly being contained in the turn1-caption; 4. Whether the turn2-caption is a valid caption for the edited image, which is supposed to be exactly aligned with turn1-caption, i.e., the only differences between the two captions should be those things/features that need to be edited. Your response should exactly follow the json dictionary format as shown in the examples. Your response should only contain the json dictionary and do not generate anything else.

Task json dictionary:
Your inspection json dictionary:

---

## Text2Image Task Rewritter Prompt

**System:** In this task, I want you to act as a caption extractor, and imagine that you are extracting captions from user instructions for image generation.

**Main:** You will be provided with a user instruction that asks an image generation model to generate images. Below are some examples.

Example 1:
User Instruction: Create an image of a serene forest at sunrise, with deer by a stream, birds on branches, and rabbits in the underbrush, all bathed in the soft, golden light of the morning. The Extracted Caption: A serene forest at sunrise, with deer by a stream, birds on branches, and rabbits in the underbrush, all bathed in the soft, golden light of the morning.

Example 2:
User Instruction: How about exhibiting a Lamborghini with open hood and doors, detailing the engine, dashboard, and leather seats in a brightly lit setting, emphasizing the car's luxury and mechanical complexity? The Extracted Caption: A Lamborghini with open hood and doors, detailing the engine, dashboard, and leather seats in a brightly lit setting, emphasizing the car's luxury and mechanical complexity.

The extracted caption must contain exactly the same content as that of the user instruction, i.e., do not add or reduce any image content/feature description that is included in the original user instruction. You only need to change the format from the instruction format to the caption format. Your response should directly start with and only contain the extracted caption. Do not generate anything else.

User Instruction:
The Extracted Caption:

---

## Text2Video Adaptation-Rectification Prompt - Adaptation 1

**System:** In this task, I want you to act as an instruction rewritter, and imagine that you are rewritting instructions for short video generation (vision-only, without audio).

**Main:** The rewritten instruction will be used to instruct an video generation model that generates videos. You will be provided with a noisy raw user instruction that decides the topic or content of the video generation task and you need to rewrite the raw instruction to make it clearer and more specific. Meanwhile, it should be practical for the video generation model to generate the corresponding video. The below examples are simplified, while your rewritten instructions could be either more detailed or

more concise.

Example 1:
Raw User Instruction: How to Use Your Smartphone to Project Holograms? The rewritten instruction: Create a concise video illustrating the construction of a hologram projector from a smartphone. Highlight key steps: selecting materials (CD case, tape, pen, scissors), crafting a trapezoid from the CD case, forming a pyramid structure, and positioning it atop the smartphone to project a hologram. Focus on clear, visual steps, ending with the hologram projection.

Example 2:
Raw User Instruction: Produce an action film? The rewritten instruction: Give me a thrilling action sequence showcasing a chase scene in an urban setting. Highlight intense character expressions, swift movements through crowds, jumps over obstacles, and a clever escape. Use narrow alleyways, bustling streets, and iconic landmarks to enrich the visual narrative.

Given a noisy raw user instruction that implies the topic or content of the video generation task, brainstorm and rewrite the instruction to make it more clear and specific. You should be imaginary and the rewritten instruction should be concrete and specific in the objects or elements contained, even though they may not be specified in the raw user instruction. The rewritten instruction is not required to always use a fixed format starting from "Create an video..." or "Generate an video...". Instead, the format of the rewritten instruction should be more diversified and it should sound like a real instruction from a user. However, it's important to always include the request for video creation in the rewritten instruction. The rewritten instruction should not request for generating videos with too long content, e.g., hour-level videos. Instead, they should be second-level, while you should not directly mention the target length of the video in the rewritten instruction. The videos to generate are vision-only and do not include audios. Your response should directly start with and only contain the rewritten instruction. Do not generate anything else.

Raw User Instruction:
The rewritten instruction:

---

## Text2Video Adaptation-Rectification Prompt - Adaptation 2

**System:** In this task, I want you to act as an video edit task designer, and imagine that you are instructing an video edit model to edit videos (vision-only, without audio).

**Main:** You will be provided with the video caption of an video to edit and you need to randomly pick some editing aspects to formulate an editing instruction. Besides that, you should provide the video caption for the edited video. Below are some examples.

Example 1:
Video Caption: A concise video illustrating the construction of a hologram projector from a smartphone. Highlight key steps: selecting materials (CD case, tape, pen, scissors), crafting a trapezoid from the CD case, forming a pyramid structure, and positioning it atop the smartphone to project a hologram. Focus on clear, visual steps, ending with the hologram projection. The editing json dict: "editing instruction":"Remove the material selection", "caption of the edited video":"A concise video illustrating the construction of a hologram projector from a smartphone. Highlight key steps: crafting a trapezoid from the CD case, forming a pyramid structure, and positioning it atop the smartphone to project a hologram. Focus on clear, visual steps, ending with the hologram projection.".

Example 2:
Video Caption:A thrilling action sequence showcasing a chase scene in an urban setting. Highlight intense character expressions, swift movements through crowds, jumps over obstacles, and a clever escape. Use narrow alleyways, bustling streets, and iconic landmarks to enrich the visual narrative. The editing json dict: "editing instruction":"Emphasize the jumps over obstacles", "caption of the edited video":"A thrilling action sequence showcasing a chase scene in an urban setting. Highlight intense character expressions, swift movements through crowds, and a clever escape. Use narrow alleyways, bustling streets, and iconic landmarks to enrich the visual narrative. Emphasize the jumps over obstacles.".

You should randomly pick a thing/feature in the original video caption to edit. Note that the editing instruction must be practical, i.e., the things/features to edit must be contained in the provided video caption. The caption of the edited video should be exactly aligned with that of the original video, i.e., the only differences between the two captions should be those things/features that need to be edited. The videos to edit are vision-only and do not include audios. Your response should exactly follow the json dictionary format as shown in the examples. Do not generate anything else.

Video Caption:
The editing json dict:

---

## Text2Video Adaptation-Rectification Prompt - Rectification

**System:** In this task, I want you to act as an inspector of the short video generation task (vision-only, without audio), and imagine that you are inspecting the quality of the designed video generation tasks.
**Main:** You will be provided with a task dictionary which is designed to instruct video generation models to generate and edit videos, and you need to judge whether each component is valid. The task contains 2 turns, each with a user prompt and its corresponding video caption, and there is no answer or generated videos in the task dictionary. The first-turn user prompt is mainly designed to instruct the video generation model to generate videos, and the first-turn caption is the corresponding caption format of the first-turn user prompt, which contains the same features/elements as that of the first-turn user prompt. The second-turn user prompt is a short prompt designed to instruct the video generation model to edit videos generated in the first turn, and the second-turn caption is the caption designed for the edited video, which is almost the same as the first-turn caption except for the items/features being edited. In the inspection json dictionary, answer 'Yes' if a component is correct; answer 'No' and specify the reason if it's not. Below examples only illustrate the format and some simple situations, while it does not cover all conditions.

Example 1:
Task json dictionary: "turn1-prompt": "Generate a compilation video of the most captivating moments from the linked content. Focus on visually striking scenes, dynamic actions, and key highlights that can stand alone without audio explanation. Aim for a seamless transition between segments to maintain viewer engagement.", "turn1-caption": "A compilation video of the most captivating moments from the linked content. Focus on visually striking scenes and key highlights that can stand alone without audio explanation. Aim for a seamless transition between segments to maintain viewer engagement.", "turn2-prompt": "Highlight the facial expression more prominently", "turn2-caption": "A compilation video of the most captivating moments from the linked content. Focus on facial expression more prominently, dynamic actions, and key highlights that can stand alone without audio explanation. Aim for a seamless transition between segments to maintain viewer engagement." Your inspection json dictionary: "turn1-prompt correct": "Yes", "turn1-caption correct": "No, the 'dynamic actions' is missing compared to turn1-prompt.", "turn2-prompt correct": "No. The facial expression does not exist in turn1-caption, thus not being eligible for edit.", "turn2-caption correct": "No. The facial expression does not exist in turn1-caption, thus not being eligible for edit."

Example 2:
Task json dictionary: "turn1-prompt": "Craft a 15-min video demonstrating the enhancement of a website using Drupal 8. Focus on visual guides for integrating categories, enabling and managing comments, applying custom styles, and adding unique features to the site. Display step-by-step actions taken within the Drupal 8 interface to achieve each enhancement, ensuring clarity and precision in each visual step demonstrated.", "turn1-caption": "A 15-min video demonstrating the enhancement of a website using Drupal 8. Focus on visual guides for integrating categories, enabling and managing comments, applying custom styles, and adding unique features to the site. Display step-by-step actions taken within the Drupal 8 interface to achieve each enhancement, ensuring clarity and precision in each visual step demonstrated.", "turn2-prompt": "Increase clarity on applying custom styles", "turn2-caption": "A 15-min video demonstrating the enhancement of a website using Drupal 8. Focus on visual guides for integrating categories, enabling and managing comments, applying custom styles with increased clarity, and adding unique features to the site. Display step-by-step actions taken within the Drupal 8 interface to achieve each enhancement, ensuring clarity and precision in each visual step demonstrated." Your inspection json dictionary: "turn1-prompt correct": "No. The expected video is too long. It should be less than 120 seconds instead.", "turn1-caption correct": "Yes", "turn2-prompt correct": "Yes", "turn2-caption correct": "Yes"

Given a task dictionary, check whether each component is correct in logic and format. Especially check the below four aspects: 1. Whether the turn1-prompt is a valid video generation request that instructs a model to generate second-level videos less than 120 seconds (the explicit length restriction is not required to be included in the turn1-prompt; only inspect the estimated length of the content to generate); 2. Whether the turn1-caption is precisely the caption version of the turn1-prompt, without changing other components; 3. Whether the turn2-prompt is a valid video editing request, with all its editing components exactly being contained in the turn1-caption; 4. Whether the turn2-caption is a valid caption for the edited video, which is supposed to be exactly aligned with turn1-caption, i.e., the only differences between the two captions should be those things/features that need to be edited. Besides, the videos to generate are vision-only and do not include audios. Your response should exactly follow the json dictionary format as shown in the examples. Your response should only contain the json dictionary and do not generate anything else.

Task json dictionary:
Your inspection json dictionary:

## Text2Video Task Rewritter Prompt

**System:** In this task, I want you to act as a caption extractor, and imagine that you are extracting captions from user instructions for video generation.

**Main:** You will be provided with a user instruction that asks an video generation model to generate videos. Below are some examples.

Example 1:
User Instruction: Create a concise video illustrating the construction of a hologram projector from a smartphone. Highlight key steps: selecting materials (CD case, tape, pen, scissors), crafting a trapezoid from the CD case, forming a pyramid structure, and positioning it atop the smartphone to project a hologram. Focus on clear, visual steps, ending with the hologram projection. The Extracted Caption: A concise video illustrating the construction of a hologram projector from a smartphone. Highlight key steps: selecting materials (CD case, tape, pen, scissors), crafting a trapezoid from the CD case, forming a pyramid structure, and positioning it atop the smartphone to project a hologram. Focus on clear, visual steps, ending with the hologram projection.

Example 2:
User Instruction: Give me a thrilling action sequence showcasing a chase scene in an urban setting. Highlight intense character expressions, swift movements through crowds, jumps over obstacles, and a clever escape. Use narrow alleyways, bustling streets, and iconic landmarks to enrich the visual narrative. The Extracted Caption: A thrilling action sequence showcasing a chase scene in an urban setting. Highlight intense character expressions, swift movements through crowds, jumps over obstacles, and a clever escape. Use narrow alleyways, bustling streets, and iconic landmarks to enrich the visual narrative.

The extracted caption must contain exactly the same content as that of the user instruction, i.e., do not add or reduce any video content/feature description that is included in the original user instruction. You only need to change the format from the instruction format to the caption format. Your response should directly start with and only contain the extracted caption. Do not generate anything else.

User Instruction:
The Extracted Caption:

## Text2Audio Adaptation-Rectification Prompt - Adaptation 1

**System:** In this task, I want you to act as an instruction rewritter, and imagine that you are rewritting instructions for short audio generation.

**Main:** The rewritten instruction will be used to instruct an audio generation model that generates audios. You will be provided with a noisy raw user instruction that decides the topic or content of the audio generation task and you need to rewrite the raw instruction to make it clearer and more specific. Meanwhile, it should be practical for the audio generation model to generate the corresponding audio.

The below examples are simplified, while your rewritten instructions could be either more detailed or more concise.

Example 1:
Raw User Instruction: just read how you will dress. The rewritten instruction: Create an audio clip of a calm and composed voice describing an outfit choice for a formal event in detail, including the color and material of the clothing, any accessories being worn, and the reasons behind these choices.

Example 2:
Raw User Instruction: Search for similar music: funky, upbeat, ambient, excitement, business, pop, atmospheres, Bass-Guitar. The rewritten instruction: Give me an audio track that blends elements of funk and pop with an upbeat and exciting rhythm, featuring prominent bass guitar lines. The composition should include ambient textures to create a dynamic atmosphere suitable for a business environment. The track should evoke a sense of motivation and energy, making use of synthesizers and drum beats to enhance its lively mood.

Given a noisy raw user instruction that implies the topic or content of the audio generation task, brainstorm and rewrite the instruction to make it more clear and specific. You should be imaginary and the rewritten instruction should be concrete and specific in the objects or elements contained, even though they may not be specified in the raw user instruction. The rewritten instruction is not required to always use a fixed format starting from "Create an audio..." or "Generate an audio...". Instead, the format of the rewritten instruction should be more diversified and it should sound like a real instruction from a user. However, it's important to always include the request for audio creation in the rewritten instruction. The rewritten instruction should not request for generating audios with too long content, e.g., hour-level audios. Instead, they should be second-level, while you should not directly mention the target length of the audio in the rewritten instruction. The rewritten audio generation instruction can be either human speech generation or general audio generation. When the topic is possible, the rewritten instruction should b a general audio generation instruction without human speech or human voice generation. Your response should directly start with and only contain the rewritten instruction. Do not generate anything else.

Raw User Instruction:
The rewritten instruction:

---

**Text2Audio Adaptation-Rectification Prompt - Adaptation 2**

**System:** In this task, I want you to act as an audio edit task designer, and imagine that you are instructing an audio edit model to edit audios.

**Main:** You will be provided with the audio caption of an audio to edit and you need to randomly pick some editing aspects to formulate an editing instruction. Besides that, you should provide the audio caption for the edited audio. Below are some examples.

Example 1:
Audio Caption: A calm and composed voice describing an outfit choice for a formal event in detail, including the color and material of the clothing, any accessories being worn, and the reasons behind these choices. The editing json dict: "editing instruction":"Remove the description of the color and material of the clothing", "caption of the edited audio":"A calm and composed voice describing an outfit choice for a formal event in detail, including any accessories being worn, and the reasons behind these choices.".

Example 2:
Audio Caption: An audio track that blends elements of funk and pop with an upbeat and exciting rhythm, featuring prominent bass guitar lines. The composition should include ambient textures to create a dynamic atmosphere suitable for a business environment. The track should evoke a sense of motivation and energy, making use of synthesizers and drum beats to enhance its lively mood. The editing json dict: "editing instruction":"Remove the ambient textures and enhance the synthesizers", "caption of the edited audio":"An audio track that blends elements of funk and pop with an upbeat and exciting rhythm, featuring prominent bass guitar lines. The composition should be suitable for a business environment. The track should evoke a sense of motivation and energy, making use of enhanced synthesizers and drum beats to enhance its lively mood.".

You should randomly pick a thing/feature in the original audio caption to edit. Note that the editing instruction must be practical, i.e., the things/features to edit must be contained in the provided audio caption. The caption of the edited audio should be exactly aligned with that of the original audio, i.e., the only differences between the two captions should be those things/features that need to be edited. Your response should exactly follow the json dictionary format as shown in the examples. Do not generate anything else.

Audio Caption:
The editing json dict:

---

**Text2Audio Adaptation-Rectification Prompt - Rectification**

**System:** In this task, I want you to act as an inspector of the short audio generation task, and imagine that you are inspecting the quality of the designed audio generation tasks.

**Main:** You will be provided with a task dictionary which is designed to instruct audio generation models to generate and edit audios, and you need to judge whether each component is valid. The task contains 2 turns, each with a user prompt and its corresponding audio caption, and there is no answer or generated audios in the task dictionary. The first-turn user prompt is mainly designed to instruct the audio generation model to generate audios, and the first-turn caption is the corresponding caption format of the first-turn user prompt, which contains the same features/elements as that of the first-turn user prompt. The second-turn user prompt is a short prompt designed to instruct the audio generation model to edit audios generated in the first turn, and the second-turn caption is the caption designed for the edited audio, which is almost the same as the first-turn caption except for the items/features being edited. In the inspection json dictionary, answer 'Yes' if a component is correct; answer 'No' and specify the reason if it's not. Below examples only illustrate the format and some simple situations, while it does not cover all conditions.

Example 1:
Task json dictionary: "turn1-prompt": "Craft an audio message that succinctly conveys the essential details one might include in an SMS, ensuring the tone is informative yet brief, suitable for mobile notification sounds.", "turn1-caption": "An audio message that succinctly conveys the essential details one might include in an SMS, ensuring the tone is informative, suitable for mobile notification sounds.", "turn2-prompt": "Make the background music louder.", "turn2-caption": "An audio message that succinctly conveys the essential details one might include in an SMS, ensuring the tone is informative, suitable for mobile notification sounds. Make the background music louder." Your inspection json dictionary: "turn1-prompt correct": "Yes", "turn1-caption correct": "No, the 'brief' is missing from the turn1-prompt.", "turn2-prompt correct": "No. 'background music' is not contained in the turn-1 caption.", "turn2-caption correct": "No. 'background music' is not contained in the turn-1 caption, so this change is invalid."

Example 2:
Task json dictionary: "turn1-prompt": "Produce a 15-min audio snippet featuring a stern yet professional voice explaining the necessity of paying fees exclusively within Russia, emphasizing the rule's importance and potential consequences of non-compliance.", "turn1-caption": "A 15-min audio snippet featuring a stern yet professional voice explaining the necessity of paying fees exclusively within Russia, emphasizing the rule's importance and potential consequences of non-compliance.", "turn2-prompt": "Remove the emphasis on the rule's importance and potential consequences of non-compliance", "turn2-caption": "A 15-min audio snippet featuring a stern yet professional voice explaining the necessity of paying fees exclusively within Russia." Your inspection json dictionary: "turn1-prompt correct": "No. The expected audio is too long. It should be less than 120 seconds instead.", "turn1-caption correct": "Yes", "turn2-prompt correct": "Yes", "turn2-caption correct": "Yes"

Given a task dictionary, check whether each component is correct in logic and format. Especially check the below four aspects: 1. Whether the turn1-prompt is a valid audio generation request that instructs a model to generate second-level audios less than 120 seconds (the length restriction is not required to be included in the turn1-prompt, only inspect the estimated length of the content to generate); 2. Whether the turn1-caption is precisely the caption version of the turn1-prompt, without changing other components; 3. Whether the turn2-prompt is a valid audio editing request, with all its editing components exactly being contained in the turn1-caption; 4. Whether the turn2-caption is a

valid caption for the edited audio, which is supposed to be exactly aligned with turn1-caption, i.e., the only differences between the two captions should be those things/features that need to be edited. Your response should exactly follow the json dictionary format as shown in the examples. Your response should only contain the json dictionary and do not generate anything else.

Task json dictionary:
Your inspection json dictionary:

---

**Text2Audio Task Rewritter Prompt**

**System:** In this task, I want you to act as a caption extractor, and imagine that you are extracting captions from user instructions for audio generation.

**Main:** You will be provided with a user instruction that asks an audio generation model to generate audios. Below are some examples.

Example 1:
User Instruction: Create an audio clip of a calm and composed voice describing an outfit choice for a formal event in detail, including the color and material of the clothing, any accessories being worn, and the reasons behind these choices. The Extracted Caption: A calm and composed voice describing an outfit choice for a formal event in detail, including the color and material of the clothing, any accessories being worn, and the reasons behind these choices.

Example 2:
User Instruction: Give me an audio track that blends elements of funk and pop with an upbeat and exciting rhythm, featuring prominent bass guitar lines. The composition should include ambient textures to create a dynamic atmosphere suitable for a business environment. The track should evoke a sense of motivation and energy, making use of synthesizers and drum beats to enhance its lively mood. The Extracted Caption: An audio track that blends elements of funk and pop with an upbeat and exciting rhythm, featuring prominent bass guitar lines. The composition should include ambient textures to create a dynamic atmosphere suitable for a business environment. The track should evoke a sense of motivation and energy, making use of synthesizers and drum beats to enhance its lively mood.

The extracted caption must contain exactly the same content as that of the user instruction, i.e., do not add or reduce any audio content/feature description that is included in the original user instruction. You only need to change the format from the instruction format to the caption format. Your response should directly start with and only contain the extracted caption. Do not generate anything else.

User Instruction:
The Extracted Caption:

---

**Text2Action Adaptation-Rectification Prompt - Adaptation**

**System:** In this task, I want you to act as a task designer, and imagine you are designing real-world planning tasks that are not too complicated to break down, to be executed by embodied agents or robots in text.

**Main:** You will only be provided with an instruction, which implies the topic of the task content, and you need to make the task content complete by designing the task description, the allowed actions, the visible objects, and the already executed actions. The designed task content will be used to test the planning abilities of embodied agents or robots.

The below examples are simplified, while your completed task description could be either detailed or concise, depending on the context.

Example 1:
<Instruction>: How about an egg feast? The task json dictionary: "task description":"Put a heated egg in the sink.", "allowed actions":"[OpenObject], [CloseObject], [PickupObject], [PutObject], [ToggleObjectOn], [ToggleObjectOff], [SliceObject], [Navigation]","visible objects":"<microwave>, <sink>, <toaster>, <coffee maker>, <fridge>, <blender>, <potato>, <bows>, <egg>,

<garbagecan>","already executed actions":"[Navigation] <fridge>, [OpenObject] <fridge>, [PickupObject] <egg>, [CloseObject] <fridge>, [Navigation] <microwave>, [PutObject] <egg><microwave>"

Example 2:
<Instruction>: rush for the ticket! The task json dictionary: "task description": "Purchase tickets in that hall.", "allowed actions": "[Navigation], [InteractWithObject], [PickupObject], [PutObject], [UseObject], [Speak], [Listen], [PaymentTransaction], [IdentifyObject]", "visible objects": "<ticket booth>, <information desk>, <seats>, <hall entrance>, <hall exit>, <ticket machine>, <posters>, <map>, <cash>, <credit card>, <other visitors>, <staff members>", "already executed actions": "[Navigation] <hall entrance>, [IdentifyObject] <ticket booth>, [Speak] <staff members>"

Example 3:
<Instruction>: Click here for the L2TP setup The task json dictionary: "task description": "Set up a new L2TP VPN connection in the Network Preferences.", "allowed actions": "[Navigation], [Click], [InputText], [ToggleSwitch], [ConfirmAction], [ReadText], [Scroll], [OpenApplication], [CloseApplication], [OpenMenu], [ChooseNetworkType], [EnterCredentials], [SaveSettings]", "visible objects": "<computer>, <network preferences menu>, <VPN option>, <L2TP option>, <server address field>, <account name field>, <password field>, <shared secret field>, <save button>, <cancel button>, <status indicators>, <dropdown menus>, <text fields>, <checkboxes>", "already executed actions": "[Navigation] <computer>, [OpenMenu] <network preferences menu>, [Click] <VPN option>, [Click] <L2TP option>, [InputText] <server address field>, [InputText] <account name field>"

Considering the above examples, given an <Instruction>that implies the topic of the task content, brainstorm and complete the task design. For each planning task you should randomly pick a setting that has a similar topic to the given <Instruction>and directly generate the specific information. The 'task description' should be natural and sound like a user instruction and make sure the task is not too complexed to break down and is practical to execute for embodied agents or robots. The provided 'allowed actions' and 'visible objects' should be highly practical for the designed planning task, and they should be more than those required to plan this task, so that it is more challenging for the embodied agents to plan. Each element in the 'allowed actions' should be placed in a pair of square brackets '[]', and each element in the 'visible objects' should be placed in a pair of angle brackets '<>'. The same applies to the elements in the 'already executed actions' which are combinations of elements from 'allowed actions' and 'visible objects', and sometimes an element in 'already executed actions' may be a combination of an element from 'allowed actions' and multiple elements from 'visible objects', e.g., '[PutObject] <egg><microwave>'. The 'already executed actions' contains action-object sequences that are assumed to have been completed by the embodied agents for the designed task and you should control an appropriate number of elements in the 'already executed actions' as a hint for the embodied agents to plan the remaining actions for the designed task. Note that the 'already executed actions' should not contain the plans to be completed by the embodied agents. Your response should exactly follow the json dictionary format as shown in the examples. Your response should only contain the json dictionary and do not generate anything else.

<Instruction>:
The task json dictionary:

## Text2Action Adaptation-Rectification Prompt - Rectification

**System:** I want you to act as a task verifier and rewriter, and imagine you are verifying real-world action planning tasks to be executed by embodied agents or robots in text.

**Main:** You will be provided with a task json dictionary, which formulates a real-world action planning task. The json dictionary contains four keys: 'task description', 'allowed actions', 'visible objects', and 'already executed actions'. The 'task description' is a user instruction that instructs the embodied agents or robots to complete the task. The 'allowed actions' is a list of actions that are allowed to be used by the embodied agents or robots to complete the task. The 'visible objects' is a list of objects that are visible to the embodied agents or robots when they are completing the task. The 'already executed actions' is a list of action-object sequences that are assumed to have been completed by the embodied agents or robots at the time of task designing. You need to verify whether the task is a valid

action planning task in terms of the following requirements:

Requirement 1: The task format should be correct. It should contain the above-mentioned four keys and their corresponding values; each element in the 'allowed actions' should be placed in a pair of square brackets '[]', and each element in the 'visible objects' should be placed in a pair of angle brackets '<>'; each element of the 'already executed actions' is exactly a combination of one action element from the 'allowed actions' and one or more object elements from the 'visible objects'.

Requirement 2: The task description should be a natural user instruction. It should not be too complicated to break down and should be practical to execute for embodied agents or robots.

Requirement 3: The provided 'allowed actions' and 'visible objects' should be highly practical for the designed planning task, i.e., the actions in the 'allowed actions' and the objects in the 'visible objects' should be commonsensical in the designed task and its environment.

Requirement 4: The provided 'allowed actions' and 'visible objects' should be redundant, i.e., they should be more than the real requirement of the designed task.

Requirement 5: The 'already executed actions' should have an appropriate number of steps and should be correct in logic as a part of the actions of the designed task.

Requirement 6: The 'already executed actions' should not contain the plans to be completed by the embodied agents, i.e., the task is not completely solved given the 'already executed actions'.

Below are two simplified examples.

Example 1:
Task json dictionary:  "task description":"Put a heated egg in the sink.", "allowed actions":"[OpenObject], [CloseObject], [PickupObject], [PutObject], [ToggleObjectOn], [ToggleObjectOff], [SliceObject], [Navigation]","visible objects":"<microwave>, <sink>, <toaster>, <coffee maker>, <fridge>, <blender>, <potato>, <bows>, <egg>, <garbagecan>","already executed actions":"[Navigation] <fridge>, [OpenObject] <fridge>, [PickupObject] <egg>, [CloseObject] <fridge>, [Navigation] <microwave>, [PutObject] <egg><microwave>" Your verification (and rewriting): "Requirement 1": "Yes", "Requirement 2": "Yes", "Requirement 3": "Yes", "Requirement 4": "Yes", "Requirement 5": "Yes", "Requirement 6": "Yes", "need rewrite?": "No", "rewritten version": ""

Example 2:
Task json dictionary: "task description": "Purchase tickets in that hall.", "allowed actions": "Navigation, InteractWithObject, PickupObject, PutObject, UseObject, Speak, Listen, PaymentTransaction, IdentifyObject, CloseApplication, OpenMenu, ChooseNetworkType", "visible objects": "<fridge>, <blender>, <potato>, <ticket booth>, <information desk>, <seats>, <hall entrance>, <hall exit>, <ticket machine>, <credit card>, <other visitors>, <staff members>", "already executed actions": "Navigation <hall entrance>, IdentifyObject <ticket booth>, Speak <staff members>"
Your verification (and rewriting): "Requirement 1": "Yes", "Requirement 2": "No", "Requirement 3": "No", "Requirement 4": "Yes", "Requirement 5": "Yes", "Requirement 6": "Yes", "need rewrite?": "Yes", "rewritten version": ""task description": "Purchase tickets in that hall.", "allowed actions": "[Navigation], [InteractWithObject], [PickupObject], [PutObject], [UseObject], [Speak], [Listen], [PaymentTransaction], [IdentifyObject]", "visible objects": "<ticket booth>, <information desk>, <seats>, <hall entrance>, <hall exit>, <ticket machine>, <posters>, <map>, <cash>, <credit card>, <other visitors>, <staff members>", "already executed actions": "[Navigation] <hall entrance>, [IdentifyObject] <ticket booth>, [Speak] <staff members>""

In the above two examples, the first one is a qualified task, and the second one isn't because it does not meet Requirements 2 and 3. Any unqualified task requires to be modified or rewritten, and if you think it is qualified, the 'rewritten version' should be empty. Considering the above context and examples, verify (and rewrite) the below task json dictionary. Please exactly follow the json format shown in the above examples. Your response should only contain the verification json dictionary, do not generate anything else.

Task json dictionary:
Your verification (and rewriting):

## Text2Action Adaptation-Rectification Prompt - Reference Answer Annotation

**System:** In this task, I want you to act as a real-world agent, and you will plan action-object sequences for the real-world tasks.

**Main:** You will be provided with a task json dictionary, which formulates a real-world action planning task. The json dictionary contains four keys: 'task description', 'allowed actions', 'visible objects', and 'already executed action-object sequences'. The 'task description' is a user instruction that instructs you to complete the task. The 'allowed actions' is a list of actions that are allowed to be used by you to complete the task. The 'visible objects' is a list of objects that are visible to you when you are completing the task. The 'already executed action-object sequences' is a list of action-object sequences that are assumed to have been completed by you. You need to plan the remaining action-object sequences to complete the task.

Below are two simplified examples.

Example 1:
Task json dictionary: "task description":"Put a heated egg in the sink.", "allowed actions":"[OpenObject], [CloseObject], [PickupObject], [PutObject], [ToggleObjectOn], [ToggleObjectOff], [SliceObject], [Navigation]","visible objects":"<microwave>, <sink>, <toaster>, <coffee maker>, <fridge>, <blender>, <potato>, <bows>, <egg>, <garbagecan>","already executed action-object sequences":"[Navigation] <fridge>, [OpenObject] <fridge>, [PickupObject] <egg>, [CloseObject] <fridge>, [Navigation] <microwave>, [PutObject] <egg><microwave>"
Your planning: [ToggleObjectOn] <microwave>, [ToggleObjectOff] <microwave>, [PickupObject] <egg>, [Navigation] <sink>, [PutObject] <egg><sink>

Example 2:
Task json dictionary: "task description": "Purchase tickets in that hall.", "allowed actions": "[Navigation], [InteractWithObject], [PickupObject], [PutObject], [UseObject], [Speak], [Listen], [PaymentTransaction], [IdentifyObject]", "visible objects": "<ticket booth>, <information desk>, <seats>, <hall entrance>, <hall exit>, <ticket machine>, <posters>, <map>, <cash>, <credit card>, <other visitors>, <staff members>, <tickets>", "already executed action-object sequences": "[Navigation] <hall entrance>, [IdentifyObject] <ticket booth>, [Speak] <staff members>"
Your planning: [Listen] <staff members>, [Navigation] <ticket booth>, [IdentifyObject] <ticket machine>, [InteractWithObject] <ticket machine>, [PickupObject] <cash>, [PaymentTransaction] <ticket machine>, [PickupObject] <tickets>.

Considering the above examples, given a task json dictionary, plan the remaining action-object sequences to complete the task. Each element in the 'allowed actions' are placed in a pair of square brackets '[]', and each element in the 'visible objects' are placed in a pair of angle brackets '<>'; the same applies to the elements in your planning, where each element is the combination of an element from 'allowed actions' and an element from 'visible objects'. Sometimes an element in your planning may be a combination of a special element from 'allowed actions' and multiple elements from 'visible objects', e.g., '[PutObject] <egg><microwave>', where the '[PutObject]' action is a special action that should be followed with two objects. Your planning should be efficient to complete the task description, do not plan irrelevant steps or miss crucial steps. Your response should directly start with and only contain the planned action-object sequences, do not generate anything else.

Task json dictionary:
Your planning:

---

**Image2Action Adaptation-Rectification Prompt - Adaptation**

**System:** In this task, I want you to act as a practical visual description designer.
**Main:** You will only be provided with a user prompt. The user prompt refers to an external image. Specifically, the user prompt is asking an embodied agent to perform a task based on the content of the image, and the image content is the whole picture of what the embodied agent sees. The content seen by the embodied agent is not provided and sometimes the user prompt may be unclear or noisy, but your job is to design the description of the content seen by the embodied agent that is specific. You need to imagine the seen content and write the concise description for it based on the user prompt. The below examples are simplified.

Example 1:
User Prompt: How do we dig in this landscape? The description of the content seen by the embodied agent: A rugged, mountainous landscape under a clear blue sky. A range of tall mountains with jagged peaks extends into the distance, suggesting a challenging environment for excavation.

Example 2:
User Prompt: Click the pic below to get full-size. The description of the content seen by the embodied agent: A computer screen displaying an image thumbnail within a digital photo gallery. The thumbnail shows a picturesque landscape, possibly hinting at a larger, more detailed image. Surrounding the thumbnail are user interface elements like a 'Click to Enlarge' button, and other thumbnails showcasing different images, indicative of a typical photo viewing or editing software environment.

Pick a practical and common setting for the possible seen contents based on the User Prompt and directly describe the content. You should be imaginary and the written description should be concrete and specific in the objects or elements contained, even though they may not be specified in the user prompt. Try to be concise, i.e., do not describe too many unrelated elements, describe those important ones instead. And make sure the described seen content is practical for the embodied agent to perform the task specified in the user prompt. Your response should directly start with and only contain the content description. Do not generate anything else.

User Prompt:
The description of the content seen by the embodied agent:

---

**Image2Action Adaptation-Rectification Prompt - Reference Answer Annotation**

**System:** You will be provided with a task json dictionary and its corresponding task image, which formulates a real-world action planning task with image input.
**Main:** The json dictionary contains three keys: 'task description', 'allowed actions', and 'already executed action-object sequences'. The 'task description' is a user instruction that instructs you to complete the task. The 'allowed actions' is a list of actions that are allowed to be used by you to complete the task. The 'already executed action-object sequences' is a list of action-object sequences that are assumed to have been completed by you. The corresponding task image contains the visible or hidden objects for you to complete the task and indicates the task environment. You need to plan the remaining action-object sequences to complete the task.

Below are two simplified examples.

Example 1:
Task json dictionary: "task description":"Get the egg from the fridge, and put the heated egg in the sink.", "allowed actions":"[OpenObject], [CloseObject], [PickupObject], [PutObject], [ToggleObjectOn], [ToggleObjectOff], [SliceObject], [Navigation]","already executed steps":"[Navigation] <fridge>, [OpenObject] <fridge>, [PickupObject] <egg>, [CloseObject] <fridge>, [Navigation] <microwave>, [PutObject] <egg><microwave>" <task image>:
Your planning: [ToggleObjectOn] <microwave>, [ToggleObjectOff] <microwave>, [Navigation] <sink>, [PickupObject] <egg>, [PutObject] <egg><sink>

Example 2:
Task json dictionary: "task description": "Purchase tickets from the counter.", "allowed actions": "[Navigation], [InteractWithObject], [PickupObject], [PutObject], [UseObject], [Speak], [Listen], [PaymentTransaction], [IdentifyObject]", "already executed steps": "[IdentifyObject] <counter>, [Navigation] <counter>, [Speak] <staff members>" <task image>:
Your planning: [Listen] <staff members>, [PickupObject] <cash>, [PaymentTransaction] <staff members>, [PickupObject] <tickets>

Considering the above examples, given a task json dictionary and its corresponding task image, plan the remaining action-object sequences to complete the task. Each action in the 'allowed actions' are placed in a pair of square brackets '[]', and each object is placed in a pair of angle brackets '<>'; the same applies to the elements in your planning, where each element is the combination of an action from 'allowed actions' and a visible or hidden object in the image. Sometimes an element in your planning may be a combination of a special action from 'allowed actions' and multiple visible or hidden objects in the image, e.g., '[PutObject] <egg><microwave>', where the '[PutObject]' action is a special action that should be followed with two objects. Note that the objects you can use are not only limited to those visible ones, there are also some hidden objects that exist for sure in the environment of the provided image and you can also use them based on your commonsense. Your planning should be efficient to complete the task description, do not plan irrelevant steps or miss crucial steps. Your response should directly start with and only contain the planned action-object

sequences, do not generate anything else.

<Instruction>:
<task image>:
Your planning:

## F    MODEL PARSE PROMPTS

**Image2Text Free-form**

In this task, I want you to act as a judge.

You will be provided with a question, its golden answer(s), and the model's answer, while the context of the question, which is one or more images, is not given here. Your task is to judge how correct the model's answer is based on the golden answer(s), without seeing the input images of the question, and then give a correctness score. The correctness score should be one of the below numbers: 0.0 (totally wrong), 0.1, 0.2, 0.3, 0.4, 0.5, 0.6, 0.7, 0.8, 0.9, or 1.0 (totally right). Your should first briefly give your reasoning process regarding how the model's answer conforms to or contradicts the golden answer(s), and then give the correctness score. The correctness score must strictly follow this format: [[score]], e.g., "The correctness score: [[0.5]]". Below are some examples.

Example 1:
Question: what is this advertising?
Golden Answer(s): <answer 1>garden annual; <answer 2>seeds; <answer 3>seeds; <answer 4>seeds; <answer 5>seeds; <answer 6>seeds; <answer 7>seeds; <answer 8>seeds; <answer 9>seeds; <answer 10>cole's garden annual
Model's Answer: Seed
Your Judgment: The golden answers consistently mention "seeds" suggesting an advertisement for a seed catalog. The model's answer, "Seed", aligns exactly with this description. The Correctness Score: [[1.0]]

Example 2:
Question: Who is making a face?
Golden Answer: <answer 1>child
Model's Answer: A man.
Your Judgment: The golden answer specifies a "child" making a face, but the model answered "A man", which is incorrect as it refers to a different age group. The Correctness Score: [[0.0]]

Example 3:
Question: what road is to the right?
Golden Answer: <answer 1>troublesome valley rd; <answer 2>troublesome valley rd.; <answer 3>troublesome valley; <answer 4>troublesome valley road; <answer 5>valley road; <answer 6>troublesome valley; <answer 7>troublesome valley road; <answer 8>troublesome valley ; <answer 9>troublesome valley rd; <answer 10>troublesome valley rd.
Model's Answer: troublesome road
Your Judgment: The golden answers all specify the name of the road as "troublesome valley rd" or variations of this phrase with consistent reference to "troublesome valley." The model's answer, "troublesome road," captures the "troublesome" aspect but omits the critical "valley" part of the name, which is crucial for full accuracy. Thus, the model's answer partially matches the golden answer but lacks complete specificity. The Correctness Score: [[0.6]]

Note that each one of the golden answers is considered correct. Thus if the model's answer matches any one of the golden answers, it should be considered correct. Judge the below case, give the brief reasoning process and the correctness score.

Question: prompt
Golden Answer(s): gold_ans
Model's Answer: response
Your Judgment:

**Image2Text Multiple-choice**

In this task, I want you to act as an option extractor.

You will be provided with a multiple-choice question, its options, and the model's answer, while the context of the question, which is one or more images, is not given here. Your task is to extract or judge which option is chosen by the model based on its response, without seeing the context of the question. The extracted option should be one of the provided option letters. Your should first briefly give your reasoning process, and then give the extracted option letter. The extracted option must strictly follow this format: [[option letter]], e.g., "The option chosen by the model: [[A]]."
Below are some examples.

Example 1:
Question: Where are the cast of the television show located in the image?
Options:
A. In the foreground
B. In the background
C. In the center
D. At the edges
Model's Answer: C. In the center
Your Judgment: The model's answer clearly states "C. In the center", indicating that the correct option, according to the model, is in the center. The option chosen by the model: [[C]].

Example 2:
Question: <image_1>on the left was painted during the
Options:
A. first or second century C. E.
B. sixth or seventh century C. E.
C. tenth or eleventh century C.E.
D. fourteenth or fifteenth century C. E.
Model's Answer: The correct answer is option D, the fourteenth or fifteenth century C.E.
Your Judgment: The model's response specifies "option D, the fourteenth or fifteenth century C.E." directly as the correct answer. The option chosen by the model: [[D]].

Example 3:
Question: what does the diagram show's you information about
Options:
A. Photosynthesis
B. The plant getting fed
C. A picture of the plant
D. What happens to a plant daily
Model's Answer: The diagram shows the process of photosynthesis, which is the process by which plants convert sunlight, carbon dioxide, and water into oxygen and glucose. Your Judgment: The model's answer mentions "the process of photosynthesis," which directly corresponds to option A, "Photosynthesis". Therefore, the correct option according to the model is photosynthesis. The option chosen by the model: [[A]].

Give the brief reasoning process and the extracted option for the below case:

Question: prompt
Options:
options
Model's Answer: response
Your Judgment:

**Video2Text Free-form**

In this task, I want you to act as a judge.

You will be provided with a question, its golden answer(s), and the model's answer, while the context of the question, which is one or more videos, is not given here. Your task is to judge how correct the model's answer is based on the golden answer(s), without seeing the input videos of the question, and

then give a correctness score. The correctness score should be one of the below numbers: 0.0 (totally wrong), 0.1, 0.2, 0.3, 0.4, 0.5, 0.6, 0.7, 0.8, 0.9, or 1.0 (totally right). Your should first briefly give your reasoning process regarding how the model's answer conforms to or contradicts the golden answer(s), and then give the correctness score. The correctness score must strictly follow this format: "[[score]]", e.g., "The correctness score: [[0.5]]". Below are some examples.

Example 1:
Question: what does this video want to express
Golden Answer(s): <answer 1>introduce method of playing
Model's Answer: Volleyball serve
Your Judgment: The model's answer "Volleyball serve" suggests a specific action, which may be part of what the video demonstrates. However, it misses the broader educational intent implied by the golden answer "introduce method of playing". Therefore, the answer is partially correct. The Correctness Score: [[0.5]]

Example 2:
Question: who do two other boys with surprised looks assist up?
Golden Answer(s): <answer 1>boy
Model's Answer: Boy.
Your Judgment: The model's answer "Boy." precisely matches the golden answer which states the two other boys assist a "boy". The Correctness Score: [[1.0]]

Example 3:
Question: what did the lady do at the end of the video after their performance
Golden Answer(s): <answer 1>picks up her phone
Model's Answer: Nothing.
Your Judgment: The model's answer "Nothing." directly contradicts the golden answer which states that the lady "picks up her phone" at the end of the video after their performance. Since the model's response completely misses the specific action described in the golden answer, it is incorrect. The Correctness Score: [[0.0]]

Note that each one of the golden answers is considered correct. Thus if the model's answer matches any one of the golden answers, it should be considered correct. Judge the below case, give the brief reasoning process and the correctness score.

Question: prompt
Golden Answer(s): gold_ans
Model's Answer: response
Your Judgment:

---

### Video2Text Multiple-choice

In this task, I want you to act as an option extractor.

You will be provided with a multiple-choice question, its options, and the model's answer, while the context of the question, which is one or more videos, is not given here. Your task is to extract or judge which option is chosen by the model based on its response, without seeing the context of the question. The extracted option should be one of the provided option letters. Your should first briefly give your reasoning process, and then give the extracted option letter. The extracted option must strictly follow this format: "[[option letter]]", e.g., "The option chosen by the model: [[A]]".
Below are some examples.

Example 1:
Question: What did he do to the car?
Options:
A. Paint the car
B. Put plastic over the car
C. Put metal over the car
D. Cut the car
Model's Answer: put plastic over the car.
Your Judgment: The model's response directly aligns with option B, which is "Put plastic over the

car." The response given is a paraphrase of this option without deviating in meaning. The option chosen by the model: [[B]]

Example 2:
Question: How did Eddie know Pam and Justin before Justin was killed?
Options:
A. They were part of the theater company
B. They were high school friends
C. They went to college together
D. They were cousins
E. They were siblings
Model's Answer: A.
Your Judgment: The model's answer directly provides the option letter "A." The option chosen by the model: [[A]]

Example 3:
Question: why do the people move in the same manner
Options:
A. uniform
B. dancing with the baby
C. exercising together
D. stay together
E. singing and dancing
Model's Answer: sing and dance
Your Judgment: The model's response "sing and dance" closely aligns with option E, which is "singing and dancing." The response provided is a direct paraphrase of this option, modifying only slightly the form of the words (from gerund to infinitive) but maintaining the same core activities described in the option. The option chosen by the model: [[E]]

When you think that the model's answer does not match any of the given options, please choose the option that is the closest to the model's answer.
Give the brief reasoning process and the extracted option for the below case.

Question: prompt
Options:
options
Model's Answer: response
Your Judgment:

---

## Audio2Text Free-form

In this task, I want you to act as a judge.

You will be provided with a question, its golden answer(s), and the model's answer, while the context of the question, which is one or more audios, is not given here. Your task is to judge how correct the model's answer is based on the golden answer(s), without seeing the input audios of the question, and then give a correctness score. The correctness score should be one of the below numbers: 0.0 (totally wrong), 0.1, 0.2, 0.3, 0.4, 0.5, 0.6, 0.7, 0.8, 0.9, or 1.0 (totally right). Your should first briefly give your reasoning process regarding how the model's answer conforms to or contradicts the golden answer(s), and then give the correctness score. The correctness score must strictly follow this format: [[score]], e.g., The correctness score: [[0.5]]. Below are some examples.

Example 1:
Question: Are the people isolated?
Golden Answer(s): <answer 1>no
Model's Answer: yes
Your Judgment: The model's answer contradicts the golden answer directly. The question asked if the people are isolated, to which the golden answer is "no," indicating that the people are not isolated. However, the model's answer is "yes," implying that the people are isolated. The correctness score: [[0.0]]

Example 2:
Question: Who is speaking?
Golden Answer(s): <answer 1>man
Model's Answer: men
Your Judgment: The model's answer is almost correct but imprecise. The question asked about the identity of the speaker, to which the golden answer specifies a singular "man." However, the model's answer is "men," which suggests multiple individuals rather than one. This small pluralization error suggests a misunderstanding of the query about the exact number of people speaking. The correctness score: [[0.6]]

Example 3:
Question: What did you hear after the door slamming?
Golden Answer(s): <answer 1>dog making noise
Model's Answer: dog
Your Judgment: The model's answer "dog" matches the golden answer's essential element, "dog making noise," by correctly identifying the dog. Although it omits "making noise," it captures the key information needed. The correctness score: [[1.0]]

Note that each one of the golden answers is considered correct. Thus if the model's answer matches any one of the golden answers, it should be considered correct. Judge the below case, give the brief reasoning process and the correctness score.

Question: prompt
Golden Answer(s): gold_ans
Model's Answer: response
Your Judgment:

---

## Text2Image - turn 1

In this task, you will act as an impartial judge for image generation tasks.

Please act as an impartial judge and evaluate the quality of an image generated by an AI assistant given the provided user prompt or caption.

You must first analyze the generated image based on the provided prompt carefully. After providing your analysis, you must give the final score on a scale of 1 to 10 (1 means extremely bad and 10 means extremely good), and the rating must strictly follow this format: [[rating]], for example: Rating: [[5]].

You should consider the following core aspects when analyzing the image:
1. **Alignment**: Assess how accurately the image reflects the given prompt. Check if all elements and requirements are correctly represented.
2. **Realism**: Judge if the image looks realistic and natural.
3. **Quality**: Identify if there's any flaw in the image, such as distortion, blurriness, or illogical representation of facial features, limbs, fingers, objects, or text. In addition, evaluate the overall quality of the image.

Analyze and judge the below case:

Generation Prompt/Caption: prompt1
Generated Image: image1
Your Analysis and Judgment:

---

## Text2Image - turn 2

In this task, you will act as an impartial judge for an image editing task.

You will be provided with an image to edit, the user prompt to edit the image, and the edited image. Your task is to evaluate the quality of the edited image based on the given information.

You must first analyze the edited image based on the provided editing prompt and the image to edit carefully. After providing your analysis, you must give the final score on a scale of 1 to 10 (1 means extremely bad and 10 means extremely good), and the rating must strictly follow this format: [[rating]], for example: Rating: [[5]].

You should consider the following core aspects when analyzing the image:
1. **Alignment**: Assess how accurately the edited image reflects the changes indicated in the given editing prompt. Check if all elements and requirements are correctly meeted.
2. **Consistency**: Evaluate if the edited image is consistent with the original image in terms of details, style, color, overall appearance, etc.
3. **Realism**: Judge if the edited image looks realistic and natural after the editing process.
4. **Quality**: Identify if there's any flaw in the edited image, such as distortion, blurriness, or illogical representation of facial features, limbs, fingers, objects, or text. In addition, evaluate the overall quality of the image.

Analyze and judge the below case:

Editing Prompt/Caption: prompt1
The Image to Edit:

## Text2Action

In this task, you will act as an impartial judge for a real-world planning task.

Your job is to evaluate the quality of the action-object sequences planned by an AI assistant for a real-world task. You will be provided with the Task Description, Allowed Actions, Visible Objects, Already Executed Action-Object Sequences, the target, and the model's response. The 'Task Description' is a user instruction that instructs the AI assistant, which is being evaluated, to complete the task. The 'Allowed Actions' is a list of actions that are allowed to be used by the AI assistant to complete the task. The 'Visible Objects' is a list of objects that are assumed to be visible to the AI assistant when it's completing the task. The 'Already Executed Action-Object Sequences' is a list of action-object sequences that are assumed to have been completed by the AI assistant at the moment of starting the planning. The 'Reference Answer' is an example action-object sequence output for your reference, which is annotated by a human and may not be the only correct answer. The 'Model Response' is the output of the AI assistant you are evaluating.

Your task is to analyze the model's response and evaluate how well it plans given the above-mentioned information and the reference answer. After providing your analysis, you must give the final score on a scale of 1 to 10 (1 means extremely bad and 10 means extremely good), and the rating must strictly follow this format: [[rating]], for example: Rating: [[5]].

Below is a simplified example of how to judge the model's response:

**Start of Example**
Task Description: Put a heated egg in the sink.
Allowed Actions: [OpenObject], [CloseObject], [PickupObject], [PutObject], [ToggleObjectOn], [ToggleObjectOff], [SliceObject], [Navigation]
Visible Objects: <microwave>, <sink>, <toaster>, <coffee maker>, <fridge>, <blender>, <potato>, <bows>, <egg>, <garbagecan>
Already Executed Action-Object Sequences: [Navigation] <fridge>, [OpenObject] <fridge>, [PickupObject] <egg>, [CloseObject] <fridge>, [Navigation] <microwave>, [PutObject] <egg><microwave>
Reference Answer: [ToggleObjectOn] <microwave>, [ToggleObjectOff] <microwave>, [PickupObject] <egg>, [Navigation] <sink>, [PutObject] <egg><sink>
Model Response: [PickupObject] <egg>, [Navigation] <sink>, [PutObject] <egg><sink>Your Analysis and Judgment: The model's response omits crucial steps for heating the egg, assuming it is already heated without evidence from prior actions. It correctly performs the transport and placement of the egg, using appropriate actions and objects. However, by neglecting the heating process essential to the task description, the response is incomplete. My Final Rating: [[3]].
**End of Example**

With the above description and example, analyze and judge the below case:

Task Description: task_description
Allowed Actions: allowed_actions
Visible Objects: visible_objects
Already Executed Action-Object Sequences: already_executed_steps
Reference Answer: target
Model Response: model_response
Your Analysis and Judgment:

## Image2Action

In this task, you will act as an impartial judge for a real-world planning task.

Your job is to evaluate the quality of the action-object sequences planned by an AI assistant with visual perception for a real-world task. You will be provided with the Task Description, Allowed Actions, Visible Objects, Already Executed Action-Object Sequences, the target, and the model's response. The 'Task Description' is a user instruction that instructs the AI assistant, which is being evaluated, to complete the task. The 'Allowed Actions' is a list of actions that are allowed to be used by the AI assistant to complete the task. The 'Visible Objects' is a list of objects that are assumed to be visible to the AI assistant when it's completing the task. Note that some invisible objects may still be usable to the AI assistant, but their existence must be consistent with the commonsense. The 'Already Executed Action-Object Sequences' is a list of action-object sequences that are assumed to have been completed by the AI assistant at the moment of starting the planning. The 'Reference Answer' is an example action-object sequence output for your reference, which is annotated by a human and may not be the only correct answer. The 'Model Response' is the output of the AI assistant you are evaluating.

Your task is to analyze the model's response and evaluate how well it plans given the above-mentioned information and the reference answer. After providing your analysis, you must give the final score on a scale of 1 to 10 (1 means extremely bad and 10 means extremely good), and the rating must strictly follow this format: [[rating]], for example: Rating: [[5]].

Below is a simplified example of how to judge the model's response:

**Start of Example**
Task Description: Get the egg from the fridge, and put the heated egg in the sink.
Allowed Actions: [OpenObject], [CloseObject], [PickupObject], [PutObject], [ToggleObjectOn], [ToggleObjectOff], [SliceObject], [Navigation]
Visible Objects: image1
Already Executed Action-Object Sequences: [Navigation] <fridge>, [OpenObject] <fridge>, [PickupObject] <egg>, [CloseObject] <fridge>, [Navigation] <microwave>, [PutObject] <egg><microwave>
Reference Answer: [ToggleObjectOn] <microwave>, [ToggleObjectOff] <microwave>, [PickupObject] <egg>, [Navigation] <sink>, [PutObject] <egg><sink>
Model Response: [PickupObject] <egg>, [Navigation] <sink>, [PutObject] <egg><sink>
Your Analysis and Judgment: The model's response omits crucial steps for heating the egg, assuming it is already heated without evidence from prior actions. It correctly performs the transport and placement of the egg, using appropriate actions and objects. However, by neglecting the heating process essential to the task description, the response is incomplete. My Final Rating: [[3]].
**End of Example**

With the above description and example, analyze and judge the below case:

Task Description: task_description
Allowed Actions: allowed_actions
Visible Objects: image2
Already Executed Action-Object Sequences: already_executed_steps
Reference Answer: target
Model Response: model_response
Your Analysis and Judgment:

## G  BENCHMARK POOL DETAILS

**Image2Text:** MMMU (Yue et al., 2024), MMBench (Liu et al., 2023b), SEED-Bench (Li et al., 2023b), SEED-Bench 2 (Li et al., 2024b), ChartQA (Masry et al., 2022), A-OKVQA (Schwenk et al., 2022), HallusionBench (Guan et al., 2024), MathVista (Lu et al., 2023), GQA (Hudson & Manning, 2019), MM-Vet (Yu et al., 2023b), ScienceQA (Saikh et al., 2022), DocVQA (Mathew et al., 2021), POPE (Li et al., 2023e), InfographicVQA (Mathew et al., 2022), Q-Bench (Wu et al., 2023), VisWiz (Gurari et al., 2018), and TextVQA (Singh et al., 2019)

**Video2Text:** ActivityNet-QA (Yu et al., 2019), HowToQA (Li et al., 2020), STAR (Wu et al., 2024), TVQA (Lei et al., 2018), TGIF-QA (Jang et al., 2017), EgoSchema (Mangalam et al., 2023), SUTD-TrafficQA (Xu et al., 2021), NextQA (Xiao et al., 2021), PororoQA (Kim et al., 2017), IVQA (Liu et al., 2018), WildQA (Castro et al., 2022), Perception-Test (Patraucean et al., 2024), MSVD-QA (Xu et al., 2017), and Social-IQ-2.0 (Zadeh et al., 2019)

**Audio2Text:** Clotho-AQA (Lipping et al., 2022), DAQA (Fayek & Johnson, 2020), and CLEAR (Lin et al., 2021)

## H  MODEL DETAILS

**Image2Text**: Claude 3.5 Sonnet (Anthropic, 2024a), GPT-4o (OpenAI, 2024a), GPT-4V (Achiam et al., 2023), Qwen2-VL-72B (Wang et al., 2024), Gemini 1.5 Pro (Reid et al., 2024), Llama 3.2 90B (Meta, 2024b), InternVL2-26B (Chen et al., 2023b), Claude 3 Opus (Anthropic, 2024b), Qwen-VL-MAX (Bai et al., 2023), LLaVA-1.6-34B (Liu et al., 2024b), Claude 3 Sonnet (Anthropic, 2024b), Reka Core (Ormazabal et al., 2024), Reka Flash (Ormazabal et al., 2024), InternVL-Chat-V1.2 (Chen et al., 2023b), Qwen-VL-PLUS (Bai et al., 2023), Claude 3 Haiku (Anthropic, 2024b), Gemini 1.0 Pro (Anil et al., 2023), InternLM-XComposer2-VL (Dong et al., 2024), Yi-VL-34B (Young et al., 2024), OmniLMM-12B (Yu et al., 2023a), DeepSeek-VL-7B-Chat (Lu et al., 2024a), Yi-VL-6B (Young et al., 2024), InfiMM-Zephyr-7B (Team, 2024c), MiniCPM-V (Yao et al., 2024), Marco-VL, LLaVA-1.5-13B (Liu et al., 2024b), SVIT (Zhao et al., 2023), mPLUG-OWL2 (Ye et al., 2024), SPHINX (Lin et al., 2023b), InstructBLIP-T5-XXL (Dai et al., 2023), InstructBLIP-T5-XL (Dai et al., 2023), BLIP-2 FLAN-T5-XXL (Li et al., 2023c), BLIP-2 FLAN-T5-XL (Li et al., 2023c), Adept Fuyu-Heavy (Team, 2024a), LLaMA-Adapter2-7B (Gao et al., 2023), Otter (Li et al., 2023a), MiniGPT4-Vicuna-13B (Zhu et al., 2023)

**Video2Text**: Claude 3.5 Sonnet (Anthropic, 2024a), GPT-4o (OpenAI, 2024a), Gemini 1.5 Pro (Reid et al., 2024), GPT-4V (Achiam et al., 2023), Qwen2-VL-72B (Wang et al., 2024), Gemini 1.5 Flash (Reid et al., 2024), LLaVA-OneVision-72B-OV (Li et al., 2024a), Qwen2-VL-7B (Wang et al., 2024), LLaVA-Next-Video-34B (Zhang et al., 2024a), Claude 3 Haiku (Anthropic, 2024b), LLaVA-Next-Video-7B (Zhang et al., 2024a), Reka-edge (Ormazabal et al., 2024), LLaMA-VID (Li et al., 2023d), VideoLLaVA (Lin et al., 2023a), Video-ChatGPT (Maaz et al., 2023), mPLUG-video (Li et al., 2022)

**Audio2Text**: Gemini 1.5 Pro (Reid et al., 2024), Gemini 1.5 Flash (Reid et al., 2024), Qwen2-Audio-7B-Instruct (Chu et al., 2024), Qwen2-Audio-7B (Chu et al., 2024), SALMONN-13B (Tang et al., 2023), Qwen-Audio (Chu et al., 2023), Qwen-Audio-Chat (Chu et al., 2023), SALMONN-7B (Tang et al., 2023), Pengi (Deshmukh et al., 2023)

**Text2Image**: Flux (BlackForestLabs, 2024), DALL·E 3 HD (Betker et al., 2023), PixArtAlpha (Chen et al., 2023a), PlayGround V2.5 (Li et al., 2024c), PlayGround V2 (Li et al., 2024c), SD1.5 (Rombach et al., 2022), SD3 (Esser et al.), SDXL (Podell et al., 2023), Stable Cascade (Pernias et al., 2023)

**Text2Video**: ModelScope (Wang et al., 2023a), ZeroScope V2, CogVideoX-5B (Yang et al., 2024), HotShot-XL (Mullan et al., 2023), LaVie (Wang et al., 2023b), Show-1 (Zhang et al., 2023), VideoCrafter2 (Chen et al., 2024)

**Text2Audio**: AudioLDM 2 (Liu et al., 2024a), Make-An-Audio 2 (Huang et al., 2023), Stable Audio (Evans et al., 2024), Tango 2 (Majumder et al., 2024), ConsistencyTTA (Bai et al., 2024), AudioGen (Kreuk et al., 2022), Magnet (Ziv et al., 2024)

**Text2Action**: GPT-4-Turbo (Achiam et al., 2023), Gemini 1.5 Pro (Reid et al., 2024), Mistral-Large-2 (Team, 2024b), GPT-4o (OpenAI, 2024a), Reka Core (Ormazabal et al., 2024), Claude 3.5 Sonnet (Anthropic, 2024a), Gemma-2-9B-Instruct (Team et al., 2024), Reka Flash (Ormazabal et al., 2024), Claude 3 Haiku (Anthropic, 2024b), Mistral-Medium (Jiang et al., 2023), Gemini 1.5 Flash (Reid et al., 2024), Qwen-2-72B-Instruct (Wang et al., 2024), LLaMA-3.1-70B-Instruct (Meta, 2024a), Mistral-Small (Jiang et al., 2023), GPT-4o-Mini (OpenAI, 2024b), Yi-1.5-34B-Chat (Young et al., 2024), LLaMA-3.1-8B-Instruct (Meta, 2024a), GPT-3.5-Turbo (Achiam et al., 2023), Qwen-2-7B-Instruct (Wang et al., 2024), Yi-1.5-9B-Chat (Young et al., 2024), Reka Edge (Ormazabal et al., 2024)

**Image2Action**: GPT-4V (Achiam et al., 2023), Claude 3.5 Sonnet (Anthropic, 2024a), GPT-4o (OpenAI, 2024a), Claude 3 Opus (Anthropic, 2024b), Claude 3 Sonnet (Anthropic, 2024b), Claude 3 Haiku (Anthropic, 2024b), Gemini 1.5 Pro (Reid et al., 2024), InternVL-Chat-V1.5 (Chen et al., 2023b), Qwen-VL-MAX (Bai et al., 2023), Qwen-VL-PLUS (Bai et al., 2023), InfiMM-Zephyr-7B (Team, 2024c), DeepSeek-VL-7B-Chat (Lu et al., 2024a), MiniCPM-V (Yao et al., 2024), Yi-VL-34B (Young et al., 2024), Qwen2-VL-72B (Wang et al., 2024), InternLM-XComposer2-VL (Dong et al., 2024), LLaVA-1.6-13B (Liu et al., 2024b), LLaVA-1.6-34B (Liu et al., 2024b)

Table 2: Correlations between Model Judges and Human Preference Elo

|          | GPT  | Claude | Gemini | Avg. |
|----------|------|--------|--------|------|
| 1st turn | 0.82 | 0.68   | 0.78   | 0.83 |
| 2nd turn | 0.67 | 0.56   | 0.6    | 0.58 |
| Avg.     | 0.75 | 0.8    | 0.83   | 0.78 |

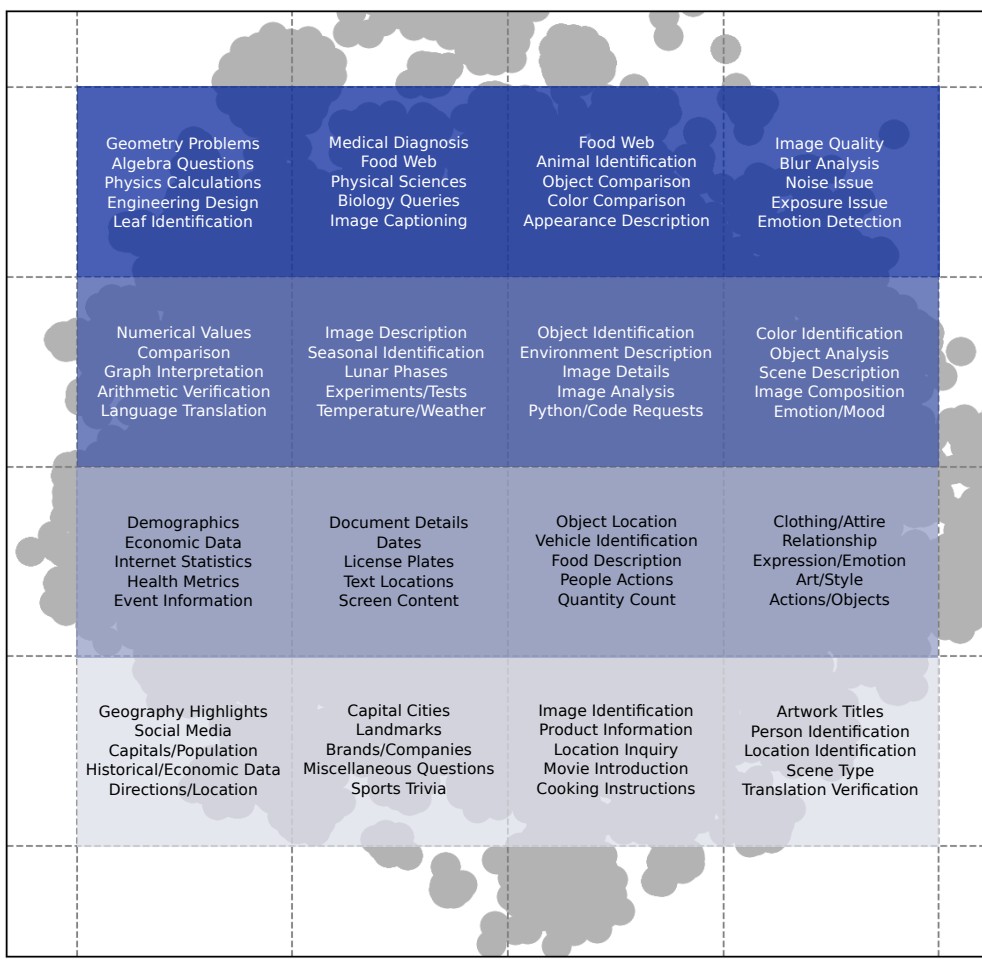

Figure 35: Query topic summarization for the Image2Text queries in Figure 9. The plot aggregates all queries and divides them into 16 regions. From each region, 100 queries are uniformly sampled and analyzed by GPT-4 for topic summarization.

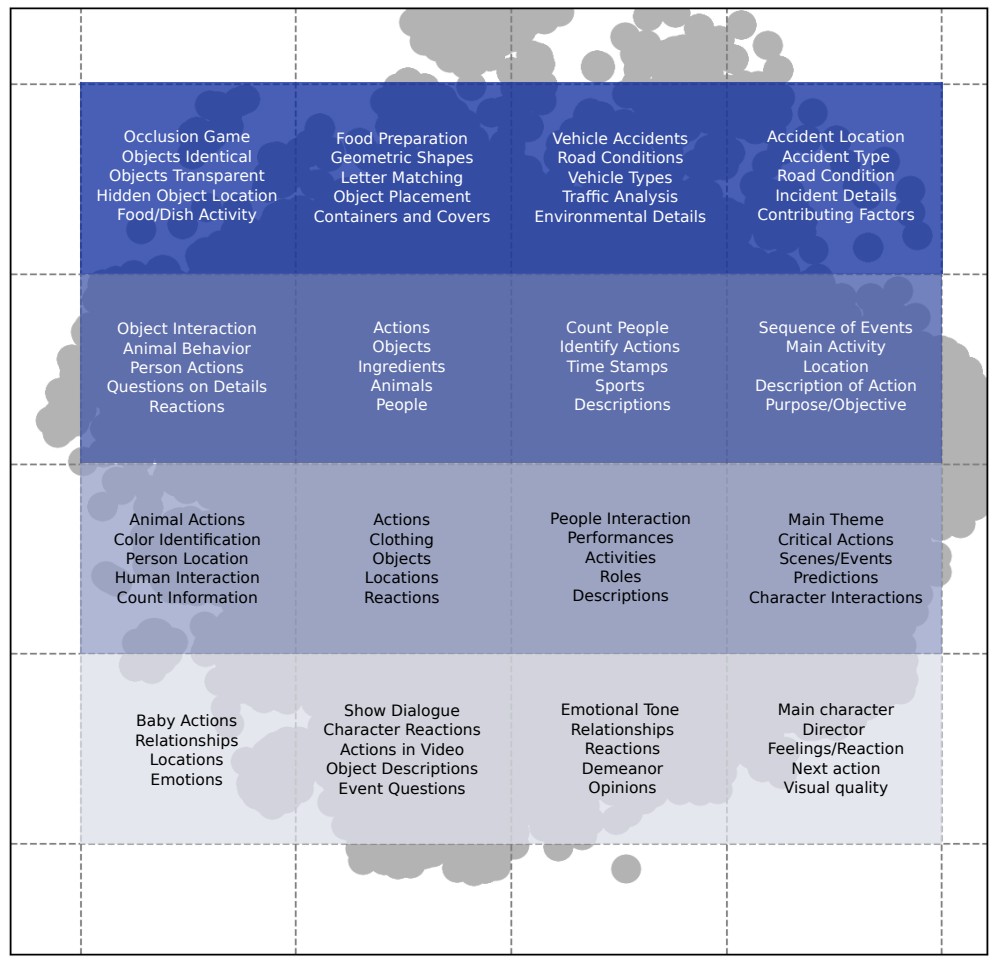

Figure 36: Query topic summarization for the Video2Text queries in Figure 9. The plot aggregates all queries and divides them into 16 regions. From each region, 100 queries are uniformly sampled and analyzed by GPT-4 for topic summarization.

| | | | |
|---|---|---|---|
| Location
Environment
Nearby Buildings
Piano Lyrics
Product | Weather Conditions
Water Sounds
Location
Objects and Actions
Miscellaneous | Instrument Notes
Sound Characteristics
Scene Position
Sequence of Sounds
Count of Specific Notes | Sound Position
Sound Loudness
Sound Brightness
Instrument Note |
| Activity Inquiry
Work Progress
Location Query
Speech Content
Miscellaneous | Speaking Detection
Speech Content
Ambient Sounds
Scene Description
Audio Captions | Animal Sounds
Vehicle Sounds
Musical Instruments
Natural Environment
Sound Characteristics | Brightness
Loudness
Sound Count
Sound Sequence
Volume Change |
| Conversations
Arguments
Contextual Inquiries | Audio Interpretation
Human Speech Analysis
Emotional Response
Podcast Preferences
Miscellaneous | Sound Counts
Sound Sequences
Sound Identification
Human Reactions
Comparative Sounds | Sound Duration
Sound Loudness
Sound Event Sequence
Sound Comparison
Specific Sound Inquiry |
| Invitations
Calls
Requests
Statements | Phone Ringing
Court/Legal
Personal Experiences
Questions
Credit Card WarninG | Human Typing
Emergency Vehicles
Phone Ringing
Crowds Rioting
Vehicles Passing | Loudness Comparison
Frequency Comparison
Equal Occurrences
Crowd vs. Other Noises
Shapiro Query |

Figure 37: Query topic summarization for the Audio2Text queries in Figure 9. The plot aggregates all queries and divides them into 16 regions. From each region, 100 queries are uniformly sampled and analyzed by GPT-4 for topic summarization.

Figure 38: Query topic summarization for the Text2Action queries in Figure 9. The plot aggregates all queries and divides them into 16 regions. From each region, 100 queries are uniformly sampled and analyzed by GPT-4 for topic summarization.

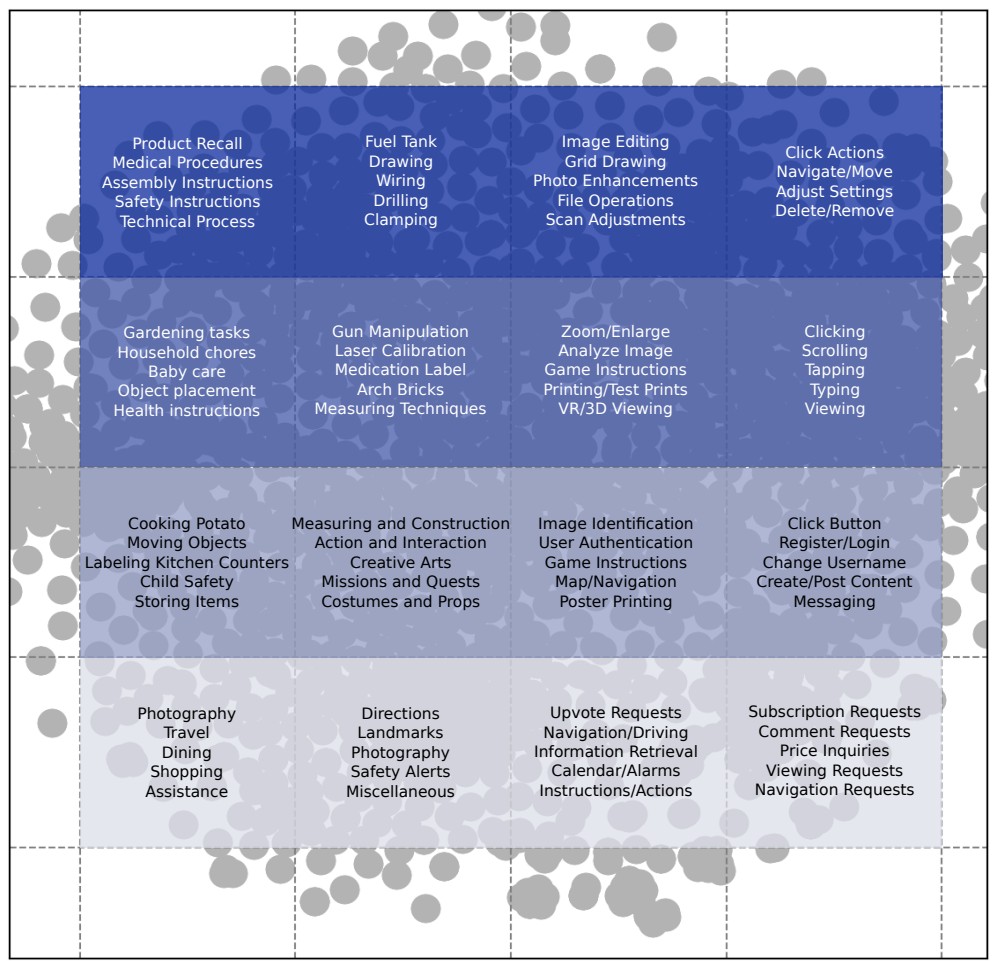

Figure 39: Query topic summarization for the Image2Action queries in Figure 9. The plot aggregates all queries and divides them into 16 regions. From each region, 100 queries are uniformly sampled and analyzed by GPT-4 for topic summarization.

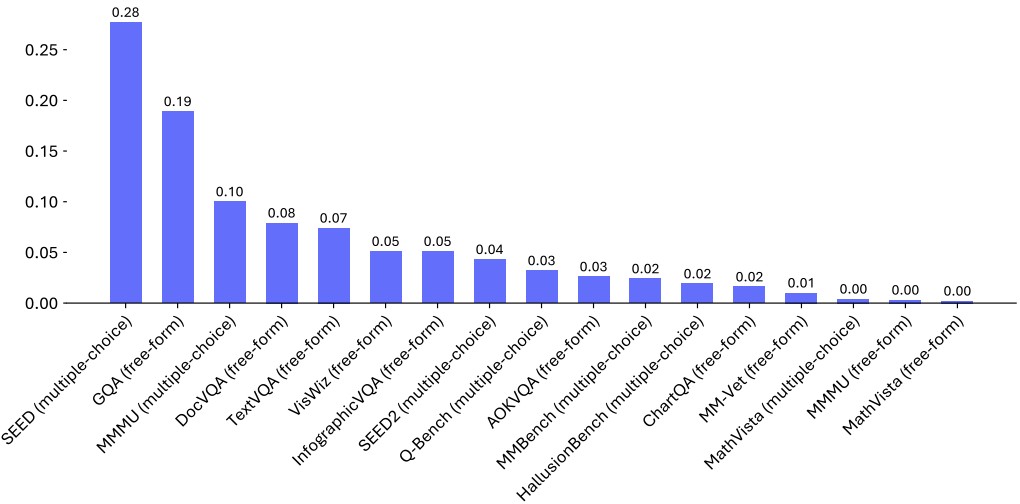

Figure 40: Image2Text benchmark pool distribution on benchmark level.

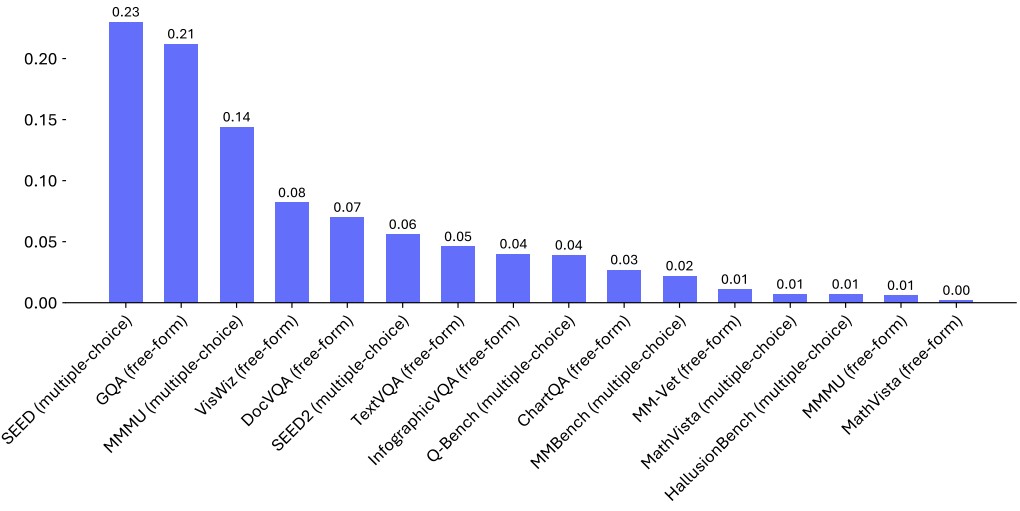

Figure 41: Image2Text-Hard benchmark pool distribution on benchmark level.

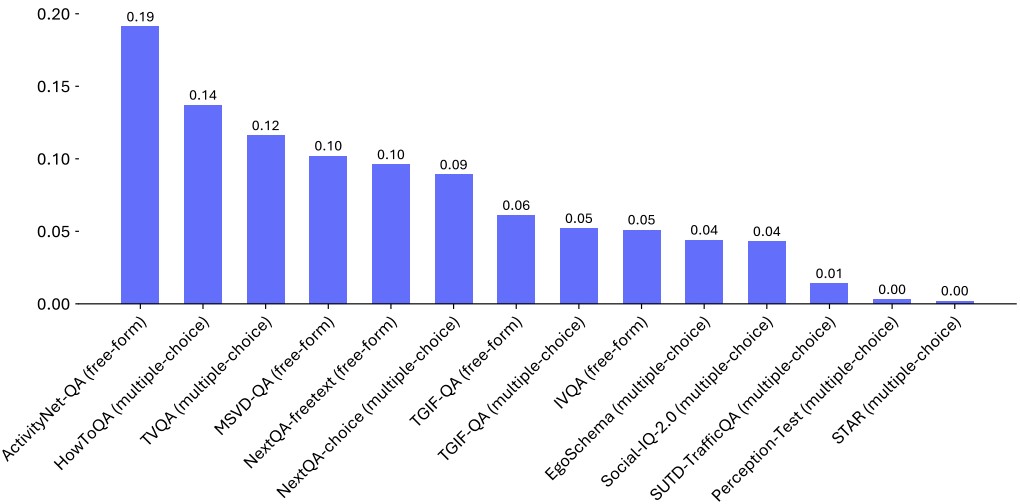

Figure 42: Video2Text benchmark pool distribution on benchmark level.

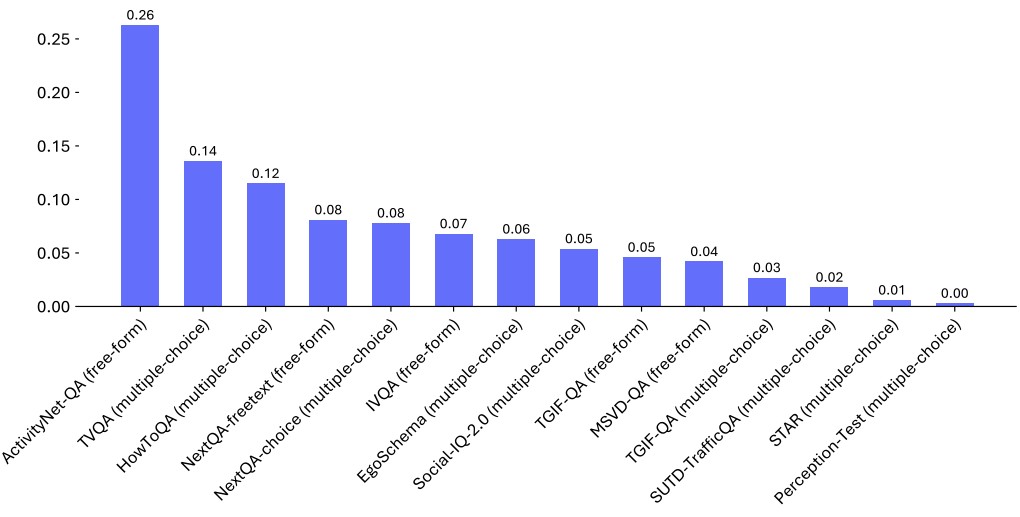

Figure 43: Video2Text-Hard benchmark pool distribution on benchmark level.

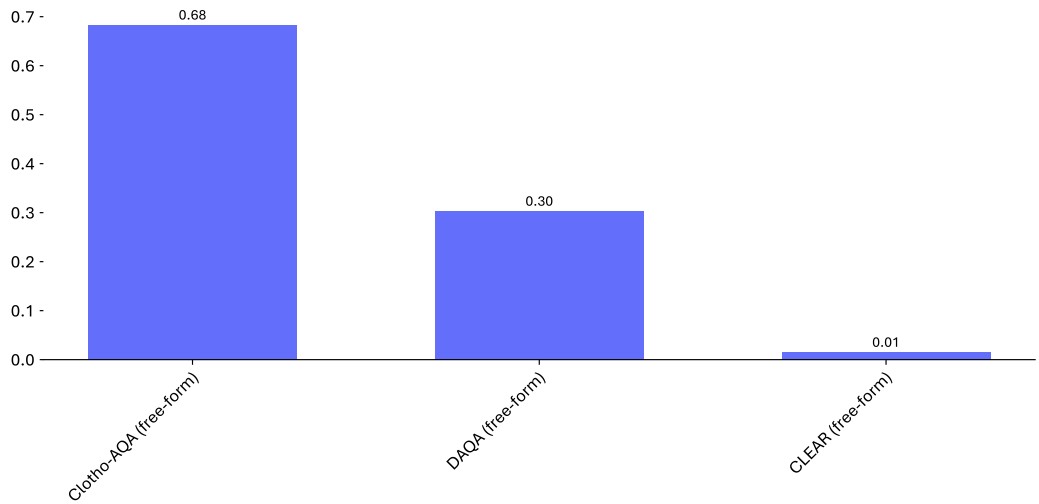

Figure 44: Audio2Text benchmark pool distribution on benchmark level.

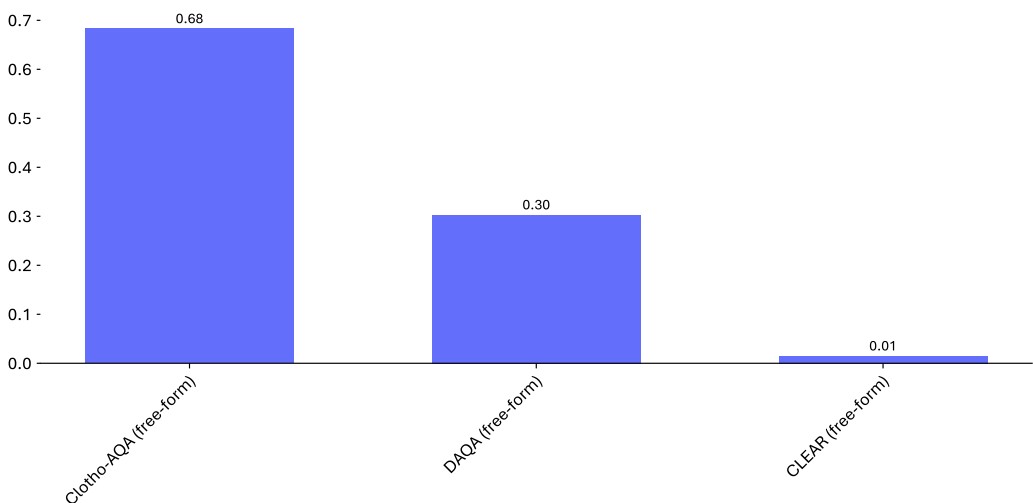

Figure 45: Audio2Text-Hard benchmark pool distribution on benchmark level.

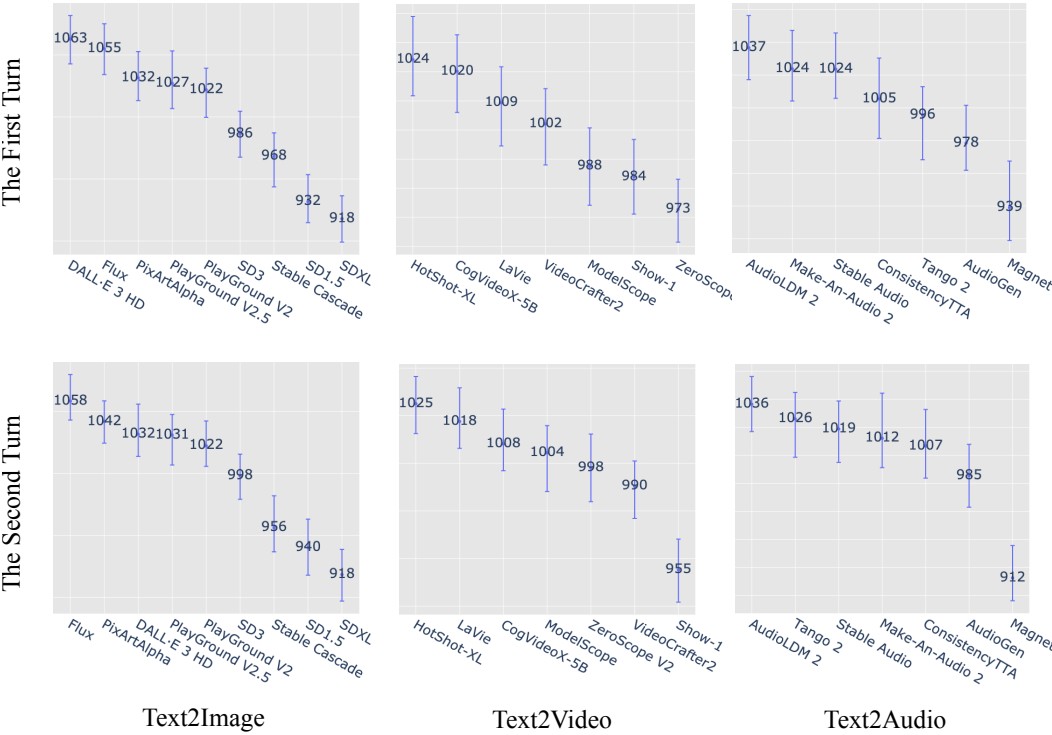

Figure 46: The turn-level scores of MMG tasks. MMG tasks are designed to be a two-turn interleaved ones, where the first turn is a generation task and the second turn is an editing task based on the content generated in the first turn and the history user instruction.

