# OpenReview forum: "MixEval-X: Any-to-any Evaluations from Real-world Data Mixture"
_ICLR.cc/2025/Conference — ICLR 2025 Spotlight_

### Official Review · Reviewer_fCLM · 2024-11-03

**Soundness:** 3
**Presentation:** 3
**Contribution:** 3
**Rating:** 8
**Confidence:** 2

**Summary:**

MixEval-X extends MixEval to support multiple modalities beyond text, comprising eight new subsets (excluding the original MixEval) categorized into three types: multi-modal understanding (MMU), multi-modal generation (MMG), and agent tasks.

- **Benchmark Creation**: The creation process differs for each type. For MMU tasks (Text2Text, Image2Text, Audio2Text), a benchmark mixture pipeline aligns web queries with similar tasks from existing benchmarks, similar to MixEval. For MMG and agent tasks, an adaptation-rectification pipeline standardizes real-world queries, though the exact adaptation-rectification process requires further clarification.
- **Evaluation**: Grading methods are customized for each task type. MMU and agent tasks are evaluated with LM-based parsers, with smaller models for MMU and Frontier LMs for agent tasks. MMG tasks, harder to automate, rely on crowd-sourced evaluation.
- **Intuition & Interesting Claim**: MixEval-X retains MixEval's core philosophy: simulating web-distributed questions as a proxy for real-world questions, forming a compressed benchmark for efficient evaluation. Image2Text outputs in MixEval-X show a high correlation with Vision Arena, supporting this approach. A key finding indicates that Judge models in multimodal-generation tasks often diverge from human preferences.

Overall, this work offers a comprehensive evaluation of models across diverse multimodal tasks.

**Strengths:**

- **[Important] Comprehensive Analysis & Leaderboard**: MixEval-X offers extensive benchmarks and detailed leaderboards to rerank existing models and organizations. I appreciated the thorough, high-standard benchmarks applied across modalities and communities.
- **[Important] Clarity**: The paper effectively motivates the setting for readers unfamiliar with the topic. It is easy to understand, with clearly stated contributions, proposed methods that address the defined problem, and well-presented and interpreted results. However, some sections may feel dense due to space constraints in the 10-page limit.

**Weaknesses:**

I have a few concerns asked in Questions section. However, note that I could not find major weaknesses specific to this work, which did not apply for the general MixEval framework.

Overall, the method is a compelling and substantial extension of MixEval, with sound intuition and strong performance demonstrated through extensive comparisons. I would be pleased to see this paper presented at ICLR, but note that I am not a domain expert -- looking forward to seeing the other reviews and discussions with the authors.

**Questions:**

Q1. **Automatic Benchmark Creation & Grading for Agent Tasks**
- a) Creation: Could you clarify if there are any specific checks and balances in place to ensure the action set was complete and the context made sense, especially if these tasks were primarily generated by models?
- b) Grading: Since the grading appears to be automated, was there any human oversight to detect and correct potential errors in this pipeline or checks to indicate what fraction of inputs/model behaviour intermediate was erroneous/undefined and not caught by the grader? Ensuring clarity here would be valuable, as automated grading might lack nuanced understanding.

Q2. **Scalability and Applicability of Automatic Evaluation**
- The paper claims scalability and automated evaluation, but these seem to primarily apply to MMU tasks. For MMG tasks, wouldn’t repeated sampling from new prompts or models require costly human annotation? It seems to me that the scalability only extends to certain tasks, does this align with the broader goals and spirit of the paper?

**Details Of Ethics Concerns:**

Nil.

---

> ### Author Response · Authors · 2024-11-18
>
> We thank the reviewer for the strong support and considering MixEval-X to be sound, comprehensive, thorough, and high-standard. Below we answer the related questions.
>
> *Note: Unless otherwise specified, the mentioned line numbers denote that of the originally submitted paper version instead of the rebuttal revision version.*
>
> ## Question 1
>
> > Creation: Could you clarify if there are any specific checks and balances in place to ensure the action set was complete and the context made sense, especially if these tasks were primarily generated by models?
>
> Yes, as illustrated in lines 194-196 of the originally submitted paper, the model-generated tasks are verified and revised automatically through our Adaptation-Rectification pipeline. **In addition, human inspection was introduced to ensure every task was appropriately designed.**
>
> ## Question 2
>
> > Grading: Since the grading appears to be automated, was there any human oversight to detect and correct potential errors in this pipeline or checks to indicate what fraction of inputs/model behaviour intermediate was erroneous/undefined and not caught by the grader? Ensuring clarity here would be valuable, as automated grading might lack nuanced understanding.
>
> **Yes, we introduced quality control steps and human inspections to ensure the grading process is well-interpreted and accurate.**
>
> As mentioned in lines 177-179 of the originally submitted paper, we have a quality control step to improve the ground truth quality of MixEval-X MMU tasks. As detailed in lines 194-196, agent tasks are further verified by humans to enhance the task and annotation quality. The MMG tasks are all graded by a large group of humans, being robust due to the wisdom of the crowd effect [1].
>
> Beyond the above quality control steps in the pipeline, we also introduced mannual inspections on the model outputs and the grading process for all sub-benchmarks after running evaluations, to understand and ensure robustness of the grading pipeline.
>
> ## Question 3
>
> > The paper claims scalability and automated evaluation, but these seem to primarily apply to MMU tasks. For MMG tasks, wouldn’t repeated sampling from new prompts or models require costly human annotation? It seems to me that the scalability only extends to certain tasks, does this align with the broader goals and spirit of the paper?
>
> Thanks for raising this important question! The grading of MMG tasks (i.e., the image, audio, video genertion) is indeed challenging due to their open-ended nature. As discussed in lines 214-215 of the originally submitted paper, we do not restrict the grading methods for MMG tasks. **The users of MixEval-X may choose between automatic metrics, model-based grading, or human judgment depending on the specific application**. For example, if an MMG model developer prefers rapid model iteration, he may choose automatic or model-based grading to evaluate the model quickly (maybe at the sacrifice of robustness). In Section 4.2, we further discussed the reliability of model-based MMG grading methods via experiments, highlighting the need for further research into cost-effective, low-bias grading methods for MMG tasks. It is worth noting that the grading framework of MixEval-X MMG sub-benchmarks can be adapted in the future as more robust and efficient methods become available.
>
> ## References
>
> [1] Sheng Kung Michael Yi et. al. The wisdom of the crowd in combinatorial problems.

---

### Official Review · Reviewer_Edzp · 2024-11-03

**Soundness:** 4
**Presentation:** 4
**Contribution:** 4
**Rating:** 8
**Confidence:** 4

**Summary:**

This paper introduces MixEval-X, a new any-to-any capability benchmark for assessing performance of frontier multimodal models across a diverse set of tasks and input-output modality combinations. The paper proposes to use the framework introduced in MixEval to perform benchmark mixtures from pre-existing benchmarks to construct a well-curated benchmark that is more generic and follows real-world task distributions.

**Strengths:**

- This is a very comprehensive piece of work that is very well motivated and provides a significant step in unifying multimodal frontier model evaluation going forward.

- There is a substantial amount of empirical evaluations and analyses conducted on the benchmark and its constituent sub-benchmarks to showcase its real-world utility.

- In particular, the combination of figs 36-39 + fig 9 is particularly striking. It can clearly help model developers assess which benchmarks are probing which particular capabilities to guide model development.

**Weaknesses:**

- A key limitation of the work is the lack of any discussion surrounding model evaluation costs. I believe it is important to provide details on how much it costs to evaluate any frontier models on the benchmark, i.e. both the cost of running models / APIs on the benchmark tasks but also the cost of using LLMs as judges.

- It is not clear which set of sub-benchmarks use automated LLMs-as-judges, humans as annotators and which others use pre-existing ground-truth information for evaluating models. It might well be that each sub-benchmark uses a combination of these, but I believe it would be useful to augment Tab 1 with this information on what set of ground-truths and evaluation strategies each sub-benchmark uses (i.e. LLM-as-judge, preferences, ground-truth etc).

- The challenge of test-set contamination: From the construction of the benchmark, it is not made abundantly clear how the proposed benchmark alleviates the problem of test-set contamination---as I understand, it is merely a combination of pre-existing benchmark samples in a principled manner by aligning with the real-world task distribution? If this is the case, it does indeed ensure a more robust distribution in the evaluation splits as well as better alignment to real-world usage, but it does not actually alleviate the concern of test-set contamination. A broad discussion point on this would be useful.

**Questions:**

- Where are the actual web-queries being mined from? Would it possible to add some details around this mining strategy as well as add a few of these web-queries to give a feel of what the real-world task distribution looks like?

- For the non-hard version of the benchmarks, the best models already perform quite well (76.9%, 74.2%, 62.7%). Does this mean that this version of the benchmark is quite easy for current frontier models already, and has a lower saturation ceiling? Could the authors please comment on this.

---

> ### Author Response · Authors · 2024-11-18
>
> We thank the reviewer for considering our work to be comprehensive, well-motivated, and groundbreaking. The reviewer’s strong support is very important to us. Below are our responses to the concerns.
>
> *Note: Unless otherwise specified, the mentioned line numbers denote that of the originally submitted paper version instead of the rebuttal revision version.*
>
> ## Concern 1
>
> > A key limitation of the work is the lack of any discussion surrounding model evaluation costs. I believe it is important to provide details on how much it costs to evaluate any frontier models on the benchmark, i.e. both the cost of running models / APIs on the benchmark tasks but also the cost of using LLMs as judges.
>
> Thank you for raising this important concern. Indeed, the cost is an essential consideration when running evaluations. **Below we report key cost statistics.** Note that the cost may vary based on different settings, hardwares, and models.
>
> Table 2: The cost statistics for running MixEval-X sub-benchmarks. The open-source models are evaluated on a single node with 8*80G A100 GPUs.
> |    Benchmark    |        Model       | Inference Cost | Judge Cost |
> |:---------------:|:------------------:|:--------------:|:----------:|
> |    Image2Text   |     Qwen2-VL-7B    |     16 min     |     $3     |
> |    Image2Text   |       GPT-4o       |       $10      |     $5     |
> | Image2Text-Hard |     Qwen2-VL-7B    |      9 min     |    $1.5    |
> | Image2Text-Hard |       GPT-4o       |       $6       |    $2.5    |
> |    Video2Text   |     Qwen2-VL-7B    |     37 min     |    $3.5    |
> |    Video2Text   |       GPT-4o       |       $30      |     $5     |
> | Video2Text-Hard |     Qwen2-VL-7B    |     20 min     |    $1.5    |
> | Video2Text-Hard |       GPT-4o       |       $17      |    $2.5    |
> |    Audio2Text   |   Qwen2-Audio-7B   |     25 min     |     $3     |
> |    Audio2Text   |   Gemini 1.5 Pro   |       $12      |     $4     |
> | Audio2Text-Hard |   Qwen2-Audio-7B   |     12 min     |    $1.5    |
> | Audio2Text-Hard |   Gemini 1.5 Pro   |       $7       |     $2     |
> |   Text2Action   | Qwen-2-7B-Instruct |     5 min    |     $5     |
> |   Image2Action  |     Qwen2-VL-7B    |     6 min    |     $9     |
> |    Text2Image   |        FLUX        |      8 min     |      -     |
> |    Text2Video   |     HotShot-XL     |     12 min     |      -     |
> |    Text2Audio   |     AudioLDM 2     |     12 min     |      -     |
>
>
> ## Concern 2
>
> > It is not clear which set of sub-benchmarks use automated LLMs-as-judges, humans as annotators and which others use pre-existing ground-truth information for evaluating models. It might well be that each sub-benchmark uses a combination of these, but I believe it would be useful to augment Tab 1 with this information on what set of ground-truths and evaluation strategies each sub-benchmark uses (i.e. LLM-as-judge, preferences, ground-truth etc).
>
> Thanks for the suggestion! This is indeed a good point, and we have added this information to Table 1 in this revision. Feel free to check!

---

> ### Author Response · Authors · 2024-11-18
>
> ## Concern 3
>
> > The challenge of test-set contamination: From the construction of the benchmark, it is not made abundantly clear how the proposed benchmark alleviates the problem of test-set contamination---as I understand, it is merely a combination of pre-existing benchmark samples in a principled manner by aligning with the real-world task distribution? If this is the case, it does indeed ensure a more robust distribution in the evaluation splits as well as better alignment to real-world usage, but it does not actually alleviate the concern of test-set contamination. A broad discussion point on this would be useful.
>
> Nice suggestion! Below we provide more illustrations on the dynamism and contamination concerns of MixEval-X:
>
> First of all, we wish to highlight that MixEval-X is only mitigating the contamination instead of solving it completely, as mentioned in line 161 of the originally submitted paper.
>
> There’re basically two kinds of potential contaminations: **natural contamination**, meaning the possible existence of evaluation tasks in the pre-training data; and **deliberate contamination**, meaning that model developers deliberately add the evaluation data to the training data to raise the model rankings on the leaderboard.
>
> **For natural contamination, MixEval-X mitigates it via benchmark mixture and contamination detection**. The effectiveness of benchmark mixture in mitigating the natural contamination has been illustrated in [1]. According to [1], contamination levels of existing benchmarks range from 1.1% to 40.6%. Generally, more popular benchmarks exhibit higher contamination. For example, MMLU shows a relatively high contamination ratio (24.3%), yet remains crucial to the community and the benchmark pool. They addressed this by mixing popular benchmarks with less contaminated ones smartly (e.g., CommonsenseQA), thus reducing the natural contamination ratio. In MixEval-X, the benchmark mixture similarly mitigates the natural contamination. Additionally, we also included contamination detection in our pipeline to exclude the seriously contaminated samples.
>
> **For deliberate contamination, MixEval-X mitigates it by dynamically updating web user queries and the benchmark pool with the automatic pipeline**. Note that if model developers deliberately overfit evals, contamination is nearly impossible to fully eliminate. Even with dynamic systems like Chatbot Arena, evaluations can still be hacked, e.g., fitting on LMSys user data or hiring biased workers. Developers may hack MixEval-X by (1) directly fitting on MixEval-X data, or (2) fitting the benchmark pool. We address method (1) by periodically updating MixEval-X data points through "batch web query update" (sampling new web query batches from the crawled web query pool) or "source web query update" (updating the whole web query pool with the latest Common Crawl), and then perform benchmark mixture. Method (2) is tackled by "benchmark pool update", incorporating new ground-truth benchmarks in the community, e.g., replacing MMMU with MMMU-pro, which also helps to mitigate natural contaminations.
>
>
>
>
>
> ## Question 1
>
> > Where are the actual web-queries being mined from? Would it possible to add some details around this mining strategy as well as add a few of these web-queries to give a feel of what the real-world task distribution looks like?
>
> Thanks for raising this important question! **The web queries are mined from the CommonCrawl with the same pipeline as in MixEval [2], and the mining details can be found in [2].** Below we show some examples from our web query pool:
>
> 1. Explain how electromagnetic radiations are produced using this set-up.
>
> 2. Could this really be Cuba or perhaps its Yacht Heaven?
>
> 3. What proportion of the maple trees had a trunk circumference greater than 76cm?
>
> Generally, real-world queries are noisier and much more diverse than that of existing benchmarks (as shown in the Figure 9 of the originally submitted paper), representing real-world user intents and query topics.
>
> ## Question 2
>
> > For the non-hard version of the benchmarks, the best models already perform quite well (76.9%, 74.2%, 62.7%). Does this mean that this version of the benchmark is quite easy for current frontier models already, and has a lower saturation ceiling? Could the authors please comment on this.
>
> From my perspective, 76.9% is not a high score for frontier models, which still indicates a substantial room for improvement. Generally, when the top model achieves 90+ performance on a benchmark, we consider it starting to saturate.
>
> ## References
>
> [1] Ni, Jinjie. “Don’t Build Random Evals: Principles for General-Purpose Model Evaluation”. https://beneficial-chips-08e.notion.site/Don-t-Build-Random-Evals-Principles-for-General-Purpose-Model-Evaluation-bd5a85ba10f447bc9ac560050f67270b
>
> [2] Ni, Jinjie et. al. “MixEval: Deriving Wisdom of the Crowd from LLM Benchmark Mixtures”

---

> > ### Comment · Reviewer_Edzp · 2024-11-19
> > **Response to rebuttal**
> >
> > Thank you for the detailed response, the rebuttal has answered most of my concerns. I still have a few outstanding questions/comments.
> >
> > > For natural contamination, MixEval-X mitigates it via benchmark mixture and contamination detection. The effectiveness of benchmark mixture in mitigating the natural contamination has been illustrated in [1].
> >
> > I could not find the citation [1] (Ni, Jinjie. “Don’t Build Random Evals: Principles for General-Purpose Model Evaluation”) anywhere with a brief search on scholar/arxiv. Could the authors please share the link to this paper?
> >
> > > For deliberate contamination, MixEval mitigates it by dynamically updating web user queries and the benchmark pool with the automatic pipeline.
> >
> > This seems like a reasonable strategy. How frequently are the authors planning to update (1) the benchmark's web source pool? and (2) replace existing benchmarks with new benchmarks? Essentially what is the threshold point at which a particular benchmark, or a pool of samples is identified to be contaminated?
> > Further, please add a discussion section on the paper regarding these points if not already done so, I believe it is an important set of points to improve the paper.

---

> > > ### Author Response · Authors · 2024-11-20
> > >
> > > Thanks for the comments! Below we answer the additional questions:
> > >
> > > > I could not find the citation [1] (Ni, Jinjie. “Don’t Build Random Evals: Principles for General-Purpose Model Evaluation”) anywhere with a brief search on scholar/arxiv. Could the authors please share the link to this paper?
> > >
> > > This is a notion blog written by the MixEval authors. Here's the link: https://beneficial-chips-08e.notion.site/Don-t-Build-Random-Evals-Principles-for-General-Purpose-Model-Evaluation-bd5a85ba10f447bc9ac560050f67270b
> > >
> > > You may scroll down to the bottom to click "A Contamination Analysis of MixEval" for the relevant contamination analysis.
> > >
> > > > This seems like a reasonable strategy. How frequently are the authors planning to update (1) the benchmark's web source pool? and (2) replace existing benchmarks with new benchmarks? Essentially what is the threshold point at which a particular benchmark, or a pool of samples is identified to be contaminated? Further, please add a discussion section on the paper regarding these points if not already done so, I believe it is an important set of points to improve the paper.
> > >
> > > Good questions and suggestions! At present, we plan to update the web source pool every six months and update the benchmark pool as long as the new acceptable benchmarks come out or a benchmark is considered to be contaminated. Generally, we consider a benchmark is contaminated if the contamination ratio exceeds 30%, and for important benchmarks, this threshold could be higher. We have added a discussion section in the revised paper (Section A.5). Feel free to check!

---

> > > > ### Comment · Reviewer_Edzp · 2024-11-25
> > > >
> > > > Having read all other reviews and the responses to my own questions, I am confident that this work is quite impactful and could potentially be useful for a lot of practitioners and model developers. I am increasing my confidence to 4 and highly recommend this paper for acceptance.

---

> > > > > ### Author Response · Authors · 2024-11-25
> > > > >
> > > > > Thank you for raising the confidence!
> > > > >
> > > > > And thank you once again for recognizing MixEval-X and for your insightful comments!

---

### Official Review · Reviewer_U4Qy · 2024-11-04

**Soundness:** 3
**Presentation:** 3
**Contribution:** 3
**Rating:** 6
**Confidence:** 3

**Summary:**

This paper focuses on introducing reliable evaluation methods to guide the development of multimodal large language models (LLMs). The authors extend MixEval to cover a wider range of tasks and incorporate additional modalities by proposing a multimodal benchmark mixture and adaptation-rectification pipelines to better reflect real-world task distributions. Furthermore, the paper benchmarks multimodal LLMs on their capabilities across text-to-X, X-to-text, and agent tasks, where X represents modalities such as images, videos, and more.

**Strengths:**

1. This approach is simple yet effective for creating a benchmark that encompasses a wide range of tasks.

2. The benchmark has been shown to align closely with the distribution of real-world datasets.

**Weaknesses:**

1. Missing Any-to-Any Models: Several relevant and popular any-to-any models have been omitted from the benchmark, such as:

    CoDi: Any-to-Any Generation via Composable Diffusion

    NExT-GPT: Any-to-Any Multimodal LLM

    Including these models would provide a more comprehensive evaluation.

2. Discussion of Limitations: Although not mandatory, a discussion on the limitations of the study would be valuable. It could provide context for the results and suggest areas for future research.

3. Clarity of Benchmark Creation: The process of creating the benchmark is very difficult to follow. Including a diagram or visual representation would greatly aid understanding and improve the overall readability of the paper. Additionally, the method for generating the hard dataset is not well explained, and providing more detail would improve comprehension.

**Questions:**

1. Ensuring Accuracy of Web Queries: How is the accuracy of information retrieved from web queries verified or ensured?

2. Handling Temporal Shifts: Given that accurate information can change over time (e.g., updates in data or events), how does the system handle these temporal shifts to maintain reliability?

3. Preventing Data Leakage: How does the benchmarking process ensure that there is no data leakage, especially considering that LLMs may have been pretrained on data sourced from the web?

---

> ### Author Response · Authors · 2024-11-18
>
> We thank the reviewer for recognizing MixEval-X as a sound, simple yet effective work. Meanwhile, we understand the reviewer's concerns, which are also very important to us. Below we clarify.
>
> *Note: Unless otherwise specified, the mentioned line numbers denote that of the originally submitted paper version instead of the rebuttal revision version.*
>
> ## Concern 1
>
> > Missing Any-to-Any Models: Several relevant and popular any-to-any models have been omitted from the benchmark, such as:  CoDi: Any-to-Any Generation via Composable Diffusion,  NExT-GPT: Any-to-Any Multimodal LLM.  Including these models would provide a more comprehensive evaluation.
>
> Thanks for the pointers! We will carefully evaluate and discuss these models in our paper revisions. Meanwhile, we will keep updating the results of more single-modality, multi-modality, and any-to-any models on our leaderboard website.
>
> **In the revised pdf version, we have already added the discussion for CoDi, NExT-GPT, and other existing any-to-any models in the related work section, with proper citations. Feel free to check!** However, due to the large amount of work involved in running various any-to-any models through our 11 sub-benchmarks and the delay in mTurk workers to grade the MMG tasks, we may not be able to update the results timely during the discussion period. We will update the results to our leaderboard and paper once they are ready.
>
> Meanwhile, we wish to highlight two primary reasons for not including any-to-any models in the submitted paper, given the constraints of resources and paper space. **We underscore that the evaluated models in the paper are appropriate and sufficient for the submitted version.**
>
> 1. **Existing any-to-any models are not mature enough.** Most of these models have only recently been introduced, with inconsistent protocols, incomplete checkpoints and documentations. Notably, many models lack support for user query inputs, a key requirement for MixEval-X and an emerging trend in evaluation frameworks. Even among the models that do support query inputs, the input/output formats and inference configurations are often highly specialized, posing significant challenges for establishing consistent evaluation settings across models. Furthermore, these models exhibit unstable and suboptimal performance on benchmarks, yielding results that are not robust enough for direct comparison in this study.
>
> 2. **The current set of evaluated models is both comprehensive and sufficient.** We conducted extensive evaluations across each sub-benchmark, encompassing a total of 125 models and covering nearly all prominent models within each modality. This large-scale effort required substantial human labor and significant computational resources. It is important to note that including only the most relevant models is common practice in benchmark studies [1][2][3], especially within the resource and paper space constraints.
>
> ## Concern 2
>
> > Discussion of Limitations: Although not mandatory, a discussion on the limitations of the study would be valuable. It could provide context for the results and suggest areas for future research.
>
> Thanks for the valuable suggestion! As a kind reminder, we discussed the limitations of this work in the form of frequently asked questions in Section A FREQUENTLY ASKED QUESTIONS of the submitted paper. **Beyond that, we now also include a separate limitation section in the revised pdf (Section B), feel free to check that!**
>
> ## Concern 3
>
> > Clarity of Benchmark Creation: The process of creating the benchmark is very difficult to follow. Including a diagram or visual representation would greatly aid understanding and improve the overall readability of the paper. Additionally, the method for generating the hard dataset is not well explained, and providing more detail would improve comprehension.
>
> Thanks for the kind suggestion! We also noticed that a diagram would help understand the whole pipeline. **Therefore, a diagram has been updated in the revised pdf (Figure 2), feel free to check!**
>
> Note that the hard dataset sampling method is the same as that of MixEval [4], which may take a significant space to illustrate in the paper. To save space, we gave the pointer instead (line 155 of the originally submitted version).

---

> ### Author Response · Authors · 2024-11-18
>
> ## Question 1
>
> > Ensuring Accuracy of Web Queries: How is the accuracy of information retrieved from web queries verified or ensured?
>
> As illustrated in section 2.1, line 141 of the originally submitted paper, we reuse the web detection pipeline from the MixEval [4]. The quantitative results of the whole detection pipeline, as reported in MixEval, is shown below (on their devised benchmarks). Before being trained, a language model achieving high recall was chosen (Vicuna-33B), with 99.12% recall and 46.21% precision; The looped training significantly improves the precision while maintaining the recall, illustrating the low error accumulation rate of the devised web query detection pipeline.
>
> Table 1: The breakdown metrics of the web query detection pipeline.
> | Model           | Param | Pipeline Recall | Pipeline Precision | Pipeline F1 |
> |-----------------|-----------------|--------------|-----------|-----------|
> | Web Detector (initial) | 33B            | 99.12          | 46.21      | 63.03      |
> | Web Detector (trained)        | 33B            | 99.55          | 98.61      | 99.07      |
>
>
> ## Question 2
>
> > Handling Temporal Shifts: Given that accurate information can change over time (e.g., updates in data or events), how does the system handle these temporal shifts to maintain reliability?
>
> This is an important question. Though most of the knowledge or reasoning tasks in the dataset are not affected from the temporal shift, i.e., the correct answer does not change with the time, some tasks do suffer from temporal shifts. This is indeed an issue for traditional static benchmarks. However, as illustrated in Section 2.1 of the paper, MixEval-X is dynamic, where one of the important features is that its benchmark pool will be updated with time, **meaning that it’s easy for us to replace the temporal shifted datapoints/benchmarks in the benchmark pool with the latest ones**, effectively mitigating the temporal shift issues.
>
> ## Question 3
>
> > Preventing Data Leakage: How does the benchmarking process ensure that there is no data leakage, especially considering that LLMs may have been pretrained on data sourced from the web?
>
> Thanks for raising this important question. This question can be interpreted as: How does MixEval-X deal with the data contamination issues? Below we clarify that in detail:
>
> First of all, we wish to highlight that MixEval-X is only mitigating the contamination instead of solving it completely, as mentioned in line 161 of the originally submitted paper.
>
> There’re basically two kinds of potential contaminations: **natural contamination**, meaning the possible existence of evaluation tasks in the pre-training data; and **deliberate contamination**, where model developers deliberately add the evaluation data to the training data to raise the model rankings on the leaderboard.
>
> **For natural contamination, MixEval-X mitigates it via benchmark mixture and contamination detection**. The effectiveness of benchmark mixture in mitigating the natural contamination has been illustrated in [1]. According to [1], contamination levels of existing benchmarks range from 1.1% to 40.6%. Generally, more popular benchmarks exhibit higher contamination. For example, MMLU shows a relatively high contamination ratio (24.3%), yet remains crucial to the community and the benchmark pool. They addressed this by mixing popular benchmarks with less contaminated ones smartly (e.g., CommonsenseQA), thus reducing the natural contamination ratio. In MixEval-X, the benchmark mixture similarly mitigates the natural contamination. Additionally, we also included contamination detection in our pipeline to exclude the seriously contaminated samples.
>
> **For deliberate contamination, MixEval-X mitigates it by dynamically updating web user queries and the benchmark pool with the automatic pipeline**. Note that if model developers deliberately overfit evals, contamination is nearly impossible to fully eliminate. Even with dynamic systems like Chatbot Arena, evaluations can still be hacked, e.g., fitting on LMSys user data or hiring biased workers. Developers may hack MixEval-X by (1) directly fitting on MixEval-X data, or (2) fitting the benchmark pool. We address method (1) by periodically updating MixEval-X data points through "batch web query update" (sampling new web query batches from the crawled web query pool) or "source web query update" (updating the whole web query pool with the latest Common Crawl), and then perform benchmark mixture. Method (2) is tackled by "benchmark pool update", incorporating new ground-truth benchmarks in the community, e.g., replacing MMMU with MMMU-pro, which also helps to mitigate natural contaminations.
>
>
> **If the above clarifications and revisions look good to you, please kindly consider raising the score. Your support is very important to us. Thank you!**

---

> ### Author Response · Authors · 2024-11-18
>
> ## References
>
> [1] Yue, Xiang et. al. “MMMU: A Massive Multi-discipline Multimodal Understanding and Reasoning Benchmark for Expert AGI”
>
> [2] Fu, Chaoyou et. al. “Video-MME: The First-Ever Comprehensive Evaluation Benchmark of Multi-modal LLMs in Video Analysis”
>
> [3] Chiang, Wei-Lin et. al. “Chatbot Arena: An Open Platform for Evaluating LLMs by Human Preference”
>
> [4] Ni, Jinjie et. al. “MixEval: Deriving Wisdom of the Crowd from LLM Benchmark Mixtures”
>
> [5] Ni, Jinjie. “Don’t Build Random Evals: Principles for General-Purpose Model Evaluation”. https://beneficial-chips-08e.notion.site/Don-t-Build-Random-Evals-Principles-for-General-Purpose-Model-Evaluation-bd5a85ba10f447bc9ac560050f67270b

---

> > ### Author Response · Authors · 2024-11-24
> > **Gentle Reminder for Discussion on Rebuttal.**
> >
> > Dear Reviewer,
> >
> > Thank you for your dedication and time spent reviewing our paper.
> >
> > As the discussion period draws to a close, we wish to remind you of the approaching deadline.
> >
> > Please feel free to raise any points or questions about the paper that may need additional clarification.
> >
> > We deeply appreciate your valuable feedback.
> >
> > Thank you once again.

---

> ### Comment · Reviewer_U4Qy · 2024-11-25
>
> Thanks for addressing my concerns, I've raised my score to 6.

---

> > ### Author Response · Authors · 2024-11-29
> >
> > Thank you once again for recognizing MixEval-X and for your insightful comments! Your support is very important to us!

---

### Official Review · Reviewer_xVxi · 2024-11-04

**Soundness:** 4
**Presentation:** 3
**Contribution:** 4
**Rating:** 8
**Confidence:** 4

**Summary:**

The paper introduces MixEval-X, a large benchmark which can be used for evaluating large multimodal models in large number of multimodal tasks, ranging from understanding to generation. The paper also shows an extensive evaluation of several existing approaches to the considered tasks, providing a reference for future works. It also introduced a carefully designed pipeline for the evaluation of different models.

**Strengths:**

- The evaluation provided in the paper is extensive, both considering the number of tasks and models evaluated. This paper can really become a reference for many future works.
- The tasks considered are handling very different modalities, which can foster the adoption of this dataset from different AI sub-communities.
- The user studies are carefully designed and involved a high number of annotators.

**Weaknesses:**

The paper is very solid, with few weak points. It presents several contributions. This can make it somewhat challenging to read due to the frequent cross-references throughout the document. However, I see this as a minor issue, largely due to the richness of the content.

**Questions:**

- The discussion about dynamism and contamination is not very clear (L153). Could the authors clarify this?

**Details Of Ethics Concerns:**

The paper uses human evaluators to assess model outputs, particularly for multi-modal generation. However this is appropriately discussed in the manuscript. Similarly, the paper introduces a new benchmark, based on existing datasets. Legal and discrimination issues can arise if the original datasets suffer from these problems.

---

> ### Author Response · Authors · 2024-11-18
>
> We thank the reviewer for the strong support and for considering this work to be solid! Below we address the raised concerns.
>
> *Note: Unless otherwise specified, the mentioned line numbers denote that of the originally submitted paper version instead of the rebuttal revision version.*
>
> ## Question 1
>
> > The discussion about dynamism and contamination is not very clear (L153). Could the authors clarify this?
>
> Sure! Below are more illustrations on the dynamism of MixEval-X:
>
> First of all, we wish to highlight that MixEval-X is only mitigating the contamination instead of solving it completely, as mentioned in line 161 of the originally submitted paper.
>
> There’re basically two kinds of potential contaminations: natural contamination, meaning the possible existence of evaluation tasks in the pre-training data; and deliberate contamination, meaning that model developers deliberately add the evaluation data to the training data to raise the model rankings on the leaderboard.
>
> **For natural contamination, MixEval-X mitigates it via benchmark mixture and contamination detection**. The effectiveness of benchmark mixture in mitigating the natural contamination has been illustrated in [1]. According to [1], contamination levels of existing benchmarks range from 1.1% to 40.6%. Generally, more popular benchmarks exhibit higher contamination. For example, MMLU shows a relatively high contamination ratio (24.3%), yet remains crucial to the community and the benchmark pool. They addressed this by mixing popular benchmarks with less contaminated ones smartly (e.g., CommonsenseQA), thus reducing the natural contamination ratio. In MixEval-X, the benchmark mixture similarly mitigates the natural contamination. Additionally, we also included contamination detection in our pipeline to exclude the seriously contaminated samples.
>
> **For deliberate contamination, MixEval-X mitigates it by dynamically updating web user queries and the benchmark pool with the automatic pipeline**. Note that if model developers deliberately overfit evals, contamination is nearly impossible to fully eliminate. Even with dynamic systems like Chatbot Arena, evaluations can still be hacked, e.g., fitting on LMSys user data or hiring biased workers. Developers may hack MixEval-X by (1) directly fitting on MixEval-X data, or (2) fitting the benchmark pool. We address method (1) by periodically updating MixEval-X data points through "batch web query update" (sampling new web query batches from the crawled web query pool) or "source web query update" (updating the whole web query pool with the latest Common Crawl), and then perform benchmark mixture. Method (2) is tackled by "benchmark pool update", incorporating new ground-truth benchmarks in the community, e.g., replacing MMMU with MMMU-pro, which also helps to mitigate natural contaminations.
>
> ## Ethics Concerns
>
> > The paper uses human evaluators to assess model outputs, particularly for multi-modal generation. However this is appropriately discussed in the manuscript. Similarly, the paper introduces a new benchmark, based on existing datasets. Legal and discrimination issues can arise if the original datasets suffer from these problems.
>
> As highlighted by the reviewer, we have addressed ethical concerns comprehensively in the end of the main text. Our work adheres to the principles of ethical AI research, emphasizing fairness, transparency, and accountability to contribute meaningfully to the community.
>
> Furthermore, we have verified that all datasets utilized in this study are publicly available and free from legal or discriminatory concerns.
>
> ## References
>
> [1] Ni, Jinjie. “Don’t Build Random Evals: Principles for General-Purpose Model Evaluation”. https://beneficial-chips-08e.notion.site/Don-t-Build-Random-Evals-Principles-for-General-Purpose-Model-Evaluation-bd5a85ba10f447bc9ac560050f67270b

---

> > ### Comment · Reviewer_xVxi · 2024-11-24
> > **Response to reviewer**
> >
> > I have read all reviewers comments and the response to reviewers'comments. The authors have addressed all my concerns and provided clarifications about contamination. In my opinion this is a very solid piece of work.

---

> > > ### Author Response · Authors · 2024-11-25
> > >
> > > Thank you once again for recognizing MixEval-X and for your insightful comments!

---

### Comment · Area_Chair_VxpA · 2024-11-22
**Discussion period**

Dear reviewers,

The authors have responded to your reviews.

Until November 26th @ 2359 (AOE time) reviewers and authors can freely exchange responses, so if there any clarifications you require from the authors, now is the time to seek them!

Best,

AC

---

### Meta-Review · Area_Chair_VxpA · 2024-12-11

**Metareview:**

This paper proposes MixEval-X, which is a benchmark that can be used to evaluate multi-modal models on a variety of different tasks with different input and output modalities. This paper received very good final scores (8,8,8,6) and was appreciated by all reviewers. Most weaknesses identified related to discussions and clarity and the authors satisfactorily addressed these in their responses.

I see no grounds for rejection, and believe this paper should be highlighted at the conference as a spotlight because of the interest received from the reviewers.

**Additional Comments On Reviewer Discussion:**

During the discussion period, Reviewer U4Qy was satisfied by the author responses and raised their score. Given that the response to this paper from all reviewers was almost entirely positive, I don't feel there is much more to add here and that the decision was straightforward.

---

### Decision · Program_Chairs · 2025-01-22

Accept (Spotlight)